# Near-optimal Offline and Streaming Algorithms for Learning Non-Linear Dynamical Systems

**Prateek Jain**
Google AI Research Lab,
Bengaluru, India 560016
prajain@google.com

**Suhas S Kowshik**
Department of EECS
MIT,
Cambridge, MA 02139
suhask@mit.edu

**Dheeraj Nagaraj**
Department of EECS
MIT,
Cambridge, MA 02139
dheeraj@mit.edu

**Praneeth Netrapalli**
Google AI Research Lab,
Bengaluru, India 560016
pnetrapalli@google.com

## Abstract

We consider the setting of vector valued non-linear dynamical systems $X_{t+1} = \phi(A^* X_t) + \eta_t$, where $\eta_t$ is unbiased noise and $\phi : \mathbb{R} \to \mathbb{R}$ is a known link function that satisfies certain *expansivity property*. The goal is to learn $A^*$ from a single trajectory $X_1, \cdots, X_T$ of *dependent or correlated* samples. While the problem is well-studied in the linear case, where $\phi$ is identity, with optimal error rates even for non-mixing systems, existing results in the non-linear case hold only for mixing systems. In this work, we improve existing results for learning nonlinear systems in a number of ways: a) we provide the first offline algorithm that can learn non-linear dynamical systems without the mixing assumption, b) we significantly improve upon the sample complexity of existing results for mixing systems, c) in the much harder one-pass, streaming setting we study a SGD with Reverse Experience Replay (SGD − RER) method, and demonstrate that for mixing systems, it achieves the same sample complexity as our offline algorithm, d) we justify the expansivity assumption by showing that for the popular ReLU link function — a non-expansive but easy to learn link function with i.i.d. samples — any method would require exponentially many samples (with respect to dimension of $X_t$) from the dynamical system. We validate our results via simulations and demonstrate that a naive application of SGD can be highly sub-optimal. Indeed, our work demonstrates that for correlated data, specialized methods designed for the dependency structure in data can significantly outperform standard SGD based methods.

## 1 Introduction

Non-linear dynamical systems (NLDS) are commonly used to model the data in a variety of domains like control theory, time-series analysis, and reinforcement learning (RL) [1–4]. Standard NLDS models the data points $(X_0, X_1, \ldots, X_T)$ as:

$$X_{t+1} = \phi(A^* X_t) + \eta_t, \tag{1}$$

where $X_t \in \mathbb{R}^d$ are the states, $\eta_t \in \mathbb{R}^d$ are i.i.d. noise vectors, $A^* \in \mathbb{R}^{d \times d}$ and $\phi : \mathbb{R} \to \mathbb{R}$ is an increasing function called the 'link function'. Here, $\phi$ is supposed to act component wise over $\mathbb{R}^d$.

*System identification* problem is a foundational problem for NLDS, i.e., given $(X_0, X_1, \ldots, X_T)$ generated from (1), the goal is to estimate $A^*$ accurately from a single trajectory $(X_0, X_1, \ldots, X_T)$.

35th Conference on Neural Information Processing Systems (NeurIPS 2021).

The system identification problem is heavily studied in control theory [5–8] as well as time-series analysis [9]. For instance, the non-linear dynamical system considered here has an application in modeling non-linear distortions in power amplifiers [10]. The problem is challenging as data points $X_0, X_1, \ldots, X_T$ are not i.i.d. as usually encountered in machine learning, but form a Markov process. If the mixing time $\tau_{\mathsf{mix}}$ of the process is finite ($\tau_{\mathsf{mix}} < \infty$), then we can make the data approximately i.i.d. by considering only the points separated by $\tilde{O}(\tau_{\mathsf{mix}})$ time. While this allows using standard techniques for i.i.d. data, it reduces the effective number of samples to $O(\frac{T}{\tau_{\mathsf{mix}}})$, which typically gives an error of the order $O(\frac{\tau_{\mathsf{mix}}}{T})$. In fact, even the state-of-the-art results have error bounds which are sub-optimal by a factor of $\tau_{\mathsf{mix}}$.

Interestingly, for the special case of linear systems, i.e., when $\phi(x) = x$, the results are significantly stronger. For example, [11, 12] showed that the matrix $A^*$ can be estimated with an error $O(1/T)$ even when the mixing time $\tau_{\mathsf{mix}} > T$. But these results rely on the fact that for linear systems, the estimation problem reduces to an ordinary least squares (OLS) problem for which a closed form expression is available and can be analyzed effectively.

On the other hand, NLDS do not admit such closed form expressions. In fact the existing techniques mostly rely on mixing time arguments to induce i.i.d. like behavior in a subset of the points which leads to sub-optimal rates by $\tau_{\mathsf{mix}}$ factor. Similarly, a direct application of uniform convergence results [13] to show that the minimizer of the empirical risk is close to the population minimizer still gives sub-optimal rates as off-the-shelf concentration inequalities (cf. [14]) incur an additional factor of mixing time. Finally, existing results are mostly focused on offline setting, and do not apply to the case where the data points are streaming which is critical in several practical problems like reinforcement learning (RL) and control theory.

In this work, we provide algorithms and their corresponding error rates for the NLDS system identification problem in both offline and online setting, assuming the link function to be expansive (Assumption 1). The main highlight of our results is that the error rates are *independent* of the mixing time $\tau_{\mathsf{mix}}$, which to the best of our knowledge is first such result for any non-linear system identification in any setting. In fact, for offline setting, our analysis holds even for systems which do not mix within time $T$ and even for marginally stable systems which do not mix at all. Furthermore, we analyze SGD-Reverse Experience Replay (SGD-RER) method, we provide the first streaming method for NLDS identification with error rate that is independent of $\tau_{\mathsf{mix}}$ (in the leading order term) while still ensuring small space and time complexity. This algorithm was first discovered in the experimental RL setting in [15] based on Hippocampal reverse replay observed in biological networks [16–18]. It was introduced independently in [19] for the case of linear systems and efficiently unravels the complex dependency structure present in the problem. Finally, through a lower bound for ReLU— a non-expansive function—we provide strong justification for why expansivity might be necessary for a non-trivial result.

Instead of mixing time arguments, our proofs for learning NLDS without mixing use a natural exponential martingale of the kind considered in the analysis of self normalized process ([20, 21]). For streaming setting, while we do use mixing time arguments (proof of Theorems 2 and 3), we combine them with a delicate stability analysis of the specific algorithm and the machinery developed in [19] to obtain strong error bounds. See Section 6 for a description of these techniques.

**Our Contributions.** Key contributions of the paper are summarized below:

1. Assuming expansive and monotonic link function $\phi$ and sub-Gaussian noise, we show that the offline Quasi Newton Method (Algorithm 1) estimates the parameter $A^*$ with near optimal errors of the order $O(1/T)$, even when the dynamics does not mix within time $T$.

2. Assuming mixing NLDS, finite fourth moment on the noise, and expansive monotonic link function, we show that offline Quasi Newton Method again estimates the parameter $A^*$ with near-optimal error of $O(1/T)$, independent of mixing time $\tau_{\mathsf{mix}}$.

3. We give a one-pass, streaming algorithm inspired by $\mathsf{SGD - RER}$ method by [19], and show that it achieves near-optimal error rates under the assumption of sub-Gaussian noise, NLDS stability (see section 2.1 for the definition), uniform expansivity and second differentiability of the link function.

4. We then show that learning with ReLU link function, which is non-expansive but is known to be easy to learn with if data points are all i.i.d. [22], requires exponential (in $d$) many samples.

We believe that the techniques developed in this work can be extended to provide efficient algorithms for learning with dependent data in more general settings.

**Related Works.** NLDS has been studied in a variety of domains like time-series and recurrent neural networks (RNN). [9] studies specific NLDS models from a time series perspective and establishes non-asymptotic convergence bounds for natural estimators; their error rates suffer from mixing time factor $\tau_{\text{mix}}$. [23] considers asymptotic learning of NLDS via neural networks trained using SGD, whereas [24] shows that overparametrized LSTMs trained with SGD learn to memorize the given data. [25–27] consider learning dynamical systems of the form $h_{t+1} = \phi(A^* h_t + B^* u_t)$ for states $h_t$ and inputs $u_t$; this setting is different from standard NLDS model we study. [28] considers the non linear dynamical systems of the form $x_{t+1} = A\phi(x_t, u_t) + \eta_t$ which $\phi$ is a known non-linearity and matrix $A$ is to be estimated. [29, 30] consider essentially linear dynamics but allow for certain non-linearities that can be modeled as process noise. All these again differ from the model we consider.

Standard NLDS identification (1) has received a lot of attention recently, with results by [31, 32] being the most relevant. [31] uses uniform convergence results via. mixing time arguments to obtain parameter estimation error for offline SGD. [32] obtains similar bounds via. the analysis of the GLMtron algorithm [33]. However, both these works suffer from sub-optimal dependence on the mixing time. We refer to Table 1 for a comparison of the results. [34], which appeared after the initial manuscript of this work, considers the question from a perspective of time series forecasting. This work considers sparsity in $A^*$ and an unknown link function which is estimated with isotonic regression. Their recovery guarantees eschew the mixing time dependence. However, the setting, assumptions and the error rates are incomparable to our setting.

[32] also obtains within sample prediction error in the case when $\phi$ is not uniformly expansive along with parameter recovery bounds when $\phi$ is the ReLU function and the driving noise is Gaussian. However, the parameter estimation bounds for ReLU suffer from an exponential dependence on the dimension $d$ and mixing time $\tau_{\text{mix}}$. In Theorem 4 we establish that indeed we cannot improve the exponential dependence in the dimension $d$ for the case of parameter estimation. We note that the exponential dependence arises due to the dynamics present in the system since ReLU regression with isotropic i.i.d. data in well specified case has only a polynomial dependence in $d$ [22].

Linear system identification (LSI) literature has been well studied with strong minimax optimal bounds [11, 35, 36]. These results primarily consider the (convex) empirical square loss which has a closed form solution. However, the square loss in the non-linear case is non-convex. Under the assumption that the link function is increasing, we consider a convex proxy loss which is widely used in generalized linear regression literature [22, 33, 37]. Similarly, GLMtron algorithm for learning NLDS in[32] (see Equation (3)) also considers a similar proxy loss. In [38], the authors consider a family of GLMtron-like algorithms call Reflectron under the i.i.d. data setting. But they compare the performance of these algorithms experimentally on an NLDS similar to one considered in this work under low rank assumption on the system matrix.

Finally, streaming setting for LSI has been recently studied in different model settings [19, 39]. These methods observe that by exploiting techniques like experience replay ([40]) along with squared loss error, one can obtain strong error rates. $\mathsf{SGD} - \mathsf{RER}$ method studied in this work is inspired by a similar method by [19] which was primarily studied for the linear case.

## 2 Problem Statement

Let $\phi : \mathbb{R} \to \mathbb{R}$ be an increasing, 1-Lipschitz function such that $\phi(0) = 0$. Suppose $X_0 \in \mathbb{R}^d$ is a random variable and $A^* \in \mathbb{R}^{d \times d}$. We consider the following non-linear dynamical system (NLDS):

$$X_{t+1} = \phi(A^* X_t) + \eta_t, \tag{2}$$

where the noise sequence $\eta_0, \ldots, \eta_T$ is i.i.d random vectors independent of $X_0$. The noise $\eta_t$ is such that $\mathbb{E}\eta_t = 0$, $\mathbb{E}\eta_t \eta_t^\top = \sigma^2 I$ for some $\sigma > 0$. We will also assume that $M_4 := \mathbb{E}\|\eta_t\|^4 < \infty$. Let $\mu$ be the law of noise $\eta$. We denote the model above as $\mathsf{NLDS}(A^*, \mu, \phi)$. Whenever a stationary distribution exists for the process, we will denote it by $\pi(A^*, \mu, \phi)$ or just $\pi$ when the process is clear from context. We will call the trajectory $X_0, X_1, \ldots, X_T$ 'stationary' if $X_0$ is distributed according to the measure $\pi(A^*, \mu, \phi)$. Unless specified otherwise, we take $X_0 = 0$ almost surely.

| Paper | Guarantee | Link Function | System | Noise | Algorithm |
|---|---|---|---|---|---|
| [31] THEOREM 6.2 | $\frac{d^2\tau_{\text{mix}}}{T}$ | INCREASING,LIPSCHITZ EXPANSIVE | MIXING | SUB-GAUSSIAN | OFFLINE |
| [32] THEOREM 2 | $\frac{d^2\tau_{\text{mix}}}{T}$ | INCREASING,LIPSCHITZ EXPANSIVE | MIXING | SUB-GAUSSIAN | OFFLINE |
| THIS PAPER THEOREM 1 | $\frac{d^2\sigma^2}{T\lambda_{\min}(\hat{G})}$ | INCREASING,LIPSCHITZ EXPANSIVE | NON-MIXING | SUB-GAUSSIAN | OFFLINE |
| THIS PAPER THEOREM 2 | $\frac{d^2\sigma^2}{T\lambda_{\min}(G)}$ | INCREASING,LIPSCHITZ EXPANSIVE | MIXING | 4-TH MOMENT | OFFLINE |
| THIS PAPER THEOREM 3 | $\frac{d^2\sigma^2}{T\lambda_{\min}(G)}$ | INCREASING,LIPSCHITZ EXPANSIVE BOUNDED SECOND DERIVATIVE | MIXING | SUB-GAUSSIAN | STREAMING |

Table 1: Comparison of our results with existing results in terms of mixing time $\tau_{\text{mix}}$, stablility and number of samples $T$. Here, we take $\tau_{\text{mix}} = \tilde{\Omega}(\frac{1}{1-\|A^*\|_{op}})$ as a proxy for the mixing time. Note that $\lambda_{\min}(G) \geq \sigma^2$ in the worst case, and hence our bounds are better by a factor of $\tau_{\text{mix}}$.

The goal is to estimate $A^*$ given a single trajectory $X_0, X_1, \ldots, X_T$. A natural approach would be to minimize the empirical square loss, i.e, $\mathcal{L}_{\text{sq}}(A; X) := \frac{1}{T} \sum_{t=0}^{T-1} \|\phi(AX_t) - X_{t+1}\|^2$. However, when the link function $\phi$ is not linear, then this would be non-convex and hard to optimize. Instead, we use a convex proxy loss given by:

$$\mathcal{L}_{\text{prox}}(A; X) = \frac{1}{T} \sum_{t=0}^{T-1} \sum_{i=1}^{d} \bar{\phi}(\langle a_i, X_t\rangle) - \langle e_i, X_{t+1}\rangle\langle a_i, X_t\rangle, \tag{3}$$

where $\bar{\phi}$ is the indefinite integral of the link function $\phi$ and $a_i$ is the $i$-th row of $A$. Note that the gradient of $\mathcal{L}_{\text{prox}}(A; X)$ with respect to $A$ is given by:

$$\nabla\mathcal{L}_{\text{prox}}(A; X) = \frac{1}{T} \sum_{t=0}^{T-1} (\phi(AX_t) - X_{t+1}) X_t^\top. \tag{4}$$

When the model is clear from context and the stationary distribution exists, we will denote the second moment matrix under the stationary distribution by $G := \mathbb{E}[X_t X_t^\top]$. Note that $G \succeq \mathbb{E}[\eta_t \eta_t^\top] = \sigma^2 I$. Also, the empirical second moment matrix is denoted by $\hat{G} := \frac{1}{T} \sum_{t=0}^{T-1} X_t X_t^\top$.

## 2.1 Assumptions

We now state the assumptions below and use only a subset of the assumptions for each result.

**Assumption 1** (Lipschitzness and Uniform Expansivity). *$\phi$ is 1-Lipschitz and $|\phi(x) - \phi(y)| \geq \zeta|x - y|$, for some $\zeta > 0$.*

Note that when $\phi$ is only weakly differentiable but satisfy Assumption 1, with a slight abuse of notation, we will write down $\phi(x) - \phi(y) = \phi'(\beta)(x - y)$ for some $\phi'(\beta) \in [\zeta, 1]$.

**Assumption 2** (Bounded 2nd Derivative). *$\phi$ is twice continuously differentiable and $|\phi''|$ is bounded.*

**Assumption 3** (Noise Sub-Gaussianity). *For any unit norm vector $x \in \mathbb{R}^d$, we have $\langle \eta_t, x\rangle$ to be sub-Gaussian with variance proxy $C_\eta \sigma^2$.*

Next, we extend the definition of exponential stability in [31] to 'exponential regularity' to allow unstable systems.

**Assumption 4** (Exponential Regularity). *Let $X_T = h_{T-1}(X_0, \eta_0, \ldots, \eta_T)$ be the function representation of $X_T$. We say that $\mathsf{NLDS}(A^*, \mu, \phi)$ is $(C_\rho, \rho)$ exponentially regular if for any choice of $T \in \mathbb{N}$ and $X_0, X_0', \eta_0, \ldots, \eta_T \in \mathbb{R}^d$:*

$$\|h_T(X_0, \eta_0, \ldots, \eta_T) - h_T(X_0', \eta_0, \ldots, \eta_T)\|_2 \leq C_\rho\rho^{T-1}\|X_0 - X_0'\|_2.$$

*When $\rho < 1$, we will call the system stable. When $\rho = 1$ we will call it 'possibly marginally stable' and when $\rho > 1$, we will call it 'possibly unstable'.*

---

**Algorithm 1:** Quasi Newton Method

---

**Input** : Offline data $\{X_0, \ldots, X_T\}$, horizon $T$, no. of iterations $m$, link function $\phi$, step size $\gamma$

**Output**: Estimate $A_m$

1 **begin**

2     $A_0^0 = 0$ /*Initialization*/

3     $\hat{G} \leftarrow \frac{1}{T}\sum_{t=0}^{T-1} X_t X_t^\top$; If $\hat{G}$ is not invertible, then **return** $A_m = 0$

4     **for** $i \leftarrow 0$ **to** $m-1$ **do**

6        $A_{i+1} \leftarrow A_i - 2\gamma\left(\nabla\mathcal{L}_{\mathsf{prox}}(A_i; X)\right)\hat{G}^{-1}$

---

Note that when Assumption 4 holds with $\rho < 1$, the system necessarily mixes and converges to a stationary distribution as $T \to \infty$. Such systems forget their initial conditions in time scales of the order $\tau_{\mathsf{mix}} = O\big(\frac{1+\log C_\rho}{\log\frac{1}{\rho}}\big) = O\left(\frac{1+\log C_\rho}{1-\rho}\right)$, and hence we use this as a proxy for the mixing time. In what follows, when we say 'the system does not mix' we either mean that it does not mix within time $T$ or it does not converge to a stationary distribution (ex: $\rho \geq 1$).

**Assumption 5** (Norm Boundedness). $\|A^*\|_{\mathsf{op}} = \rho < 1$

That is, if $A^*$ satisfies Assumption 5, we have for arbitrary $X, X' \in \mathbb{R}^d$: $\|\phi(A^*X) - \phi(A^*X')\| \leq \rho\|X - X'\|$ and $\big\|(\phi \circ A^*)^k(X)\big\| \leq \rho^k\|X\|$. Hence, for such $A^*$, NLDS is *necessarily stable*.

## 3 Offline Learning with Quasi Newton Method

In this section we consider estimating $A^*$ using a single trajectory $(X_1, \ldots, X_T)$ from $\mathsf{NLDS}(A^*, \mu, \phi)$. To this end, we study an offline Quasi Newton Method (Algorithm 1) where the iterates descend in the directions of the gradient of $\mathcal{L}_{\mathsf{prox}}$ normalized by the inverse of the empirical second moment matrix $\hat{G} := \frac{1}{T}\sum_{t=0}^{T-1} X_t X_t^\top$. That is, the iterates follow an approximation of the standard Newton update.

We now present analysis of Algorithm 1 in two settings: a) Theorem 1 provides estimation error for possibly unstable systems with sub-Gaussian noise that is close to the minimax optimal error incurred in the linear system identification case, b) Theorem 2 provides similarly tight estimation error for mixing systems but with heavy-tailed noise.

**Theorem 1** (Learning Without Mixing). *Suppose Assumptions 1, 3 and 4 hold with expansivity factor $\zeta$ and regularity parameters $(C_\rho, \rho)$. Let $\bar{C}, \bar{C}_3$ be constants depending only on $C_\eta$, and let $\delta \in (0, \frac{1}{2})$. Let $R^* := C_\rho^2 C_\eta d\sigma^2 \left(\sum_{t=1}^{T-1}\rho^t\right)^2 \log(\frac{4Td}{\delta})$, and assume*

    *1. The number samples $T \geq \bar{C}_3\left(d\log\left(\frac{R^*}{\sigma^2}\right) + \log\frac{1}{\delta}\right)$*

    *2. Step size $\gamma = \frac{1}{4}$*

    *3. $m \geq \frac{10}{\zeta} \cdot \log\left(\frac{\|A_0 - A^*\|_{\mathsf{F}}^2 \cdot TR^*}{\sigma^2 d^2}\right)$*

*Then, the output $A_m$ of Algorithm 1 after $m$ iterations and $\lambda_{\min}\left(\hat{G}\right)$ satisfy with probability at-least $\geq 1 - \delta$:*

$$\|A_m - A^*\|_{\mathsf{F}}^2 \leq \frac{\bar{C}\sigma^2}{T\zeta^2\lambda_{\min}(\hat{G})}\left[d^2\log\left(1 + \frac{R^*}{\sigma^2}\right) + d\log\left(\frac{2d}{\delta}\right)\right],$$

$$\lambda_{\min}\left(\hat{G}\right) \geq \frac{\sigma^2}{2}.$$

Note that as $\lambda_{\min}(\hat{G}) \gtrsim \sigma^2$, the error rate scales as $\approx d^2/T$, independent of $\tau_{\mathsf{mix}} \approx 1/(1-\rho)$. The theorem also holds for non-mixing or possibly unstable systems as long as $\rho < 1 + \frac{C}{T}$. Furthermore, the error bound above is similar to the *minimax optimal bound* by [11] for the *linear* setting, i.e., when

$\phi(x) = x$. As the link function $\phi$ tends to decrease the information in $x$, intuitively lower bound for linear setting should apply for NLDS as well, which would imply our error rate to be optimal; we leave further investigation into lower bound of NLDS identification for future work. Interestingly, in the linear case whenever the smallest singular value $\sigma_{\min}(A^*) > 1 + \epsilon$, it can be show than $\lambda_{\min}(\hat{G})$ grows exponentially with $T$, leading to an exponentially small error. It is not clear how to arrive at such a growth lower bound in the non-linear case.

The computational complexity of the algorithm scales as $m \cdot T$ which depends only logarithmically on $\tau_{\mathsf{mix}}$. Interestingly, the algorithm is almost hyperparameter free, and does not require knowledge of parameters $\sigma, \tau_{\mathsf{mix}}, \zeta$.

Also note that the stationary points of Algorithm 1 and GLMtron ( [32]) are the same. So, the stronger error rate in the result above compared to the result by [32] is due to a sharper analysis. However, in dynamical systems of the form 1, the squared norm of the iterates grow as $\frac{d}{1-\rho}$ even in the stable case. Hence, the GLMtron algorithm requires step sizes to be $\approx \frac{1-\rho}{d}$ which implies significantly slower convergence rate for large $\tau_{\mathsf{mix}} = 1/(1-\rho)$. In contrast, convergence rate for Algorithm 1 depends at most logarithmically on $\tau_{\mathsf{mix}}$.

**Theorem 2** (Learning with Heavy Tail Noise). *Suppose Assumptions 1 and 4 hold. In Assumption 4, let $\rho < 1$. Let $X_0, \ldots, X_T$ be a stationary trajectory drawn from* $\mathsf{NLDS}(A^*, \mu, \phi)$ *and $A_m$ be the $m$-th iterate of Algorithm 1. For some universal constants $C, C_1, C_0 > 0$, whenever $\delta \in (0, \frac{1}{2})$, $R^* := \frac{4TdC_\rho^2\sigma^2}{(1-\rho)^2\delta}$ and*

1. *$T \geq Cd\log(\frac{1}{\delta})\log(\frac{R^*}{\sigma^2})\max\left(\frac{4C_\rho^6 M_4}{(1-\rho)^4(1-\rho^2)\sigma^4}, \frac{\log\left(\frac{R^*C_1C_\rho}{\sigma^2}\right)}{\log\left(\frac{1}{\rho}\right)}\right)$*

2. *Step size $\gamma = \frac{1}{4}$*

3. *$m \geq \frac{10}{\zeta} \cdot \log\left(\frac{\|A_0 - A^*\|_{\mathsf{F}}^2 \cdot TR^*}{\sigma^2 d^2}\right)$*

*There exists an event $\mathcal{W} \in \sigma(X_0, \eta_0, \ldots, \eta_{T-1})$ with $\mathbb{P}(\mathcal{W}) \geq 1 - \delta$ and :*

$$\mathbb{E}\left[\|A_m - A^*\|_{\mathsf{F}}^2 \mathbb{1}(\mathcal{W})\right] \leq \frac{C_0 d^2 \sigma^2}{\zeta^2 T \lambda_{\min}(G)}.$$

**Obtaining High Probability Bounds:** The bound above shows that the expectation of the error restricted to a high probability set is small. This, along with Markov inequality, shows that we can have an error of at-most $\frac{Cd^2\sigma^2}{\zeta^2 T \lambda_{\min}(G)}$ with probability at-least $\frac{2}{3}$. This can be boosted to a high probability bound by splitting the horizon $T$ into $K$ contiguous segments with a gap of $O(\tau_{\mathsf{mix}}\log T)$ to maintain approximate independence (see Section C.2). We then run the Quasi Newton method on each of these 'split' data sets to obtain nearly independent estimates $\hat{A}_1, \ldots, \hat{A}_K$, which each have error at-most $\frac{Cd^2\sigma^2 K}{\zeta^2 T \lambda_{\min}(G)}$ with probability at-least $\frac{2}{3}$. Using a standard high-dimensional median of means estimator (see [41, Algorithm 3]) for $\hat{A}_1, \ldots, \hat{A}_K$, we obtain error bounds of the order $\frac{Cd^2\sigma^2 K}{\zeta^2 T \lambda_{\min}(G)}$ with probability at least $1 - e^{-\Omega(K)}$.

We refer to Section 6 for a high-level exposition of the key ideas in the analysis and Section B for the full proof of Theorems 1 and 2.

## 4 Streaming Learning with SGD-RER

In this section, we consider the one-pass, streaming setting, where the data points are presented in a streaming fashion. The goal is to continuously produce better estimates of $A^*$ while also ensuring that the space and the time complexity of the algorithm is small. This disallows approaches that would just store all the observed points and then apply offline Algorithm 1 to produce strong estimation error. Such one-pass streaming algorithms are critical in a variety of settings like large-scale and online time-series analysis [42, 43], TD learning in RL [44], econometrics.

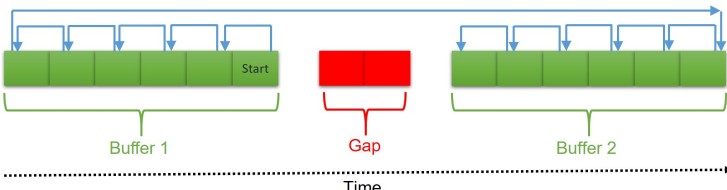

Figure 1: Data order in SGD − RER, where each block represents a data point. Blue arrows indicate the data processing order. The gaps ensure approximate independence between successive buffers.

---

**Algorithm 2:** SGD − RER

**Input** : Streaming data $\{X_\tau\}$, horizon $T$, buffer size $B$, buffer gap $u$, bound $R$, tail start: $t_0 \leq N/2$, link function $\phi$, step size $\gamma$

**Output** : Estimate $\hat{A}_{t_0,t}$, for all $t_0 \leq t \leq N-1$; $N = T/(B+u)$

1 **begin**

2      Total buffer size: $S \leftarrow B + u$, Number of buffers: $N \leftarrow T/S$

3      $A_0^0 = 0$ /\*Initialization\*/

4      **for** $t \leftarrow 1$ **to** $N$ **do**

5          Form buffer $\mathsf{Buf}^{t-1} = \{X_0^{t-1}, \ldots, X_{S-1}^{t-1}\}$, where, $X_i^{t-1} \leftarrow X_{(t-1)\cdot S+i}$

6          If $\exists i,\ s.t.,\ \left\|X_i^{t-1}\right\|^2 > R$, then **return** $\hat{A}_{t_0,t} = 0$

7          **for** $i \leftarrow 0$ **to** $B-1$ **do**

9              $A_{i+1}^{t-1} \leftarrow A_i^{t-1} - 2\gamma \left[\phi(A_i^{t-1} X_{S-i-1}^{t-1}) - X_{S-i}^{t-1}\right] X_{S-i-1}^{t-1,\top}$

10          $A_0^t = A_B^{t-1}$

11          If $t \geq t_0 + 1$, then $\hat{A}_{t_0,t} \leftarrow \frac{1}{t-t_0} \sum_{\tau=t_0+1}^{t} A_B^{\tau-1}$

---

To address this problem, we consider SGD − RER (Algorithm 2) which was introduced in [19] in the context of *linear system identification* (LSI). We apply the method for NLDS identification as well. SGD − RER uses SGD like updates, but the data is processed in a different order than it is received from the dynamical system. This algorithm is based on the observation made in [19] that for LSI, when SGD is run on the least squares loss in the forward order, there are spurious correlations which prevent the algorithm's convergence to the optimum parameter $A^*$. Surprisingly, considering the data in the reverse order *exactly* unravels these correlations to resolve the problem. Reverse order traversal of data, even though one pass, does not give a streaming algorithm. Hence, we divide the data into multiple buffers of size $B$ and leave of size $u$ between the buffers (See Figure 1). The data *within* each buffer is processed in the reverse order whereas the buffers themselves are processed in the order received. See Figure 1 for an illustration of the processing order. The gaps $u$ are set large enough so that the buffers behave approximately independently. Setting $B \geq 10u$ we note that this simple strategy improves the sample efficiency compared to naive data dropping since we use *most* of the samples for estimating $A^*$. We now present the main result for streaming setting.

**Theorem 3** (Streaming Algorithm). *Suppose Assumptions 1, 2, 3 and 5 hold and that the data points are stationary. Set $\alpha = 100$, $R = \frac{16(\alpha+2)dC_\eta\sigma^2 \log T}{1-\rho}$, $u \geq \frac{2\alpha \log T}{\log(\frac{1}{\rho})}$, $B \geq \left(\bar{C}_1 \frac{d}{(1-\rho)(1-\rho^2)} \log\left(\frac{d}{1-\rho}\right), 10u\right)$ for a global constant $\bar{C}_1$ dependent only $C_\eta$ and $\alpha$. Let $N = T/(B+u)$ be the number of buffers. Finally, set step-size $\gamma = \frac{C}{T^\nu}$ where $\nu = 6.5/7$ and let $T$ be large enough such that $\gamma \leq \min\left(\frac{\zeta}{4BR(1+\zeta)}, \frac{1}{2R}\right)$. If $N/2 > t_0 > c_1 \frac{\log T}{\zeta\gamma B\lambda_{\min}(G)} = \Theta(T^\nu \log T)$ for some large enough constant $c_1 > 0$, then output $\hat{A}_{t_0,N}$ of Algorithm 2 satisfies:*

$$\mathbb{E}\left[\|\hat{A}_{t_0,N} - A^*\|_{\mathsf{F}}^2\right] \leq C \frac{d^2\sigma^2 \log T}{T\lambda_{\min}(G)\zeta^2} + \text{Lower Order Terms} \tag{5}$$

*where $C$ is a constant dependent on $C_\eta$, $\alpha$.*

**Remark 1.** *The lower order terms are of the order $\mathrm{Poly}(R, B, \beta, 1/\zeta, 1/\lambda_{\min}, \|\phi''\|)\gamma^{7/2}T^2 + \gamma^2 R\sigma^2 d\frac{1}{T^{\alpha/2-2}} + \|A_0 - A^*\|_{\mathsf{F}}^2\left[\frac{e^{-c_2\zeta\gamma B\lambda_{\min}t_0}}{T\zeta\gamma\lambda_{\min}}\right]$. We refer to the proof in Section C.13 for details.*

**Remark 2.** *Although the bound in Theorem 3 is given for the algorithmic iterate at the end of the horizon, the proof shows that in fact we can bound the error of the iterates at the end of each buffer after $(1 + c)t_0$ i.e. if $t \geq (1 + c)t_0$ for some $c > 0$ then we obtain*

$$\mathbb{E}\left[\|\hat{A}_{t_0,t} - A^*\|_{\mathsf{F}}^2\right] \leq C \frac{d^2 \sigma^2 \log T}{(tB)\lambda_{\min}(G)\zeta^2} + \text{Lower Order Terms}$$

Note that the estimation error above matches the error by offline method up to log factors (see Theorem 2). Furthermore, while the method requires NLDS to be mixing, i.e., $\rho < 1$, but the leading term in error rate does not have an explicit dependence on it. Moreover, the space complexity of the method is only $B \cdot d$ which scales as $d^2/((1-\rho)(1-\rho^2))$, i.e., it is $1/((1-\rho)(1-\rho^2)) \sim \tau_{\mathsf{mix}}^2$ factor worse than the obvious lower bound of $O(d^2)$ to store $A$. We leave further investigation into space complexity optimization or tightening the lower bound for future work. Also, note that $u \leq B/10$, so $\mathsf{SGD} - \mathsf{RER}$ wastes only about 10% of the samples. Finally, the algorithm requires a reasonable upper bound on $\rho$ to set up various hyperparameters like $R, u, B$. However, it is not clear how to estimate such an upper bound only using the data, and seems like an interesting open question.

See Section 6 for an explanation of the elements involved in the analysis of the algorithm and to Section C.1 for a detailed overview of the proof.

## 5 Exponential Lower Bounds for Non-Expansive Link Functions

The previous results showed that we can efficiently recover the matrix $A^*$ given that the link function is uniformly expansive. We now consider non-expansive functions and show that parameter recovery is hard in this case. In particular, we show that even for the case of $\phi = \mathsf{ReLU}$, the noise being $\mathcal{N}(0, I)$, and $\|A^*\| \leq \frac{1}{2}$, the error has an information theoretic lower bound which is exponential in the dimension. We note that this is consistent with Theorem 3 in [32] which too has an exponential dependence on the dimension (since the matrix $K \succeq I$).

Before stating the results, we introduce some notation. Consider any algorithm $\mathcal{A}$, with accepts input $(X_0, \ldots, X_T)$ and outputs an estimate $\hat{A} \in \mathbb{R}^{d \times d}$. For simplicity of calculation, we will assume that $X_0 = 0$ and $X_{t+1} = \mathsf{ReLU}(A^* X_t) + \eta_t$. Since the mixing time is $O(1)$, similar results should hold for stationary sequences. We define the loss $\mathcal{L}(\mathcal{A}, T, A^*) = \mathbb{E}\|\hat{A} - A^*\|_F^2$, where the expectation is over the randomness in the data and the algorithm. By $\Theta(\frac{1}{2})$, we denote all the the elements of $B \in \mathbb{R}^{d \times d}$ such that $\|B\| \leq \frac{1}{2}$. The minimax loss is defined as:

$$\mathcal{L}(\Theta(\tfrac{1}{2}), T) := \inf_{\mathcal{A}} \sup_{A^* \in \Theta(\frac{1}{2})} \mathcal{L}(\mathcal{A}, T, A^*).$$

**Theorem 4** (ReLU Lower Bound). *For universal constants $c_0, c_1 > 0$, we have:*

$$\mathcal{L}(\Theta(\tfrac{1}{2}), T) \geq c_0 \min\left(1, \frac{\exp(c_1 d)}{T}\right).$$

We prove the theorem above using the two point method. We find a family of $A^*$ in $\Theta(\frac{1}{2}, T)$ such that $\langle X_t, e_d \rangle = \langle \eta_{t-1}, e_d \rangle$ with probability at-least $1 - \exp(-\Omega(d))$. Therefore, with a large probability, we only observe noise in the last co-ordinate and hence do not obtain any information regarding the last row of $A^*$ (i.e, $a_d^*$). We refer to Section G for a full proof.

## 6 Proof Sketch

**Quasi Newton Method.** Let $a_i^*$ be the $i$-th row of $A^*$ and $a_i(l)$ be the $i$-th row of $A_l$ both in column vector form. The proofs of Theorems 1 and 2 follow once we consider the lyapunov function $\Delta_{l,i} = \|\hat{G}^{1/2}(a_i(l) - a_i^*)\|$ and show that

$$\Delta_{l+1,i} \leq (1 - 2\gamma\zeta)\Delta_l + \gamma\|\hat{G}^{-1/2}\hat{N}_i\| \tag{6}$$

Where $\hat{N}_i := \frac{1}{T}\sum_{t=0}^{T-1}\langle e_i, \eta_t \rangle X_t$. In the case of Theorem 1, we use the sub-Gaussianity of the noise sequence and a martingale argument to obtain a high probability upper bound on $\sum_{i=1}^{d}\|\hat{G}^{-1/2}\hat{N}_i\|^2$

(see Lemma 1). In the heavy tailed case considered in Theorem 2, we use mixing time arguments along with Payley-Zygmund inequality to show the high probability lower isometry $\hat{G} \succeq c_0 G$ for some universal constant $c_0$ (see Lemma 2). Using this lower isometry, we can replace $\hat{G}$ in Equation (6) with $G$. The upper bounds follow once we note that $\mathbb{E}\hat{N}_i^\top G^{-1}\hat{N}_i = \frac{\sigma^2 d}{T}$.

**SGD-RER.** Due to the observations made in Section 4, we can split the analysis into the following parts, which are explained in detail below.

1. Analyze the reverse order SGD *within* the buffers.
2. Treat successive buffer as independent samples.
3. Give a bias-variance decomposition similar to the case of linear regression.
4. Use algorithmic stability to control 'spurious' coupling introduced by non-linearity in the bias-variance decomposition.

**Coupled Process.** We deal with the dependence *between* buffers using a fictitious coupled process, constructed just for the sake of analysis (see Definition 2). Leveraging the gap $u$, this process $(\tilde{X}_\tau)$ is constructed such that $\tilde{X}_\tau \approx X_\tau$ with high probability and the 'coupled buffers' containing data $\tilde{X}$ instead of $X$ are *exactly* independent. Since $\tilde{X}_\tau \approx X_\tau$, the output of SGD $-$ RER run with the fictitious coupled process should be close output of SGD $-$ RER run with the actual data points. We then use the strategy outlined above to analyze SGD $-$ RER with the coupled process. In the analysis given for SGD $-$ RER, all the quantities with $\tilde{\cdot}$ involve the coupled process $\tilde{X}$ instead of the real process $X$.

**Non-Linear Bias Variance Decomposition.** We use the mean value theorem to linearize the non-linear problem. This works effectively when the step size $\gamma$ is a vanishing function of the horizon $T$. Observe that the a single SGD/ SGD $-$ RER step for a single row can be written as:

$$a_i' - a_i^* = a_i - a_i^* - 2\gamma(\phi(\langle a_i, X_\tau \rangle) - \phi(\langle a_i^*, X_\tau \rangle))X_\tau + 2\gamma\langle \eta_\tau, e_i \rangle X_\tau$$
$$= \left(I - 2\gamma\phi'(\beta_\tau)X_\tau X_\tau^\top\right)(a_i - a_i^*) + 2\gamma\langle \eta_\tau, e_i \rangle X_\tau \tag{7}$$

In the second step, we have used the mean value theorem. Equation (7) can be interpreted as follows: the matrix $\left(I - 2\gamma\phi'(\beta_\tau)X_\tau X_\tau^\top\right)$ 'contracts' the distance between $a_i$ and $a_i^*$ whereas the noise $2\gamma\langle \eta_\tau, e_i \rangle X_\tau$ is due to the inherent uncertainty. This gives us a bias-variance decomposition similar to the case of SGD with linear regression. We refer to Section C.5 for details on unrolling the recursion in Equation (7) to obtain the exact bias-variance decomposition.

**Algorithmic Stability:** Unfortunately, non-linearities result in a 'coupling' between the contraction matrices through the iterates via the first derivative $\phi'(\beta_\tau)$ due to reverse order traversal. This is an important issue since unrolling the recursion in (7), we encounter terms such as $\langle \eta_\tau, e_i \rangle (I - 2\gamma\phi'(\beta_{\tau-1})X_{\tau-1}X_{\tau-1}^\top)X_\tau$, which have zero mean in the linear case. However, in the non-linear case, $\beta_{\tau-1}$ depends on $\eta_\tau$ due to reverse order traversal. We show that such dependencies are 'weak' using the idea of algorithmic stability ([45, 46]). In particular, we establish that the output of the algorithm is not affected too much if we re-sample the *entire* data trajectory by independently re-sampling a single noise co-ordinate ($\eta_\tau$ becomes $\eta_\tau'$ and $\beta_{\tau-1}$ becomes $\beta_{\tau-1}'$) when the step size $\gamma$ is small enough (in other words, the output is stable under small perturbations). Via second derivative arguments, we show that $\beta_{\tau-1} \approx \beta_{\tau-1}'$.

Now observe that resampling noise $\eta_\tau$ does not affect the past value of data i.e, $X_\tau, X_{\tau-1}$ and is independent of $\beta_{\tau-1}'$ by construction. Therefore

$$0 = \mathbb{E}\langle \eta_\tau, e_i \rangle \left(I - 2\gamma\phi'(\beta_{\tau-1}')X_{\tau-1}X_{\tau-1}^\top\right) X_\tau \approx \mathbb{E}\langle \eta_\tau, e_i \rangle \left(I - 2\gamma\phi'(\beta_{\tau-1})X_{\tau-1}X_{\tau-1}^\top\right) X_\tau$$

Such a resampling procedure is also explored in [47] for the analysis of SGD with random reshuffling.

We put together all the ingredients above in order to prove the error bounds given in Theorem 3.

## 7 Experiments

In this section, we compare performance of our methods SGD $-$ RER and Quasi Newton method on synthetic data against the performance of standard baselines SGD (called 'Forward SGD' here),

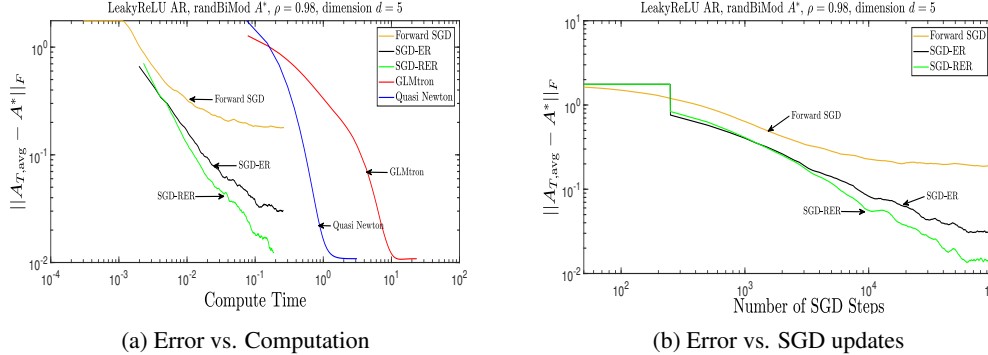

| (a) Error vs. Computation | (b) Error vs. SGD updates |

Figure 2: Performance of various algorithms for the case of $\phi = \textsf{LeakyReLU}$

GLMtron, along with the $\textsf{SGD} - \textsf{ER}$ method that applies standard experience replay technique i.e, the points from a buffer are sampled *randomly* instead of the reverse order. Since GLMtron and Quasi Newton Method are offline and $\textsf{SGD} - \textsf{RER}$, $\textsf{SGD}$ and $\textsf{SGD} - \textsf{ER}$ are streaming, we compare the algorithms by plotting parameter error measured by the Frobenius norm with respect to the compute time. We also compare error vs. number of iterations for the streaming algorithms. We show the results of additional experiments by considering various buffer sizes and heavy tailed noise in Section A.

**Synthetic data**: We sample data from $\textsf{NLDS}(A^*, \mu, \phi)$ where $\mu \sim \mathcal{N}(0, \sigma^2 I)$ and $A^* \in \mathbb{R}^{d \times d}$ is generated from the "RandBiMod" distribution. That is, $A^* = U \Lambda U^\top$ with random orthogonal $U$, and $\Lambda$ is diagonal with $\lceil d/2 \rceil$ entries on diagonal being $\rho$ and the remaining diagonal entries are set to $\rho/3$. $\phi$ is the leaky ReLU function given by $\phi(x) = 0.5x \mathbb{1}(x < 0) + x \mathbb{1}(x \geq 0)$. We set $d = 5$, $\rho = 0.98$ and $\sigma^2 = 1$. We set a horizon of $T = 10^5$.

**Algorithm Parameters** We set $B = 240$ and $u = 10$ for the buffer size and gap size respectively for both $\textsf{SGD} - \textsf{RER}$ and $\textsf{SGD} - \textsf{ER}$ and use full averaging (i.e, $\theta = 0$ in Algorithm 2 ). We set the step size $\gamma = \frac{5 \log T}{T}$ for $\textsf{SGD}$, $\textsf{SGD} - \textsf{RER}$, and $\textsf{SGD} - \textsf{ER}$ and $\gamma_{\textsf{newton}} = 0.2$ and $\gamma_{\textsf{GLMtron}} = 0.017$.

From Figure 2 observe that $\textsf{SGD} - \textsf{ER}$ and $\textsf{SGD}$ obtain sub-optimal results compared $\textsf{SGD} - \textsf{RER}$, Quasi Newton Method and GLMtron. After a single pass, the performance of $\textsf{SGD} - \textsf{RER}$ almost matches that of the offline algorithms. The step sizes for GLMtron have to be chosen to be small in-order to ensure that the algorithm does not diverge as noted in Section 3, which slows down its convergence time compared to the Quasi Newton method. We set the step size to be as large as possible without obtaining divergence to infinity.

# 8 Conclusion

In this work, we studied the problem of learning non-linear dynamical systems of the form (1) from a single trajectory and analyzed offline and online algorithms to obtain near-optimal error guarantees. In particular we showed that mixing time based arguments are not necessary for learning certain classes of non-linear dynamical systems. Even though we show that one cannot hope for efficient parameter recovery with non-expansive link functions like ReLU, we do not deal with the problem of minimizing the 'prediction' error - where we output a good predictor based on samples, without parameter recovery. We believe that this problem would require significantly different set of techniques than the ones established in this work and hope to investigate this in future work. Presently our work only deals with specific kinds of Markovian time evolution. It would also be interesting to understand in general, the kind of structures which allow for learning without mixing based arguments. Another related direction is to design simple and efficient algorithms like $\textsf{SGD} - \textsf{RER}$ for learning with various models of dependent data by unraveling the dependency structure present in the data.

## Acknowledgments and Disclosure of Funding

D.N. was supported in part by NSF grant DMS-2022448.
S.S.K was supported in part by Teaching Assistantship (TA) from EECS, MIT.

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
