

(a) Varying the Buffer Sizes          (b) GLMtron step sizes

Figure 3: Varying Parameters in the Algorithms

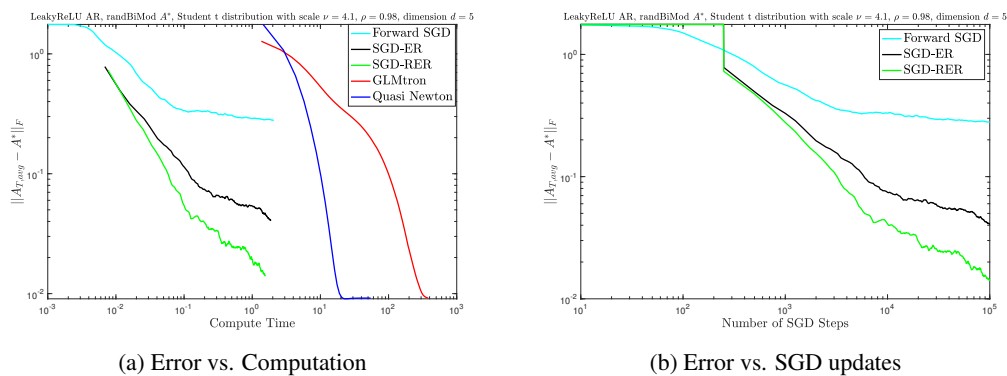

(a) Error vs. Computation          (b) Error vs. SGD updates

Figure 4: Performance of various algorithms with heavy tailed noise

## A   Additional Experiments

Based on the feedback from the reviewers, we perform the following additional experiments which explore the robustness of the choice of buffer size in SGD − RER, choice of step sizes for GLMtron and the behavior of the said algorithms with heavy tailed noise with a similar setup as in Section 7.

We first perform an experimental study about the robustness of SGD − RER to the choice of buffer size in Figure 3a. Notice that the performance remains the same for a large range of buffer sizes ( 100 from to 2000). However the performance degrades when the buffer size is too large ( ≈ 10000). We believe this is the case since the number of buffers decreases as the buffer size increases and the output is averaged over too few number of iterates (In the case of B = 10000, the final output is just an average of 10 iterates).

Next, we consider the range of step sizes which allow GLMtron to converge in Figure 3b in order to supplement the discussion in Sections 7 and 3 regarding GLMtron requiring smaller step-sizes. In smooth convex optimization, it is typically the case that the iterates diverge to infinity if the step size is chosen to be too large. Theoretically, this largest step-size is $\frac{2}{L}$ where $L$ is the largest eigenvalue of the Hessian. In the case of GLMtron, it was experimentally observed that if the step size was chosen to be about 1.5 times the step size reported in Section 7, the iterates diverged. Quasi Newton method essentially normalizes the gradient with the inverse of the Hessian (or rather an approximation of the Hessian) in order to let it converge faster with large step sizes.

In Figure 4, we consider the same system as in Section 7 but with heavy tailed noise given by the student t distribution (scale $\nu = 4.1$) so that the 4-th moment exists but higher moments do not. The typical behavior of Forward SGD, SGD-ER, SGD-RER and Quasi Newton methods seems to be similar to that observed in the Sub-Gaussian noise case. However, GLMtron requires much smaller step sizes to ensure convergence and hence it takes much longer. We believe that the reason for this is related to the explanation given above for GLMtron step sizes. The largest eigenvalue of the Hessian depends on the quantity $X_t X_t^\top$, which can be much larger than in the sub-gaussian case and hence we need to pick much smaller step sizes. However, further research is needed to confirm this

phenomenon. We also note that we did not provide theoretical guarantees for SGD-RER in the heavy tailed noise case. But it is still seen to typically perform very well.

# B   Analysis of the Quasi Newton Method

In this Section, we give the proofs of Theorems 1 and 2. Let $e_1, \ldots, e_d$ be the standard basis vectors for $\mathbb{R}^d$. We will analyze the Quasi Newton method row by row.

**Definiton 1.** *Given a matrix $A = [a_1, a_2, \cdots, a_d]^\top$, let $\mathcal{R}(A) = \{a_1, \cdots, a_d\}$ denote the set of vectors that are (transposes of) rows of the matrix $A$. We use $a^\top$ to represent a generic row of $A$.*

Follow Defintion 1, we will consider the estimation of the $i$-th row $a_i^*$. Consider the gradient $\nabla \mathcal{L}_{\mathsf{prox}}^{(i)} : \mathbb{R}^d \to \mathbb{R}^d$ given by:

$$\nabla \mathcal{L}_{\mathsf{prox}}^{(i)}(a) := \frac{1}{T} \sum_{t=0}^{T-1} \left( \phi(\langle a, X_t \rangle) - \langle e_i, X_{t+1} \rangle \right) X_t .$$

We can write

$$\nabla \mathcal{L}_{\mathsf{prox}}^{(i)}(a) = \frac{1}{T} \sum_{t=0}^{T-1} \left( \phi(\langle a, X_t \rangle) - \phi(\langle a_i^*, X_t \rangle) \right) X_t - \langle \eta_t, e_i \rangle X_t$$

$$= \frac{1}{T} \sum_{t=0}^{T-1} \phi'(\beta_t) \langle a - a_i^*, X_t \rangle X_t - \langle \eta_t, e_i \rangle X_t$$

$$= \hat{K}_{a,i}(a - a_i^*) - \hat{N}_i \qquad (8)$$

Where $\beta_t$ exist because of the mean value theorem. We can make sense of $\beta_t$ even when $\phi$ is only weakly differentiable and check that the proof below still follows. Here, $\hat{K}_{a,i} := \frac{1}{T} \sum_{t=0}^{T-1} \phi'(\beta_t) X_t X_t^\top$ and $\hat{N}_i := \frac{1}{T} \sum_{t=0}^{T-1} \langle \eta_t, e_i \rangle X_t$. In the first step we have used the dynamics in Equation (1) to write down $X_{t+1}$ in terms of $X_t$ and $\eta_t$.

We now define $\hat{G} \in \mathbb{R}^{d \times d}$ by $\hat{G} := \frac{1}{T} \sum_{t=0}^{T-1} X_t X_t^\top$. From the fact that $\zeta \le \phi'(\beta_t) \le 1$, we note that for every $a \in \mathbb{R}^d$:

$$\hat{G} \succeq \hat{K}_{a,i} \succeq \zeta \hat{G} \qquad (9)$$

Now consider the Quasi Newton Step given in Algorithm 1.

$$a_i(l+1) = a_i(l) - 2\gamma \hat{G}^{-1} \nabla \mathcal{L}_{\mathsf{prox}}^{(i)}(a_i(l))$$

Denoting $\hat{K}_{a_i(l),i}$ by $\hat{K}_{l,i}$ we use Equation (8) to conclude:

$$a_i(l+1) - a_i^* = a_i(l) - a_i^* - 2\gamma \hat{G}^{-1} \hat{K}_{l,i}(a_i(l+1) - a_i^*) + 2\gamma \hat{G}^{-1} \hat{N}_i$$

$$\implies \sqrt{\hat{G}}(a_i(l+1) - a_i^*) = (I - 2\gamma \hat{G}^{-1/2} \hat{K}_{l,i} \hat{G}^{-1/2}) \sqrt{\hat{G}}(a_i(l) - a_i^*) + 2\gamma \hat{G}^{-1/2} \hat{N}_i \quad (10)$$

Picking $\gamma < \frac{1}{2}$, we conclude from Equation (9) that:

$$(1 - 2\gamma)I \preceq I - 2\gamma \hat{G}^{-1/2} \hat{K}_{l,i} \hat{G}^{-1/2} \preceq (1 - 2\gamma\zeta)I$$

We use the equation above in Equation (10) along with triangle inequality to conclude:

$$\|\sqrt{\hat{G}}(a_i(l+1) - a_i^*)\| \le (1 - 2\gamma\zeta)\|\sqrt{\hat{G}}(a_i(l) - a_i^*)\| + 2\gamma\|\hat{G}^{-1/2} \hat{N}_i\|$$

Unrolling the recursion above, we obtain that:

$$\|\sqrt{\hat{G}}(a_i(m) - a_i^*)\| \le (1 - 2\gamma\zeta)^m \|\sqrt{\hat{G}}(a_i(0) - a_i^*)\| + \sum_{l=0}^{m-1} 2\gamma(1 - 2\gamma\zeta)^l \|\hat{G}^{-1/2} \hat{N}_i\|$$

$$\le (1 - 2\gamma\zeta)^m \|\sqrt{\hat{G}}(a_i(0) - a_i^*)\| + \frac{1}{\zeta}\|\hat{G}^{-1/2} \hat{N}_i\|$$

Letting $A_m$ be the matrix with rows $a_i(m)$, we conclude:

$$\|A_m - A^*\|_{\mathsf{F}}^2 \le 2\frac{\lambda_{\max}(\hat{G})}{\lambda_{\min}(\hat{G})}(1 - 2\gamma\zeta)^{2m}\|A_0 - A^*\|_{\mathsf{F}}^2 + \frac{2}{\zeta^2\lambda_{\min}(\hat{G})}\sum_{i=1}^d \|\hat{G}^{-1/2}\hat{N}_i\|^2 \qquad (11)$$

Proof of Theorems 1 and 2 follow once we provide high probability bounds for various terms in Equation (11). We will first define some notation. Let $S(\rho, T) := \sum_{t=0}^T \rho^{T-t}$. For $R, \kappa > 0$, we define the following events

1. $\mathcal{D}_T(R) := \{\sup_{0 \le t \le T} \|X_t\|^2 \le R\}$

2. $\mathcal{E}_T(\kappa) := \{\hat{G} \succeq \frac{\sigma^2 I}{\kappa}\}$

3. $\mathcal{D}_T(R, \kappa) := \mathcal{D}_T(R) \cap \mathcal{E}_T(\kappa)$

**Lemma 1.** *Under the Assumptions of Theorem 1, suppose $\delta \in (0, 1/2)$ and take $R = C_\rho^2 C_\eta(S(\rho, T))^2 d\sigma^2 \log(\frac{2T}{\delta})$, $\kappa = 2$, and $T \ge \bar{C}_3 \left(d\log\left(\frac{R}{\sigma^2}\right) + \log\frac{1}{\delta}\right)$*

$$\mathbb{P}\left(\sum_{i=1}^d \|\hat{G}^{-\frac{1}{2}}\hat{N}_i\|^2 \le \frac{\bar{C}\sigma^2}{T}\left[d^2\log\left(1 + \frac{R}{\sigma^2}\right) + d\log(\frac{1}{\delta})\right] \cap \mathcal{D}_T(R, \kappa)\right) \ge 1 - 2d\delta$$

*Where $\bar{C}, \bar{C}_3$ are constants depending only on $C_\eta$.*

We refer to Section F for the proof.

**Lemma 2.** *Suppose the Assumptions of Theorem 2 hold. There exist universal constants $C, C_1, c_0 > 0$ such that whenever $\delta \in (0, \frac{1}{2})$, $R := \frac{4TdC_\rho^2\sigma^2}{(1-\rho)^2\delta}$ and $T \ge Cd\log(\frac{1}{\delta})\log(\frac{R}{\sigma^2})\max\left(\frac{4C_\rho^6 M_4}{(1-\rho)^4(1-\rho^2)\sigma^4}, \frac{\log\left(\frac{RC_1C_\rho}{\sigma^2}\right)}{\log\left(\frac{1}{\rho}\right)}\right)$, we have with probability at-least $1 - \delta$:*

1.
$$\hat{G} \succeq c_0 G.$$

2.
$$\lambda_{\max}\left(\hat{G}\right) \le R$$

*Where $G := \mathbb{E}X_t X_t^\top$*

We give the proof of this lemma in Section H.

### B.1 Proof of Theorem 1

*Proof.* Note that $R$ in Lemma 1 is the same as $R^*$ in the statement of Theorem 1. We combine the result of Lemma 1 with the Equation (11). Under the event $\mathcal{D}_T(R^*, \kappa)$, we have $\frac{\lambda_{\max}(\hat{G})}{\lambda_{\min}(\hat{G})} \le \frac{2R}{\sigma^2}$ almost surely. Hence the result follows.

$\square$

### B.2 Proof of Theorem 2

*Proof.* We again begin with Equation (11).

$$\|A_m - A^*\|_{\mathsf{F}}^2 \le 2\frac{\lambda_{\max}(\hat{G})}{\lambda_{\min}(\hat{G})}(1 - 2\gamma\zeta)^{2m}\|A_0 - A^*\|_{\mathsf{F}}^2 + \frac{2}{\zeta^2\lambda_{\min}(\hat{G})}\sum_{i=1}^d \|\hat{G}^{-1/2}\hat{N}_i\|^2$$

Let the event described in Lemma 2 be $\mathcal{W}$. Under this event, $\lambda_{\min}\left(\hat{G}\right) \geq c_0 \lambda_{\min}(G)$, $\lambda_{\max}(G) \leq R$ and $\hat{G}^{-1} \preceq \frac{1}{c_0} G^{-1}$. Therefore, we conclude:

$$\|A_m - A^*\|_{\mathsf{F}}^2 \leq 2 \frac{R}{c_0 \lambda_{\min}(G)} (1 - 2\gamma\zeta)^{2m} \|A_0 - A^*\|_{\mathsf{F}}^2 + \frac{2}{\zeta^2 c_0^2 \lambda_{\min}(G)} \sum_{i=1}^{d} \hat{N}_i^\top G^{-1} \hat{N}_i$$

$\square$

Now, consider $\mathbb{E}\|A_m - A^*\|_{\mathsf{R}}^2 \mathbb{1}(\mathcal{W})$. To conclude the result of the Theorem, we will show that $\mathbb{E}\hat{N}_i^\top G^{-1} \hat{N}_i \mathbb{1}(\mathcal{W}) \leq \frac{d\sigma^2}{T}$. Indeed:

$$\mathbb{E}\hat{N}_i^\top G^{-1} \hat{N}_i \mathbb{1}(\mathcal{W}) \leq \mathbb{E}\hat{N}_i^\top G^{-1} \hat{N}_i$$

$$= \frac{1}{T^2} \sum_{t,s=0}^{T-1} \mathbb{E}\langle \eta_t, e_i \rangle \langle \eta_s, e_i \rangle X_t^\top G^{-1} X_s \tag{12}$$

Observe that whenever $t > s$, $\eta_t$ has mean zero and is independent of $X_t, X_s$ and $\eta_s$. Therefore, $\mathbb{E}\langle \eta_t, e_i \rangle \langle \eta_s, e_i \rangle X_t^\top G^{-1} X_s = 0$. When $t = s$, we conclude using the fact that $\mathbb{E}X_t X_t^\top = G$ that $\mathbb{E}\langle \eta_t, e_i \rangle^2 X_t^\top G^{-1} X_t = d\sigma^2$. Using the calculations above and Equation (12), we conclude that:

$$\mathbb{E}\|A_m - A^*\|_{\mathsf{R}}^2 \mathbb{1}(\mathcal{W}) \leq \frac{2d^2\sigma^2}{c_0^2 \zeta^2 T \lambda_{\min}(G)} + 2 \frac{R}{c_0 \lambda_{\min}(G)} (1 - 2\gamma\zeta)^{2m} \|A_0 - A^*\|_{\mathsf{F}}^2$$

## C    Analysis of SGD-RER

In this section we consider the following $(X_0, X_1, \ldots, X_T)$ to be a stationary sequence from $\mathsf{NLDS}(A^*, \mu, \phi)$. We make Assumptions 1, 2, 3 and 5. We aim to analyze Algorithm 2 and then prove Theorem 3.

The data is divided into buffers of size $B$ and the buffers have a gap of size $u$ in between them. Let $S = B + u$. The algorithm runs SGD with respect to the proxy loss $\mathcal{L}_{\mathsf{prox}}$ in the order described in Section 6. Formally, let $X_j^t \equiv X_{tS+j}$ denote the the $j$-th sample in buffer $t$. We denote, for $0 \leq i \leq B - 1$, $X_{-i}^t \equiv X_{(S-1)-i}^t$ i.e., the $i$-th processed sample in buffer $t$. We use similar notation for noise samples i.e., $\eta_j^t \equiv \eta_{tS+j}$ and $\eta_{-j}^t \equiv \eta_{(S-1)-j}^t$.

The algorithm iterates are denoted by the sequence $(A_i^t : 0 \leq t \leq N - 1, 0 \leq i \leq B - 1)$ where $A_i^t$ denotes the iterate obtained after processing $i$-th (reversed) sample in buffer $t$ and $N = T/S$ is the total number of buffers. Note that we enumerate buffers from $0, 1, \cdots N - 1$. Formally

$$A_{i+1}^{t-1} = A_i^{t-1} - 2\gamma \left( \phi(A_i^{t-1} X_{-i}^{t-1}) - X_{-(i-1)}^{t-1} \right) X_{-i}^{t-1,\top} \tag{13}$$

for $1 \leq t \leq N, 0 \leq i \leq B - 1$ and we set $A_0^t = A_B^{t-1}$ with $A_0^0 = A_0$.

The algorithm outputs the tail-averaged iterate at the end of each buffer $t$: $\hat{A}_{t_0,t} = \frac{1}{t-t_0} \sum_{\tau=t_0+1}^{t} A_B^{\tau-1}$ where $1 \leq t \leq N$ and $0 \leq t_0 \leq t - 1$.

### C.1    Proof Strategy

The proof of Theorem 3 involves many intricate steps. Therefore, we give a detailed overview about the proof below.

1. In Section C.2 we first construct a fictitious coupled process $\tilde{X}_\tau$ such that for every data point within a buffer $t$, $\|\tilde{X}_\tau - X_\tau\| \lesssim \frac{1}{T^\alpha}$ for some fixed $\alpha > 0$ chosen arbitrarily beforehand. We then show that the iterates $\tilde{A}_i^t$ which are generated with $\mathsf{SGD} - \mathsf{RER}$ is run with the coupled process $\tilde{X}_\tau$ is very close to the actual iterate $A_i^t$. The coupled process has the advantage that the data in the successive buffers are independent. We then only deal with the coupled iterates $\tilde{A}_i^t$ and appeal to Lemma 4 to obtain bounds for $A_i^t$.

2. In Section C.5, we give the bias variance decomposition as is standard in the linear regression literature. We extend it to the non-linear case using the mean value theorem and treat the buffers as independent data samples. Here, the matrices $\tilde{H}^s_{0,B-1}$ defined on the data in buffer $s$ 'contracts' the norm of $A^s_i - A^*$ giving the 'bias term' whereas the noise $\eta^s_i$ presents the 'variance' term which is due to the inherent uncertainty in the estimation problem.

3. We refer to Section E where we develop the contraction properties of the matrices $\prod^t_{s=0} \tilde{H}^s_{0,B-1}$ where we show that $\|\prod^t_{s=0} \tilde{H}^s_{0,B-1}\| \lesssim (1 - \zeta\gamma B\lambda_{\min}(G))^t$ in Theorem 10 after developing some probabilistic results regarding $\mathsf{NLDS}(A^*, \mu, \phi)$. This allows us show exponential decay of the bias.

4. We then turn to the squared variance term in Section C.6. We decompose it into 'diagonal terms' with non-zero expectation and 'cross terms' with a vanishing expectation. Bounding the diagonal term is straight forward using standard recursive arguments and we give the bound in Claim 1.

5. The 'cross terms' which vanish in expectation in the linear case, do not because of the coupling introduced by the non-linearities through the iterates (see Section 6 for a short description). However, we establish 'algorithmic stability' in Section C.7 where we show that the iterates depend only weakly on each of the noise vectors and hence the cross terms have expectation very close to zero. More specifically, we use the novel idea of re-sampling the whole trajectory $(\tilde{X}_\tau)$ by re-sampling one noise vector only and show that the iterates of $\mathsf{SGD} - \mathsf{RER}$ are not affected much.

6. We use the 'algorithmic stability' bounds to bound the cross terms in Sections C.8. We then combine the bounds to obtain the bound on the 'variance term'

7. Finally, we analyze the tail averaged output in Sections C.10, C.11 and C.12 and then combine these ingredients to prove Theorem 3.

## C.2 Basic Notations and Coupled Process

**Definiton 2** (Coupled process)**.** *Given the co-variates $\{X_\tau : \tau = 0, 1, \cdots T\}$ and noise $\{\eta_\tau : \tau = 1, 2, \cdots, T\}$, we define $\{\tilde{X}_\tau : \tau = 0, 1, \cdots, T\}$ as follows:*

1. *For each buffer $t$ generate, independently of everything else, $\tilde{X}^t_0 \sim \pi$, the stationary distribution of the $\mathsf{NLDS}(A^*, \mu, \phi)$ model.*

2. *Then, each buffer has the same recursion as eq (1):*
$$\tilde{X}^t_{i+1} = \phi(A^*\tilde{X}^t_i) + \eta^t_i, \ i = 0, 1, \cdots S - 1, \tag{14}$$
*where the noise vectors as same as in the actual process $\{X_\tau\}$.*

**Lemma 3** (Coupling Lemma)**.** *Under Assumption 4, for any buffer $t$, we have $\left\| X^t_i - \tilde{X}^t_i \right\| \leq C_\rho\rho^i \left\| X^t_0 - \tilde{X}^t_0 \right\|$, a.s. Hence*
$$\left\| X^t_i X^{t,\top}_i - \tilde{X}^t_i \tilde{X}^{t,\top}_i \right\| \leq 2(\sup_{\tau \leq T} \|X_\tau\|) \left\| X^t_i - \tilde{X}^t_i \right\| \leq 4 \sup_{\tau \leq T} \|X_\tau\|^2 C_\rho\rho^i \tag{15}$$

With the above notation, we can write (13) in terms of a generic row (say row $r$) $a^{t-1,\top}_{i+1}$ of $A^{t-1}_{i+1}$ as follows. Let $\varepsilon^{t-1}_{-i}$ denote the element of $\eta^{t-1}_{-i}$ in row $r$. Similarly let $a^{*,\top} \equiv (a^*)^\top$ denote the row $r$ of $A^*$. Then

$$a^{t-1,\top}_{i+1} = a^{t-1,\top}_i - 2\gamma \left( \phi(X^{t-1,\top}_{-i} a^{t-1}_i) - \phi(X^{t-1,\top}_{-i} a^*) \right) X^{t-1,\top}_{-i} + 2\gamma\varepsilon^{t-1}_{-i} X^{t-1,\top}_{-i} \tag{16}$$

Now, by the mean value theorem we can write
$$\phi(X^{t-1,\top}_{-i} a^{t-1}_i) - \phi(X^{t-1,\top}_{-i} a^*) = \phi'(\xi^{t-1}_{-i})(a^{t-1}_{-i} - a^*)^\top X^{t-1}_{-i} \tag{17}$$
where $\xi^{t-1}_{-i}$ lies between $X^{t-1,\top}_{-i} a^{t-1}_i$ and $X^{t-1,\top}_{-i} a^*$. Hence we obtain

$$(a^{t-1}_{i+1} - a^*)^\top = (a^{t-1}_i - a^*)^\top (I - 2\gamma\phi'(\xi^{t-1}_{-i})X^{t-1}_{-i} X^{t-1,\top}_{-i}) + 2\gamma\varepsilon^{t-1}_{-i} X^{t-1,\top}_{-i} \tag{18}$$

**Definiton 3** (Coupled SGD Iteration). *Consider the process described in Defintion 2. We define* SGD − RER *iterates run with the coupled process $(\tilde{X}_i^t)$ as follows:*

$$\tilde{A}_0^0 = A_0^0$$

$$\tilde{A}_{i+1}^{t-1} = \tilde{A}_i^{t-1} - 2\gamma \left( \phi(\tilde{A}_i^{t-1}\tilde{X}_{-i}^{t-1}) - \tilde{X}_{-(i-1)}^{t-1} \right) \tilde{X}_{-i}^{t-1,\top} \tag{19}$$

Using Lemma 3, we can show that $\tilde{A}_i^t \approx A_i^t$. Note that successive buffers for the iterates $\tilde{A}_i^t$ are actually independent. We state the following lemma which shows that we can indeed just analyze $\tilde{A}_i^t$ and then from this obtain error bounds for $A_i^t$. We refer to Section H for the the proof.

**Lemma 4.** *Suppose $\gamma < \frac{1}{2R_{\max}}$. we have for every $t \in [N]$ and $i \in [B]$.*

$$\|a_i^t - \tilde{a}_i^t\| \leq (16\gamma^2 R_{\max}^2 T^2 + 8\gamma R_{\max}T)\rho^u$$

We note that we can just analyze the iterates $\tilde{A}_i^t$ and then use Lemma 4 to infer error bounds for $A_i^t$. Henceforth, we will only consider $\tilde{A}_i^t$.

Before proceeding, we will set up some notation.

### C.3   Notations and Events

We define the following notations. Let $R > 0$ to be decided later.

$$X_{-i}^t = X_{(S-1)-i}^t, \ 0 \leq i \leq S-1, \qquad \phi'(\tilde{\xi}_{-i}^t) = \frac{\phi(\tilde{a}_{-i}^{t,\top}\tilde{X}_{-i}^t) - \phi(a^{*,\top}\tilde{X}_{-i}^t)}{\left(\tilde{a}_{-i}^t - a^*\right)^\top \tilde{X}_{-i}^t}$$

$$\tilde{P}_{-i}^t = \left( I - 2\gamma\phi'\left(\tilde{\xi}_{-i}^t\right)\tilde{X}_{-i}^t \tilde{X}_{-i}^{t,\top} \right), \quad \tilde{H}_{i,j}^t = \begin{cases} \prod_{s=i}^j \tilde{P}_{-s}^t & i \leq j \\ I & i > j \end{cases},$$

$$\hat{\gamma} = 4\gamma(1 - \gamma R), \quad \mathcal{C}_{-j}^t = \left\{ \left\| X_{-j}^t \right\|^2 \leq R \right\}, \quad \tilde{\mathcal{C}}_{-j}^t = \left\{ \left\| \tilde{X}_{-j}^t \right\|^2 \leq R \right\},$$

$$\mathcal{D}_{-j}^t = \left\{ \left\| X_{-i}^t \right\|^2 \leq R : j \leq i \leq B-1 \right\} = \bigcap_{i=j}^{B-1} \mathcal{C}_{-i}^t,$$

$$\mathcal{D}^{s,t} = \begin{cases} \bigcap_{r=s}^t \mathcal{D}_{-0}^r & s \leq t \\ \Omega & s > t \end{cases}, \quad \tilde{\mathcal{D}}_{-j}^t = \left\{ \left\| \tilde{X}_{-i}^t \right\|^2 \leq R : j \leq i \leq B-1 \right\} = \bigcap_{i=j}^{B-1} \tilde{\mathcal{C}}_{-i}^t,$$

$$\tilde{\mathcal{D}}^{s,t} = \begin{cases} \bigcap_{r=s}^t \tilde{\mathcal{D}}_{-0}^r & s \leq t \\ \Omega & s > t \end{cases}, \quad \hat{\mathcal{D}}_{-j}^t = \mathcal{D}_{-j}^t \cap \tilde{\mathcal{D}}_{-j}^t, \quad \hat{\mathcal{D}}^{s,t} = \mathcal{D}^{s,t} \cap \tilde{\mathcal{D}}^{s,t}.$$

To execute algorithmic stability arguments, we will need to independently resample individual noise co-ordinates. To that end, define $(\bar{\eta}_\tau)_\tau$ drawn i.i.d from the noise distribution $\mu$ and independent of everything else defined so far. We denote their generic rows by $\bar{\varepsilon}$. We use the following events which correspond to a generic row

$$\mathcal{E}_{i,j}^t = \left\{ \|\varepsilon_{-k}^t\|^2 \leq \beta, \|\bar{\varepsilon}_{-k}^t\|^2 \leq \beta : i \leq k \leq j \right\}$$

### C.4   Setting the Parameter Values:

We make Assumptions 1, 2, 3 and 5 throughout. We set the parameters for SGD − RER as follows for the rest of the analysis. We note that some of these parameter values were set in Section 4.

1. $\alpha \geq 10$
2. $\beta = 4C_\eta \sigma^2(\alpha+2)\log 2T$.
3. $R \geq \frac{16(\alpha+2)dC_\eta\sigma^2 \log T}{1-\rho}$

4. $\delta = 1/(2T^{\alpha+1})$

5. $u \geq \frac{2\alpha \log T}{\log(\frac{1}{\rho})} = O(\tau_{\mathsf{mix}} \log T)$

6. $B \geq \max\left(\bar{C}_1 \frac{d}{(1-\rho)(1-\rho^2)}, 10u\right)$ where $\bar{C}_1$ depends only on $C_\eta$(see Theorem 10)

7. $\gamma \leq \min\left(\frac{\zeta}{4BR(1+\zeta)}, 1/2R\right)$ (see Theorem 10)

From Assumption 3 and Theorem 8, we conclude that for this choice of $R$ and $\beta$, we must have:

$$\mathbb{P}\left[\left(\hat{\mathcal{D}}^{0,N-1} \cap \cap_{r=0}^{N-1} \mathcal{E}_{0,B-1}^r\right)^C\right] \leq \frac{1}{2T^\alpha} \tag{20}$$

## C.5 Bias-variance decomposition

Using the above notation we can unroll the recursion in (18) as follows. We will only focus on the algorithmic iterated at the end of each buffer, i.e., we set $i = B - 1$ in (18).

$$(\tilde{a}_B^{t-1} - a^*)^\top = (a_0 - a^*)^\top \prod_{s=0}^{t-1} \tilde{H}_{0,B-1}^s + 2\gamma \sum_{r=1}^{t} \sum_{j=0}^{B-1} \varepsilon_{-j}^{t-r} \tilde{X}_{-j}^{t-r,\top} \tilde{H}_{j+1,B-1}^{t-r} \prod_{s=r-1}^{1} \tilde{H}_{0,B-1}^{t-s} \tag{21}$$

We call the above the *bias-variance* decomposition where

$$(\tilde{a}_B^{t-1,b} - a^*)^\top = (a_0 - a^*)^\top \prod_{s=0}^{t-1} \tilde{H}_{0,B-1}^s \tag{22}$$

is the bias, and

$$(\tilde{a}_B^{t-1,v})^\top = 2\gamma \sum_{r=1}^{t} \sum_{j=0}^{B-1} \varepsilon_{-j}^{t-r} \tilde{X}_{-j}^{t-r,\top} \tilde{H}_{j+1,B-1}^{t-r} \prod_{s=r-1}^{1} \tilde{H}_{0,B-1}^{t-s} \tag{23}$$

is the variance. We have the following simple lemma on bias-variance decomposition.

**Lemma 5.**

$$\left\|\tilde{a}_B^{t-1} - a^*\right\|^2 \preceq 2\left(\left\|\tilde{a}_B^{t-1,b} - a^*\right\|^2 + \left\|\tilde{a}_B^{t-1,v}\right\|^2\right) \tag{24}$$

## C.6 Variance of last iterate - Diagonal Terms

In this section our goal is to decompose $\left\|\tilde{a}_B^{t-1,v}\right\|^2$ into diagonal terms and cross terms. We will then proceed to bound the diagonal terms. First, we have a preliminary lemma, which can be shown via a simple recursion.

**Lemma 6.** *For $k \leq t$ define $S_k^t$ as*

$$S_k^t = \sum_{r=k}^{t} \sum_{j=0}^{B-1} \phi'(\tilde{\xi}_{-j}^{t-r}) \left(\prod_{s=1}^{r-1} \tilde{H}_{0,B-1}^{t-s,\top}\right) \tilde{H}_{j+1,B-1}^{t-r,\top} \tilde{X}_{-j}^{t-r} \tilde{X}_{-j}^{t-r,\top} \tilde{H}_{j+1,B-1}^{t-r} \left(\prod_{s=r-1}^{1} \tilde{H}_{0,B-1}^{t-s}\right) \tag{25}$$

*Then, on the event $\tilde{\mathcal{D}}^{0,t-1}$, we have*

$$S_1^t \preceq \frac{1}{\hat{\gamma}}\left(I - \left(\prod_{s=1}^{t} \tilde{H}_{0,B-1}^{t-s,\top}\right) \left(\prod_{s=t}^{1} \tilde{H}_{0,B-1}^{t-s}\right)\right) \tag{26}$$

*where $\hat{\gamma} = 4\gamma(1 - \gamma R)$*

*Proof.* The proof is similar to that of [19, Claim 1]. □

Next, we write $\left\|\tilde{a}_B^{t-1,v}\right\|^2$ as

$$\left\|\tilde{a}_B^{t-1,v}\right\|^2 = \sum_{r=1}^{t} \sum_{j=0}^{B-1} \mathrm{Dg}(t,r,j) + \sum_{r_1,r_2} \sum_{j_1,j_2} \mathrm{Cr}(t,r_1,r_2,j_1,j_2) \tag{27}$$

where the second sum is over $(r_1,j_1) \neq (r_2,j_2)$ and

$$\mathrm{Dg}(t,r,j) = 4\gamma^2 |\varepsilon_{-j}^{t-r}|^2 \cdot \tilde{X}_{-j}^{t-r,\top} \tilde{H}_{j+1,B-1}^{t-r} \left(\prod_{s=r-1}^{1} \tilde{H}_{0,B-1}^{t-s}\right) \left(\prod_{s=1}^{r-1} \tilde{H}_{0,B-1}^{t-s,\top}\right) \tilde{H}_{j+1,B-1}^{t-r,\top} \tilde{X}_{-j}^{t-r} \tag{28}$$

and

$$\mathrm{Cr}(t,r_1,r_2,j_1,j_2) = 4\gamma^2 \varepsilon_{-j_2}^{t-r_2} \tilde{X}_{-j_2}^{t-r_2,\top} \tilde{H}_{j_2+1,B-1}^{t-r_2} \left(\prod_{s=r_2-1}^{1} \tilde{H}_{0,B-1}^{t-s}\right) \cdot$$
$$\left(\prod_{s=1}^{r_1-1} \tilde{H}_{0,B-1}^{t-s,\top}\right) \tilde{H}_{j_1+1,B-1}^{t-r_1,\top} \tilde{X}_{-j_1}^{t-r_1} \varepsilon_{-j_1}^{t-r_1} \tag{29}$$

Finally, we bound the diagonal term:

**Claim 1.**

$$\mathbb{E}\left[\sum_{r=1}^{t} \sum_{j=0}^{B-1} \mathrm{Dg}(t,r,j) \mathbf{1}\left[\tilde{\mathcal{D}}^{0,t-1}\right]\right] \leq \frac{\gamma d}{\zeta(1-\gamma R)}\beta + 16 C_\eta \sigma^2 \gamma^2 RT \frac{1}{T^{\alpha/2}} \tag{30}$$

*Proof.* Notice that we can write

$$\mathrm{Dg}(t,r,j) \leq 4\gamma^2 \left(\beta + |\varepsilon_{-j}^{t-r}|^2 \mathbf{1}[|\varepsilon_{-j}^{t-r}|^2 > \beta]\right) \cdot$$
$$\tilde{X}_{-j}^{t-r,\top} \tilde{H}_{j+1,B-1}^{t-r} \left(\prod_{s=r-1}^{1} \tilde{H}_{0,B-1}^{t-s}\right) \left(\prod_{s=1}^{r-1} \tilde{H}_{0,B-1}^{t-s,\top}\right) \tilde{H}_{j+1,B-1}^{t-r,\top} \tilde{X}_{-j}^{t-r} \tag{31}$$

Further

$$\tilde{X}_{-j}^{t-r,\top} \tilde{H}_{j+1,B-1}^{t-r} \left(\prod_{s=r-1}^{1} \tilde{H}_{0,B-1}^{t-s}\right) \left(\prod_{s=1}^{r-1} \tilde{H}_{0,B-1}^{t-s,\top}\right) \tilde{H}_{j+1,B-1}^{t-r,\top} \tilde{X}_{-j}^{t-r} \mathbf{1}\left[\tilde{\mathcal{D}}^{0,t-1}\right] \leq R \tag{32}$$

Combining the above two we obtain

$$\sum_{r=1}^{t} \sum_{j=0}^{B-1} \mathrm{Dg}(t,r,j) \mathbf{1}\left[\tilde{\mathcal{D}}^{0,t-1}\right] \leq 4\gamma^2 \beta \frac{\mathrm{Tr}\, S_1^t}{\zeta} + R \sum_{r=1}^{t} \sum_{j=0}^{B-1} |\varepsilon_{-j}^{t-r}|^2 \mathbf{1}[|\varepsilon_{-j}^{t-r}|^2 > \beta] \tag{33}$$

where $S_1^t$ is defined in (25). Now taking expectation, and using lemma 6 and Caucy-Schwarz inequality for the first and second terms, respectively, in (33) we obtain the claim. Here we use the fact that $\mathbb{E}\left[|\varepsilon_{-j}^{t-r}|^4\right] \leq 16 C_\eta^2 \sigma^4$ from [48, Theorem 2.1] □

## C.7 Algorithmic stability

In order to bound the cross terms in the variance, we need the notion of algorithmic stability. Here the idea is that if $\phi$ was identity, then $\mathbb{E}\left[\mathrm{Cr}(t,r_1,r_2,j_1,j_2)\right]$ would vanish. But in the non-linear setting, this does not happen due to dependencies between $\varepsilon_{-j}^{t-r}$ and $\tilde{H}_{j+1,B-1}^{t-r} \left(\prod_{s=r-1}^{1} \tilde{H}_{0,B-1}^{t-s}\right)$ through the algorithmic iterates. We can still show that $\mathbb{E}\left[\mathrm{Cr}(t,r_1,r_2,j_1,j_2)\right] \approx 0$ by showing that the iterates depend very weakly on each of the noise co-ordinates $\varepsilon_{-j}^{t-r}$. So our idea is to use algorithmic stability: we re-sample the whole trajectory of $X$ by re-sampling a single noise co-ordinate independently. We

then show that the iterates are not affected much by such a re-sampling, which shows that the iterates are only weakly coupled to each individual noise vector.

To that end, we need some additional notation. We have the data $(X_\tau)_\tau$ and the coupled process $(\tilde{X}_\tau)_\tau$. Let the corresponding (coupled) algorithmic iterates be $(\tilde{a}_i^s : 0 \le s \le N-1, 0 \le i \le B-1)$. Now $\tilde{a}_i^s$ are functions of $X_0$ and noise vectors $\{\eta_{-i}^s : 0 \le s \le N-1, 0 \le i \le S-1\}$. Suppose we re-sample the noise $\eta_{-j}^r$ independently of everything else to get $\bar{\eta}_{-j}^r$. So the new noise samples are:

$$\left(\eta_0^0, \eta_1^0, \cdots, \eta_0^r, \cdots, \eta_{(S-1)-(j+1)}^r, \bar{\eta}_{(S-1)-j}^r, \eta_{(S-1)-(j-1)}^r, \cdots\right).$$

We then run the dynamics in Equation (1) with the new noise samples to obtain $(\bar{X}_\tau)_\tau$ and the new coupled process $(\bar{\tilde{X}}_\tau)_\tau$ obtained through the new noise sequence (but same stationary renewal given in Definition 2), and they satisfy the following:

$$\bar{X}_{-i}^s = \begin{cases} X_{-i}^s, & s < r, 0 \le i \le S-1 \\ X_{-i}^r, & s = r, j \le i \le S-1 \end{cases}$$

$$\bar{\tilde{X}}_{-i}^s = \begin{cases} \tilde{X}_{-i}^s, & s \in \{1, \cdots, r-1, r+1, \cdots, N-1\}, 0 \le i \le S-1 \\ \tilde{X}_{-i}^r, & s = r, j \le i \le S-1 \end{cases}$$

We obtain the iterates $\bar{\tilde{a}}_i^s$ by running the update Equation (13) with the data $\bar{\tilde{X}}_\tau$ instead of $X_\tau$. Accordingly, the algorithmic iterates change to $(\bar{\tilde{a}}_i^s : 0 \le s \le N-1, 0 \le i \le B-1)$ that satisfy

$$\bar{\tilde{a}}_i^s = \tilde{a}_i^s \quad \text{for } s < r, 0 \le i \le B-1$$

This is because, resampling $\eta_\tau$ does note change the value of data $\tilde{X}_{\tau'}$ for $\tau' \le \tau$. Under the setting we have the following lemma:

**Lemma 7.** *Let $\mathcal{A}^{t-1}$ be the following event*

$$\mathcal{A}^{t-1} = \bigcap_{r=0}^{t-1} \bigcap_{j=0}^{B-1} \left\{\|\tilde{a}_j^r - a^*\| \le \|a_0 - a^*\| + \bar{C}\frac{\sqrt{R\beta}}{\zeta\lambda_{\min}}\right\} \tag{34}$$

*For some constant $\bar{C}$ depending only on $C_\eta$, we have for any $1 \le t \le N$*

$$\mathbb{P}\left[\hat{\mathcal{D}}^{0,N-1} \cap \mathcal{A}^{N-1} \cap \cap_{r=0}^{N-1}\mathcal{E}_{0,B-1}^r\right] \ge 1 - \frac{1}{T^\alpha} \tag{35}$$

*Further more, on the event $\mathcal{E}_{0,j}^r \cap \tilde{\mathcal{D}}^{r,N-1} \cap \mathcal{A}^r$ we have:*

$$\bar{\tilde{a}}_i^s = \tilde{a}_i^s, 0 \le s < r, 0 \le i \le B-1 \tag{36}$$

$$\bar{\tilde{a}}_0^r = \tilde{a}_0^r \tag{37}$$

*and for $s \ge r$ we have*

$$\|\bar{\tilde{a}}_i^s - \tilde{a}_i^s\| \le \bar{C}_2\gamma RB\frac{\sqrt{R\beta}}{\zeta\lambda_{\min}} + 8\gamma RB\|a_0 - a^*\| \le \bar{C}_2\gamma RB\frac{\sqrt{R\beta}}{\zeta\lambda_{\min}} \tag{38}$$

We give the proof in Section H

**Remark 3.** *In expression (38), we have suppressed the dependence of $\|a_0 - a^*\|$ for the ease of exposition with the rationale being that since $a_0 = 0$, it sould be lower order compared to $\sqrt{R\beta}$.*

Hence we see from the above lemma that changing a particular noise sample in a particular buffer perturbs the algorithmic iterates by $O(\gamma\text{poly}(RB))$.

Let $\mathcal{R}_{-j}^r$ denote the re-sampling operator corresponding to re-sampling $\eta_{-j}^r$. That is, for any function $f((a_\tau), (X_\tau), (\tilde{X}_\tau))$ we have

$$\mathcal{R}_{-j}^r\left(f((a_\tau), (\tilde{a}_\tau), (X_\tau), (\tilde{X}_\tau))\right) = f((\bar{a}_\tau), (\bar{\tilde{a}}_\tau), (\bar{X}_\tau), (\bar{\tilde{X}}_\tau)) \tag{39}$$

We will drop the subscripts and superscripts on $\mathcal{R}$ when there is no ambiguity on which noise is re-sampled. First we will prove a lemma that bounds the effect of re-sampling.

**Lemma 8.** *On the event $\mathcal{E}_{0,j}^r \cap \tilde{\mathcal{D}}^{0,t-1} \cap \mathcal{A}^{t-1}$, for some constant $C$ depending only on $C_\eta$:*

$$\left\| \tilde{H}_{j+1,B-1}^{t-r} \left( \prod_{s=r-1}^{1} \tilde{H}_{0,B-1}^{t-s} \right) - \mathcal{R}_{-j}^{t-r} \tilde{H}_{j+1,B-1}^{t-r} \left( \prod_{s=r-1}^{1} \mathcal{R}_{-j}^{t-r} \tilde{H}_{0,B-1}^{t-s} \right) \right\|$$

$$\leq \bar{C} \frac{Bt \, \|\phi''\| \, \gamma^2 R^3 B \sqrt{\beta}}{\zeta \lambda_{\min}} \tag{40}$$

*Proof.* First, note that since we are re-sampling $\eta_{-j}^{t-r}$, the only difference between $\mathcal{R}_{-j}^{t-r} \tilde{H}_{j+1,B-1}^{t-r} \left( \prod_{s=r-1}^{1} \mathcal{R}_{-j}^{t-r} \tilde{H}_{0,B-1}^{t-s} \right)$ and $\tilde{H}_{j+1,B-1}^{t-r} \left( \prod_{s=r-1}^{1} \tilde{H}_{0,B-1}^{t-s} \right)$ is that the algorithmic iterates $\tilde{a}_j^s$ that appear in the latter (through $\phi'(\cdot)$) are replaced by $\bar{\tilde{a}}_j^s$ in the former, but the covariates remain the same in both.

Now, the matrix $\tilde{H}_{j+1,B-1}^{t-r} \left( \prod_{s=r-1}^{1} \tilde{H}_{0,B-1}^{t-s} \right)$ is of the form $\prod_{l=1}^{k} A_l$ where $\|A_l\| \leq 1$ under the conditioned events and is of the form $I - 2\gamma\phi'(\tilde{\xi}_{-i}^{t-s})\tilde{X}_{-j}^{t-s}\tilde{X}_{-i}^{t-s,\top}$. Similarly, we write: $\mathcal{R}_{-j}^{t-r} \tilde{H}_{j+1,B-1}^{t-r} \left( \prod_{s=r-1}^{1} \mathcal{R}_{-j}^{t-r} \tilde{H}_{0,B-1}^{t-s} \right) = \prod_{l=1}^{k} \bar{A}_l$ where $\bar{A}_l = \mathcal{R}_{-j}^{t-r} A_l$. Now consider the simple inequality under the condition that $\|A_l\|, \|\bar{A}_l\| \leq 1$

$$\left\| \prod_{l=1}^{k} A_l - \prod_{l=1}^{k} \bar{A}_l \right\| \leq \sum_{l=1}^{k} \|A_l - \bar{A}_l\| \tag{41}$$

Therefore, we will just bound each of the component differences $\|A_l - \bar{A}_l\|$. To this end, consider a typical term $I - 2\gamma\phi'(\tilde{\xi}_{-i}^{t-s})\tilde{X}_{-j}^{t-s}\tilde{X}_{-i}^{t-s,\top}$. We have

$$\left( I - 2\gamma\phi'(\tilde{\xi}_{-i}^{t-s})\tilde{X}_{-i}^{t-s}\tilde{X}_{-i}^{t-s,\top} \right) - \mathcal{R}_{-j}^{t-r} \left( I - 2\gamma\phi'(\tilde{\xi}_{-i}^{t-s})\tilde{X}_{-i}^{t-s}\tilde{X}_{-i}^{t-s,\top} \right)$$

$$= 2\gamma(\phi'(\tilde{\xi}_{-i}^{t-s}) - \mathcal{R}_{-j}^{t-r}\phi'(\tilde{\xi}_{-i}^{t-s}))\tilde{X}_{-i}^{t-s}\tilde{X}_{-i}^{t-s,\top} \tag{42}$$

Now

$$\phi'(\tilde{\xi}_{-i}^{t-s}) - \mathcal{R}_{-j}^{t-r}\phi'(\tilde{\xi}_{-i}^{t-s}) = \frac{\phi(\tilde{a}_i^{t-s,\top}\tilde{X}_{-i}^{t-s}) - \phi(a^{*,\top}\tilde{X}_{-i}^{t-s})}{(\tilde{a}_i^{t-s} - a^*)^\top \tilde{X}_{-i}^{t-s}} - \frac{\phi(\bar{\tilde{a}}_i^{t-s,\top}\tilde{X}_{-i}^{t-s}) - \phi(a^{*,\top}\tilde{X}_{-i}^{t-s})}{(\bar{\tilde{a}}_i^{t-s} - a^*)^\top \tilde{X}_{-i}^{t-s}} \tag{43}$$

Now we can use the following simple result from calculus. Suppose $f$ is a real valued twice continuously differentiable function with bounded second derivative (denoted by $\|f''\|$). Fix $x_0 \in \mathbb{R}$. Let $g(x) = \frac{f(x) - f(x_0)}{x - x_0}$. By the mean value theorem, there exists $\xi$ such that:

$$g'(x) = \frac{f(x_0) - (f(x) + (x_0 - x)f'(x))}{(x - x_0)^2} = \frac{1}{2}f''(\xi)$$

Now for any $x, y$, we have

$$|g(x) - g(y)| = |g'(\xi_1)(x - y)| \leq \frac{1}{2}\|f''\| \, |x - y|$$

for some $\xi_1$ between $x$ and $y$. Here again we use the mean value theorem in the equality above. Now we will apply this result to $\phi$ with $x = \tilde{a}_i^{t-s,\top}\tilde{X}_{-i}^{t-s}$, $y = \bar{\tilde{a}}_i^{t-s,\top}\tilde{X}_{-i}^{t-s}$ and $x_0 = a^{*,\top}\tilde{X}_{-i}^{t-s}$ to get

$$\left| \phi'(\tilde{\xi}_{-i}^{t-s}) - \mathcal{R}_{-j}^{t-r}\phi'(\tilde{\xi}_{-i}^{t-s}) \right| \leq \frac{1}{2}\|\phi''\| \, \left\|\tilde{a}_i^{t-s} - \bar{\tilde{a}}_i^{t-s}\right\| \, \left\|\tilde{X}_{-i}^{t-s}\right\| \tag{44}$$

Now we appeal to lemma 7. In particular, using equation (38) we see that, on the event $\mathcal{E}^r_{0,j} \cap \tilde{\mathcal{D}}^{0,t-1} \cap \mathcal{A}^{t-1}$,

$$\left| \phi'(\tilde{\xi}^{t-s}_{-i}) - \mathcal{R}^{t-r}_{-j} \phi'(\tilde{\xi}^{t-s}_{-i}) \right| \leq \frac{1}{2} \|\phi''\| \, 128C\gamma RB \frac{\sqrt{R\beta}}{\zeta\lambda_{\min}} \sqrt{R}$$

$$= 64C \, \|\phi''\| \, \gamma R^2 B \frac{\sqrt{\beta}}{\zeta\lambda_{\min}} \qquad (45)$$

$$\implies \left\| 2\gamma(\phi'(\tilde{\xi}^{t-s}_{-i}) - \mathcal{R}^{t-r}_{-j} \phi'(\tilde{\xi}^{t-s}_{-i})) \tilde{X}^{t-s}_{-i} \tilde{X}^{t-s,\top}_{-i} \right\| \leq 128C \, \|\phi''\| \, \gamma^2 R^3 B \frac{\sqrt{\beta}}{\zeta\lambda_{\min}} \qquad (46)$$

We now use Equation (41) with $k \leq Bt$ along with Equation (46) to conclude the statement of the lemma. $\qquad \square$

## C.8   Bound $\mathrm{Cr}(t, r_1, r_2, j_1, j_2)$

Next we will bound $\sum_r \sum_{j_1 \neq j_2} \mathrm{Cr}(t, r, r, j_1, j_2)$

**Claim 2.**
$$\left| \mathbb{E}\left[ \sum_{r=1}^{t} \sum_{j_1 \neq j_2} \mathrm{Cr}(t, r, r, j_1, j_2) 1\left[\tilde{\mathcal{D}}^{0,t-1}\right] \right] \right| \leq \bar{C}\left[ \frac{\sigma^2 \gamma^2 RB}{T^{\alpha/2 - 1}} + \frac{\|\phi''\| \, \gamma^4 T^2 R^4 B^2 \sigma^2 \sqrt{\beta}}{\zeta\lambda_{\min}} \right] \qquad (47)$$
*Where $\bar{C}$ is a constant depending only on $C_\eta$*

*Proof.* Let $j_1 < j_2$. We will suppress the arguments of Cr for brevity. First, we re-sample the noise which is ahead in the time, i.e., $\eta^{t-r}_{-j_1}$ (and hence the entry $\varepsilon^{t-r}_{-j_1}$ in the row under consideration).

Let $\mathrm{Cr}'$ denote the resampled version of Cr as defined below

$$\mathrm{Cr}'(t, r, r, j_1, j_2) := 4\gamma^2 \varepsilon^{t-r}_{-j_1} \varepsilon^{t-r}_{-j_2} \mathcal{R}^{t-r}_{-j_1} \left[ \tilde{X}^{t-r,\top}_{-j_2} \tilde{H}^{t-r}_{j_2+1,B-1} \left( \prod_{s=r-1}^{1} \tilde{H}^{t-s}_{0,B-1} \right) \cdot \right.$$

$$\left. \left( \prod_{s=1}^{r-1} \tilde{H}^{t-s,\top}_{0,B-1} \right) \tilde{H}^{t-r,\top}_{j_1+1,B-1} \tilde{X}^{t-r}_{-j_1} \right]$$

$$= 4\gamma^2 \varepsilon^{t-r}_{-j_1} \varepsilon^{t-r}_{-j_2} \tilde{X}^{t-r,\top}_{-j_2} \mathcal{R}^{t-r}_{-j_1} \left( \tilde{H}^{t-r}_{j_2+1,B-1} \right) \mathcal{R}^{t-r}_{-j_1} \left( \prod_{s=r-1}^{1} \tilde{H}^{t-s}_{0,B-1} \right) \cdot$$

$$\mathcal{R}^{t-r}_{-j_1} \left( \prod_{s=1}^{r-1} \tilde{H}^{t-s,\top}_{0,B-1} \right) \mathcal{R}^{t-r}_{-j_1} \left( \tilde{H}^{t-r,\top}_{j_1+1,B-1} \right) \tilde{X}^{t-r}_{-j_1} \qquad (48)$$

where we have used the fact that $\mathcal{R}^{t-r}_{-j_1}$ has no effect on the items from the process $(\tilde{X}_\tau)_\tau$ *that appear* in the expression above. Note the this is **not** $\mathcal{R}^{t-r}_{-j_1} \mathrm{Cr}$, since in $\mathcal{R}^{t-r}_{-j_1} \mathrm{Cr}$ we would have $\bar{\epsilon}^{t-r}_{-j_1}$ instead. Now, since the new algorithmic iterates $(\bar{\tilde{a}}^s_i)$ depend on $\bar{\eta}^{t-r}_{-j_1}$ but not on $\eta^{t-r}_{-j_1}$, it is immediate that

$$\mathbb{E}\left[ \mathrm{Cr}'(t, r, r, j_1, j_2) \right] = 0$$

For convenience, we introduce some notation which is only used in this proof. $\mathrm{Cr}'(t, r, r, j_1, j_2)$ can be written in the form $4\gamma^2 \varepsilon^{t-r}_{-j_1} \varepsilon^{t-r}_{-j_2} K_1$ for some random variable $K_1$ independent of $\varepsilon^{t-r}_{-j_1}$. Under the event $\tilde{\mathcal{D}}^{0,t-1}$, we can easily show that $|K_1| \leq R$ almost surely. Let $\mathcal{F}_K = \sigma(K_1, \varepsilon^{t-r}_{-j_2})$. Let $\mathcal{M} := \{|K_1| \leq R\}$. Clearly, $\tilde{\mathcal{D}}^{0,t-1} \subseteq \mathcal{M}$ and $\varepsilon^{t-r}_{-j_1} \perp \mathcal{F}_K$. We conclude:

$$\left| \mathbb{E}\left[ \mathrm{Cr}' 1\left[\tilde{\mathcal{D}}^{0,t-1}\right] \right] \right| = \left| \mathbb{E}\left[ \mathrm{Cr}' 1\left[\tilde{\mathcal{D}}^{0,t-1}\right] 1\left[\mathcal{M}\right] \right] \right|$$

$$= 4\gamma^2 \left| \mathbb{E}\left[ \mathbb{E}\left[ \varepsilon^{t-r}_{-j_1} 1\left[\tilde{\mathcal{D}}^{0,t-1}\right] \big| \mathcal{F}_K \right] K_1 \varepsilon^{t-r}_{-j_2} 1\left[\mathcal{M}\right] \right] \right|$$

$$\leq 4\gamma^2 \mathbb{E}\left[ \left| \mathbb{E}\left[ \varepsilon^{t-r}_{-j_1} 1\left[\tilde{\mathcal{D}}^{0,t-1}\right] \big| \mathcal{F}_K \right] \right| \cdot |K_1| \cdot |\varepsilon^{t-r}_{-j_2}| 1\left[\mathcal{M}\right] \right] \qquad (49)$$

We note that: $\left| \mathbb{E} \left[ \varepsilon_{-j_1}^{t-r} \mathbb{1} \left[ \tilde{\mathcal{D}}^{0,t-1} \right] | \mathcal{F}_K \right] \right| = \left| \mathbb{E} \left[ \varepsilon_{-j_1}^{t-r} \mathbb{1} \left[ \tilde{\mathcal{D}}^{0,t-1,C} \right] | \mathcal{F}_K \right] \right| \leq$
$\sigma^2 \sqrt{\mathbb{P} \left( \mathbb{1} \left[ \tilde{\mathcal{D}}^{0,t-1,C} \right] | \mathcal{F}_K \right)}$. Using this in Equation (49), and that under event $\mathcal{M}$, $|K_1| \leq R$ we apply Cauchy-Schwarz inequality again to conclude:

$$\left| \mathbb{E} \left[ \mathrm{Cr}' \mathbb{1} \left[ \tilde{\mathcal{D}}^{0,t-1} \right] \right] \right| \leq 4\gamma^2 \sigma^2 R \sqrt{\mathbb{P}(\tilde{\mathcal{D}}^{0,t-1,C})} \leq \frac{4\gamma^2 \sigma^2 R}{T^{\alpha/2}} \tag{50}$$

Using similar technique as lemma 8, we have that on the event $\mathcal{E}_{0,j_1}^r \cap \tilde{\mathcal{D}}^{0,t-1} \cap \mathcal{A}^{t-1}$,

$$\left\| \tilde{H}_{j_2+1,B-1}^{t-r} \left( \prod_{s=r-1}^{1} \tilde{H}_{0,B-1}^{t-s} \right) - \mathcal{R}_{-j_1}^{t-r} \tilde{H}_{j_2+1,B-1}^{t-r} \left( \prod_{s=r-1}^{1} \mathcal{R}_{-j_2}^{t-r} \tilde{H}_{0,B-1}^{t-s} \right) \right\|$$
$$\leq \bar{C} \frac{T \| \phi'' \| \gamma^2 R^3 B \sqrt{\beta}}{\zeta \lambda_{\min}} \tag{51}$$

Therefore, on the event $\mathcal{E}_{0,j_1}^r \cap \mathcal{E}_{0,j_2}^r \cap \tilde{\mathcal{D}}^{0,t-1} \cap \mathcal{A}^{t-1}$, we have

$$\left| \mathrm{Cr} - \mathrm{Cr}' \right| \leq \gamma^4 \bar{C} R^4 \left| \varepsilon_{-j_1}^{t-r} \varepsilon_{-j_2}^{t-r} \right| T \| \phi'' \| B \frac{\sqrt{\beta}}{\zeta \lambda_{\min}}$$
$$\implies \mathbb{E} \left[ \left| \mathrm{Cr} - \mathrm{Cr}' \right| \mathbb{1} \left[ \mathcal{E}_{0,j_1}^r \cap \mathcal{E}_{0,j_2}^r \cap \tilde{\mathcal{D}}^{0,t-1} \cap \mathcal{A}^{t-1} \right] \right] \leq \bar{C} \| \phi'' \| \gamma^4 T R^4 B \frac{\sigma^2 \sqrt{\beta}}{\zeta \lambda_{\min}} \tag{52}$$

We note that over the event $\tilde{\mathcal{D}}^{0,t-1}$, we must have $|\mathrm{Cr} - \mathrm{Cr}'| \leq 2R |\varepsilon_{-j_1}^{t-r} \varepsilon_{-j_2}^{t-r}|$. Combining this with Equation (52) and noting that $\mathbb{P} \left( \mathcal{E}_{0,j_1}^r \cap \mathcal{E}_{0,j_2}^r \cap \tilde{\mathcal{D}}^{0,t-1} \cap \mathcal{A}^{t-1} \right) \geq 1 - \frac{1}{T^\alpha}$, we conclude:

$$\mathbb{E} \left[ \left| \sum_{r=1}^{t} \sum_{j_1 \neq j_2} \mathrm{Cr}(t,r,r,j_1,j_2) - \mathrm{Cr}'(t,r,r,j_1,j_2) \right| \mathbb{1} \left[ \tilde{\mathcal{D}}^{0,t-1} \right] \right]$$
$$\leq \bar{C} \left[ \| \phi'' \| \gamma^4 T^2 R^4 B^2 \frac{\sigma^2 \sqrt{\beta}}{\zeta \lambda_{\min}} + \sigma^2 \gamma^2 R T B \frac{1}{T^{\alpha/2}} \right] \tag{53}$$

Hence combining (50) and (53) we conclude the statement of the claim. $\qquad \square$

Next we want to bound $\mathrm{Cr}(t,r_1,r_2,j_1,j_2)$ for $r_2 > r_1$ and arbitrary $j_1$ and $j_2$. Recall the definition of $\tilde{a}_B^{t-1,v}$ from (23). Via simple rearrangement of summation, we can express $\sum_{r_2 > r_1} \sum_{j_1,j_2} \mathrm{Cr}(t,r_1,r_2,j_1,j_2)$ in terms of $\tilde{a}_B^{t-r_1-1,v}$ as follows.

**Lemma 9.**
$$\sum_{r_2 > r_1} \sum_{j_1,j_2} \mathrm{Cr}(t,r_1,r_2,j_1,j_2)$$
$$= 2\gamma \sum_{r_1=1}^{t-1} \sum_{j_1=0}^{B-1} (\tilde{a}_B^{t-r_1-1,v})^\top \left( \prod_{s=r_1}^{1} \tilde{H}_{0,B-1}^{t-s} \right) \left( \prod_{s=1}^{r_1-1} \tilde{H}_{0,B-1}^{t-s,\top} \right) \tilde{H}_{j_1+1,B-1}^{t-r_1,\top} \tilde{X}_{-j_1}^{t-r_1} \varepsilon_{-j_1}^{t-r_1} \tag{54}$$

**Claim 3.**
$$\left| \mathbb{E} \left[ \sum_{r_1 \neq r_2} \sum_{j_1,j_2} \mathrm{Cr}(t,r_1,r_2,j_1,j_2) \mathbb{1} \left[ \tilde{\mathcal{D}}^{0,t-1} \right] \right] \right| \leq \bar{C} \gamma^2 R (Bt)^2 \sigma^2 \frac{1}{T^{\alpha/2}} +$$
$$\bar{C} \left( \| \phi'' \| \gamma^3 T^2 R^3 B \frac{\sqrt{\beta}}{\zeta \lambda_{\min}} + \gamma^2 T R B \right) \sqrt{R \sigma^2} \sqrt{\sup_{s \leq N-1} \mathbb{E} \left[ \| \tilde{a}_B^{s,v} \|^2 \mathbb{1} \left[ \tilde{\mathcal{D}}^{0,s} \right] \right]} \tag{55}$$

The proof of the claim essentially proceeds similar to that of Claim 2 but with additional complications. We refer to Section H for the proof.

Combining everything in this section we have the following proposition.

**Proposition 1.** *Let*

$$\tilde{\mathcal{V}}_{t-1} = \mathbb{E}\left[\left\|\tilde{a}_B^{t-1,v}\right\|^2 1\left[\tilde{\mathcal{D}}^{0,t-1}\right]\right] \tag{56}$$

*Then for some constant $\bar{C}$ which depends only on $C_\eta$:*

$$\sup_{s\leq N-1} \tilde{\mathcal{V}}_s \leq \frac{2\gamma d}{\zeta(1-\gamma R)}\beta + \bar{C}\left\|\phi''\right\|\gamma^4 T^2 R^4 B^2 \frac{\sigma^2\sqrt{\beta}}{\zeta\lambda_{\min}} + \bar{C}\sigma^2\gamma^2 RT^2\frac{1}{T^{\alpha/2}} +$$
$$\bar{C}R\sigma^2\left(\left\|\phi''\right\|^2\gamma^6 T^4 R^6 B^2\frac{\beta}{\zeta^2\lambda_{\min}^2} + \gamma^4 T^2 R^2 B^2\right) \tag{57}$$

*Proof.* In the whole proof, we will denote any large enough constant depending on $C_\eta$ by $\bar{C}$. From claims 1, 2 and 3 along with equation (27) we have

$$\mathbb{E}\left[\left\|\tilde{a}_B^{t-1,v}\right\|^2 1\left[\tilde{\mathcal{D}}^{0,t-1}\right]\right]$$
$$\leq \frac{\gamma d}{\zeta(1-\gamma R)}\beta + \bar{C}\left\|\phi''\right\|\gamma^4 T^2 R^4 B^2\frac{\sigma^2\sqrt{\beta}}{\zeta\lambda_{\min}} + \bar{C}\sigma^2\gamma^2 RT\frac{1}{T^{\alpha/2}}(1+B+T) +$$
$$\left(\bar{C}\left\|\phi''\right\|\gamma^3 T^2 R^3 B\frac{\sqrt{\beta}}{\zeta\lambda_{\min}} + \gamma^2 TRB\right)\sqrt{R\sigma^2}\sqrt{\sup_{s\leq N-1}\mathbb{E}\left[\|\tilde{a}_B^{s,v}\|^2 1\left[\tilde{\mathcal{D}}^{0,s}\right]\right]} \tag{58}$$

Thus

$$\sup_{s\leq N-1}\tilde{\mathcal{V}}_s \leq \frac{\gamma d}{\zeta(1-\gamma R)}\beta + \bar{C}\left\|\phi''\right\|\gamma^4 T^2 R^4 B^2\frac{\sigma^2\sqrt{\beta}}{\zeta\lambda_{\min}} + \bar{C}\sigma^2\gamma^2 RT^2\frac{1}{T^{\alpha/2}} +$$
$$\bar{C}\left(\left\|\phi''\right\|\gamma^3 T^2 R^3 B\frac{\sqrt{\beta}}{\zeta\lambda_{\min}} + \gamma^2 TRB\right)\sqrt{R\sigma^2}\sqrt{\sup_{s\leq N-1}\tilde{\mathcal{V}}_s} \tag{59}$$

Finally, we need to solve the above recursive relation. We note a simple fact: Let $c_1, c_2 > 0$ be constants and let $x > 0$ satisfy

$$x^2 \leq c_1 + c_2 x \tag{60}$$

then

$$x^2 \leq \frac{1}{4}\left(c_2 + \sqrt{c_2^2 + 4c_1}\right)^2 \leq c_2^2 + 2c_1 \tag{61}$$

where in the last inequality above we used the fact that $(a+b)^2 \leq 2(a^2+b^2)$.

Thus,

$$\sup_{s\leq N-1}\tilde{\mathcal{V}}_s \leq \frac{2\gamma d}{\zeta(1-\gamma R)}\beta + \bar{C}\left\|\phi''\right\|\gamma^4 T^2 R^4 B^2\frac{\sigma^2\sqrt{\beta}}{\zeta\lambda_{\min}} + \bar{C}\sigma^2\gamma^2 RT^2\frac{1}{T^{\alpha/2}} +$$
$$\bar{C}R\sigma^2\left(\left\|\phi''\right\|^2\gamma^6 T^4 R^6 B^2\frac{\beta}{\zeta^2\lambda_{\min}^2} + \gamma^4 T^2 R^2 B^2\right) \tag{62}$$

$\square$

### C.9 Bias of last iterate

In this part, we will bound the expectation of the bias term $\left\|\tilde{a}_B^{t-1,b} - a^*\right\|^2$.

**Theorem 5.** *For some universal constant $c_0$:*

$$\mathbb{E}\left[\left\|\tilde{a}_B^{t-1,b} - a^*\right\|^2 1\left[\tilde{\mathcal{D}}^{0,t-1}\right]\right] \leq \|a_0 - a^*\|^2 (1 - c_0\zeta\gamma B\lambda_{\min})^t \tag{63}$$

*where $\tilde{a}^{t-1,b} - a^*$ is defined in* (22)

*Proof.* Define $x_v = \prod_{s=0}^{v-1} \tilde{H}_{0,B-1}^{s,\top}(a_0 - a^*)$. Here, we consider the event $\mathcal{G}_v$ considered in Claim 6 in the proof of Theorem 10, and show that for some universal constant $q_0 > 0$,

$$\mathbb{P}(\|\tilde{H}_{0,B-1}^{v,\top} x_v\|^2 \geq (1 - \zeta\gamma\lambda_{\min}(G))\|x_v\|^2 |\tilde{\mathcal{D}}^{0,t-1}, x_v) \geq q_0 \tag{64}$$

From Theorem 10 we also note that conditioned on $\tilde{\mathcal{D}}^{0,t-1}$, almost surely:

$$\|\tilde{H}_{0,B-1}^{v,\top} x_v\|^2 \leq 1$$

We let $\mathcal{G}_v$ be the event lower bounded in Equation (64).

$$
\begin{aligned}
\mathbb{E}\left[\|x_{v+1}\|^2 |\tilde{\mathcal{D}}^{0,t-1}\right] &= \mathbb{E}\left[\|\tilde{H}_{0,B-1}^{v,\top} x_v\|^2 |\tilde{\mathcal{D}}^{0,t-1}\right] \\
&= \mathbb{E}\left[\|\tilde{H}_{0,B-1}^{v,\top} x_v\|^2 \mathbb{1}\left[\mathcal{G}_v\right] + \|\tilde{H}_{0,B-1}^{v,\top} x_v\|^2 \mathbb{1}\left[\mathcal{G}_v^C\right] |\tilde{\mathcal{D}}^{0,t-1}\right] \\
&\leq \mathbb{E}\left[(1 - \gamma\zeta\lambda_{\min}(G))\|x_v\|^2 \mathbb{1}\left[\mathcal{G}_v\right] + \|x_v\|^2 \mathbb{1}\left[\mathcal{G}_v^C\right] |\tilde{\mathcal{D}}^{0,t-1}\right] \\
&= \mathbb{E}\left[\|x_v\|^2 \left[1 - \gamma\zeta\lambda_{\min}(G)\,\mathbb{P}(\mathcal{G}_v|\tilde{\mathcal{D}}^{0,t-1}\,x_v)\right] |\tilde{\mathcal{D}}^{0,t-1}\right] \\
&\leq \mathbb{E}\left[\|x_v\|^2 \left[1 - \gamma\zeta\lambda_{\min}(G)\,q_0\right] |\tilde{\mathcal{D}}^{0,t-1}\right]
\end{aligned}
\tag{65}
$$

Unrolling the recursion given by Equation (65), and noting that $\tilde{a}_B^{t-1,b} - a^* = x_t$, we conclude

$$\mathbb{E}\left[\left\|\tilde{a}_B^{t-1,b} - a^*\right\|^2 \mathbb{1}\left[\tilde{\mathcal{D}}^{0,t-1}\right]\right] \leq (1 - c_0\gamma B\lambda_{\min}\zeta)^t.$$

Hence we have the theorem.

$\square$

## C.10 Average iterate: bias-variance decomposition

In the part, we will consider the tail-averaged iterate where a generic row is given by

$$\hat{\tilde{a}}_{t_0,N} = \frac{1}{N - t_0} \sum_{t=t_0+1}^{N} \tilde{a}_B^{t-1} \tag{66}$$

where $t_0 \in \{0, 1, \cdots, N-1\}$.

Thus we can write $\hat{a}_{t_0,N} - a^*$ as

$$\hat{\tilde{a}}_{t_0,N} - a^* = (\hat{\tilde{a}}_{t_0,N}^v) + (\hat{\tilde{a}}_{t_0,N}^b - a^*) \tag{67}$$

where

$$\hat{\tilde{a}}_{t_0,N}^v = \frac{1}{N - t_0} \sum_{t=t_0+1}^{N} (\tilde{a}_B^{t-1,v}) \tag{68}$$

$$\hat{\tilde{a}}_{t_0,N}^b - a^* = \frac{1}{N - t_0} \sum_{t=t_0+1}^{N} (\tilde{a}_B^{t-1,b} - a^*) \tag{69}$$

## C.11 Variance of average iterate

**Remark 4.** *From now on we will use the following notation:*

$$\sum_t \equiv \sum_{t=t_0+1}^{N}$$

$$\sum_{t_1,t_2} \equiv \sum_{t_1,t_2=t_0+1}^{N}$$

$$\sum_{t_1 \neq t_2} \equiv \sum_{\substack{t_1,t_2=t_0+1 \\ t_1 \neq t_2}}^{N}$$

$$\sum_{t_2 > t_1} \equiv \sum_{t_1=t_0+1}^{N-1} \sum_{t_2=t_1+1}^{N}$$

Next we expand $\left\| \hat{\tilde{a}}_{t_0,N}^v \right\|^2$

$$
\left\| \hat{\tilde{a}}_{t_0,N}^v \right\|^2 = \frac{1}{(N-t_0)^2} \sum_t \left\| \tilde{a}_B^{t-1,v} \right\|^2 +
$$
$$
\frac{1}{(N-t_0)^2} \sum_{t_1 \neq t_2} (\tilde{a}_B^{t_2-1,v})^\top (\tilde{a}_B^{t_1-1,v}) \tag{70}
$$

**Claim 4.** *For $t_2 > t_1$*

$$
\left| \mathbb{E} \left[ \left[ (\tilde{a}_B^{t_2-1,v})^\top \left( (\tilde{a}_B^{t_1-1,v}) - \left( \prod_{s=t_2-t_1}^{1} \tilde{H}_{0,B-1}^{t_2-s} \right) (\tilde{a}_B^{t_1-1,v}) \right) \right] \mathbb{1} \left[ \tilde{\mathcal{D}}^{0,N-1} \right] \right] \right|
$$
$$
\leq C_1 \operatorname{Poly}(R,B,\beta,1/\zeta,1/\lambda_{\min},\|\phi''\|) \left( \gamma^{7/2}T^2 + \gamma^5 T^3 + \gamma^6 T^4 \right) \tag{71}
$$

*Proof.* From (23) we can write

$$
(\tilde{a}_B^{t_2-1,v})^\top = (\tilde{a}_B^{t_1-1,v})^\top \left( \prod_{s=t_2-t_1}^{1} \tilde{H}_{0,B-1}^{t_2-s} \right) +
$$
$$
2\gamma \sum_{r=1}^{t_2-t_1} \sum_{j=0}^{B-1} \varepsilon_{-j}^{t_2-r} \tilde{X}_{-j}^{t_2-r,\top} \tilde{H}_{j+1,B-1}^{t_2-r} \left( \prod_{s=r-1}^{1} \tilde{H}_{0,B-1}^{t_2-s} \right) \tag{72}
$$

Hence

$$
(\tilde{a}_B^{t_2-1,v})^\top (\tilde{a}_B^{t_1-1,v}) = (\tilde{a}_B^{t_1-1,v})^\top \left( \prod_{s=t_2-t_1}^{1} \tilde{H}_{0,B-1}^{t_2-s} \right) (\tilde{a}_B^{t_1-1,v}) +
$$
$$
2\gamma \sum_{r=1}^{t_2-t_1} \sum_{j=0}^{B-1} \varepsilon_{-j}^{t_2-r} \tilde{X}_{-j}^{t_2-r,\top} \tilde{H}_{j+1,B-1}^{t_2-r} \left( \prod_{s=r-1}^{1} \tilde{H}_{0,B-1}^{t_2-s} \right) (\tilde{a}_B^{t_1-1,v}) \tag{73}
$$

Now recall the noise re-sampling operator $\mathcal{R}_{-j}^{t_2-r}$ from (39). It is easy to see that

$$
\mathbb{E} \left[ 2\gamma \sum_{r=1}^{t_2-t_1} \sum_{j=0}^{B-1} \varepsilon_{-j}^{t_2-r} \mathcal{R}_{-j}^{t_2-r} \left[ \tilde{X}_{-j}^{t_2-r,\top} \tilde{H}_{j+1,B-1}^{t_2-r} \left( \prod_{s=r-1}^{1} \tilde{H}_{0,B-1}^{t_2-s} \right) (\tilde{a}_B^{t_1-1,v}) \right] \right] = 0 \tag{74}
$$

(Note that $\mathcal{R}_{-j}^{t_2-r} \tilde{X}_{-j}^{t_2-r,\top} = \tilde{X}_{-j}^{t_2-r,\top}$ )

Thus, using the decomposition

$$
\tilde{\mathcal{D}}^{0,N-1} = \tilde{\mathcal{D}}^{0,t_2-r-1} \cap \tilde{\mathcal{D}}^{t_2-r+1,N-1} \cap \tilde{\mathcal{D}}_{-j}^{t_2-r} \cap \cap_{i=0}^{j-1} \tilde{\mathcal{C}}_{-i}^{t_2-r}
$$

we get

$$
\left| \mathbb{E}\left[ 2\gamma \sum_{r=1}^{t_2-t_1} \sum_{j=0}^{B-1} \varepsilon_{-j}^{t_2-r} \tilde{X}_{-j}^{t_2-r,\top} \cdot \right.\right.
$$

$$
\left.\left. \mathcal{R}_{-j}^{t_2-r}\left[ \tilde{H}_{j+1,B-1}^{t_2-r}\left( \prod_{s=r-1}^{1} \tilde{H}_{0,B-1}^{t_2-s} \right)(\tilde{a}_B^{t_1-1,v})\right] 1\left[\tilde{\mathcal{D}}^{0,N-1}\right]\right]\right|
$$

$$
\leq 4\gamma^2 R(Bt_1)(B(t_2-t_1))C_\eta \sigma^2 \frac{1}{T^{\alpha/2}}
$$

$$
\leq 4\gamma^2 R C_\eta \sigma^2 T^2 \frac{1}{T^{\alpha/2}} \tag{75}
$$

Next we need to bound the effect due to noise re-sampling. On the event $\tilde{\mathcal{D}}^{0,N-1} \cap \mathcal{A}^{N-1} \cap \cap_{r=0}^{N-1}\mathcal{E}_{0,B-1}^{r}$, we have

$$
\left\|\tilde{H}_{j+1,B-1}^{t_2-r}\left( \prod_{s=r-1}^{1} \tilde{H}_{0,B-1}^{t_2-s} \right) - \mathcal{R}_{-j}^{t_2-r}\left[ \tilde{H}_{j+1,B-1}^{t_2-r}\left( \prod_{s=r-1}^{1} \tilde{H}_{0,B-1}^{t_2-s} \right)\right]\right\|
$$

$$
\leq C\|\phi''\|\gamma^2 T R^3 B \frac{\sqrt{\beta}}{\zeta\lambda_{\min}} \tag{76}
$$

Thus

$$
2\gamma \left| \mathbb{E}\left[ \sum_{r=1}^{t_2-t_1} \sum_{j=0}^{B-1} \varepsilon_{-j}^{t_2-r} \tilde{X}_{-j}^{t_2-r,\top} 1\left[\tilde{\mathcal{D}}^{0,N-1}\right] \cdot \right.\right.
$$

$$
\left.\left. \left( \tilde{H}_{j+1,B-1}^{t_2-r}\left( \prod_{s=r-1}^{1} \tilde{H}_{0,B-1}^{t_2-s} \right) - \mathcal{R}_{-j}^{t_2-r}\left[ \tilde{H}_{j+1,B-1}^{t_2-r}\left( \prod_{s=r-1}^{1} \tilde{H}_{0,B-1}^{t_2-s} \right)\right]\right)(\tilde{a}_B^{t_1-1,v})\right]\right|
$$

$$
\leq \bar{C}\gamma\left( \|\phi''\|\gamma^2 T R^3 B \frac{\sqrt{\beta}}{\zeta\lambda_{\min}}\right) B(t_2-t_1)\sqrt{R\sigma^2}\mathbb{E}\left[\left\|(\tilde{a}_B^{t_1-1,v})\right\| 1\left[\tilde{\mathcal{D}}^{0,t_1-1}\right]\right]
$$

$$
+ \bar{C}\gamma^2(2R)\sigma^2(Bt_1)(B(t_2-t_1))\frac{1}{T^{\alpha/2}} \tag{77}
$$

Now from proposition 1, there is a constant $C_1$ such that

$$
\left( \mathbb{E}\left[\left\|\tilde{a}^{t_1-1,v}\right\| 1\left[\tilde{\mathcal{D}}^{0,t_1-1}\right]\right]\right)^2 \leq
$$

$$
C_1\left( \frac{\gamma d}{\zeta(1-\gamma R)}\beta + \|\phi''\|\gamma^4 T^2 R^4 B^2 \frac{\sigma^2\sqrt{\beta}}{\zeta\lambda_{\min}} + \sigma^2\gamma^2 RT^2 \frac{1}{T^{\alpha/2}}+ \right.
$$

$$
\left. R\sigma^2\left( \|\phi''\|^2\gamma^6 T^4 R^6 B^2 \frac{\beta}{\zeta^2\lambda_{\min}^2} + \gamma^4 T^2 R^2 B^2\right)\right) \tag{78}
$$

So

$$
2\gamma \left| \mathbb{E}\left[ \sum_{r=1}^{t_2-t_1} \sum_{j=0}^{B-1} \varepsilon_{-j}^{t_2-r} \tilde{X}_{-j}^{t_2-r,\top} 1\left[\tilde{\mathcal{D}}^{0,N-1}\right] \cdot \right.\right.
$$

$$
\left.\left. \left( \tilde{H}_{j+1,B-1}^{t_2-r}\left( \prod_{s=r-1}^{1} \tilde{H}_{0,B-1}^{t_2-s} \right) - \mathcal{R}_{-j}^{t_2-r}\left[ \tilde{H}_{j+1,B-1}^{t_2-r}\left( \prod_{s=r-1}^{1} \tilde{H}_{0,B-1}^{t_2-s} \right)\right]\right)(\tilde{a}_B^{t_1-1,v})\right]\right|
$$

$$
\leq \mathrm{Poly}(R,B,\beta,1/\zeta,1/\lambda_{\min},\|\phi''\|)\left( \gamma^{7/2}T^2 + \gamma^5 T^3 + \gamma^6 T^4\right) \tag{79}
$$

where we absorbed terms involving $\frac{1}{T^\alpha}$ since $\alpha$ is taken to be large.

$\square$

**Claim 5.**

$$\left| \mathbb{E}\left[ (\tilde{a}_B^{t_1-1,v})^\top \sum_{t_2>t_1} \left( \prod_{s=t_2-t_1}^{1} \tilde{H}_{0,B-1}^{t_2-s} \right) (\tilde{a}_B^{t_1-1,v}) 1\left[ \tilde{\mathcal{D}}^{0,N-1} \right] \right] \right|$$

$$\leq \mathcal{V}_{t_1-1} \frac{C}{\zeta\gamma B\lambda_{\min}} + 16(N-t_1)\gamma^2 RC_\eta \sigma^2 T^2 \frac{1}{T^{\alpha/2}} \tag{80}$$

where $\mathcal{V}_{t_1-1}$ is defined in (56).

*Proof.* Note that

$$\prod_{s=t_2-t_1}^{1} \tilde{H}_{0,B-1}^{t_2-s} = \prod_{s=t_1}^{t_2-1} \tilde{H}_{0,B-1}^{s}$$

From theorem 11, there is a universal constant $C$ such that with $\delta = \frac{1}{2T^\alpha}$ we have

$$\mathbb{P}\left[ \left\| \sum_{t_2>t_1} \prod_{s=t_1}^{t_2-1} \tilde{H}_{0,B-1}^{s} \right\| > C\left( d + \log\frac{N}{\delta} + \frac{1}{\zeta\gamma B\lambda_{\min}} \right) |\tilde{\mathcal{D}}^{t_1,N-1} \right] \leq \frac{1}{2T^\alpha} \tag{81}$$

Since $\mathbb{P}\left[ \tilde{\mathcal{D}}^{t_1,N-1} \right] \leq \frac{1}{2T^\alpha}$ we obtain

$$\mathbb{P}\left[ \left\| \sum_{t_2>t_1} \prod_{s=t_1}^{t_2-1} \tilde{H}_{0,B-1}^{s} \right\| > C\left( d + \log\frac{N}{\delta} + \frac{1}{\zeta\gamma B\lambda_{\min}} \right) \right] \leq \frac{1}{T^\alpha} \tag{82}$$

Choosing $\gamma$ such that

$$d + \log\frac{N}{\delta} \leq \frac{1}{\zeta\gamma B\lambda_{\min}}$$

we get

$$\mathbb{P}\left[ \left\| \sum_{t_2>t_1} \prod_{s=t_1}^{t_2-1} \tilde{H}_{0,B-1}^{s} \right\| > \frac{C}{\zeta\gamma B\lambda_{\min}} \right] \leq \frac{1}{T^\alpha} \tag{83}$$

Thus conditioning on the event

$$\left\| \sum_{t_2>t_1} \prod_{s=t_1}^{t_2-1} \tilde{H}_{0,B-1}^{s} \right\| \leq \frac{C}{\zeta\gamma B\lambda_{\min}}$$

we obtain

$$\left| \mathbb{E}\left[ (\tilde{a}_B^{t_1-1,v})^\top \sum_{t_2>t_1} \left( \prod_{s=t_2-t_1}^{1} \tilde{H}_{0,B-1}^{t_2-s} \right) (\tilde{a}_B^{t_1-1,v}) 1\left[ \tilde{\mathcal{D}}^{0,N-1} \right] \right] \right|$$

$$\leq \mathbb{E}\left[ \left\| \tilde{a}_B^{t_1-1,v} \right\|^2 1\left[ \tilde{\mathcal{D}}^{0,t-1} \right] \right] \frac{C}{\zeta\gamma B\lambda_{\min}} + (N-t_1)4\gamma^2 R(4C_\eta\sigma^2)(Bt_1)^2 \frac{1}{T^{\alpha/2}}$$

$$\leq \mathcal{V}_{t_1-1} \frac{C}{\zeta\gamma B\lambda_{\min}} + 16(N-t_1)\gamma^2 RC_\eta \sigma^2 T^2 \frac{1}{T^{\alpha/2}} \tag{84}$$

$\square$

Thus combining everything we have the following theorem

**Theorem 6.** *Suppose $\gamma \gtrsim \frac{1}{T}$. Then*

$$\mathbb{E}\left[ \left\| \hat{\tilde{a}}_{t_0,N}^v \right\|^2 1\left[ \tilde{\mathcal{D}}^{0,N-1} \right] \right] \leq C_1 \frac{d\beta}{\zeta^2\lambda_{\min}B(N-t_0)} + \bar{P}\cdot\left( \gamma^{7/2}T^2 + \gamma^6 T^4 \right) \tag{85}$$

$\bar{P} = \text{Poly}(R, B, \beta, 1/\zeta, 1/\lambda_{\min}, \|\phi''\|, C_\eta)$ *and* $C_1 > 0$ *is some constant.*

## C.12 Bias of average iterate

**Theorem 7.** *There are constants $C, c_1, c_2$ such that :*

$$\mathbb{E}\left[\left\|\hat{\bar{a}}_{t_0,N}^b - a^*\right\|^2 1\left[\tilde{\mathcal{D}}^{0,N-1}\right]\right] \leq C\|a_0 - a\|^2 \left[e^{-c_2\zeta\gamma B\lambda_{\min}t_0} \min\left\{1, \frac{1}{(N-t_0)\zeta\gamma B\lambda_{\min}}\right\}\right]$$
(86)

The proof follows from an application of Theorem 5.

## C.13 Proof of Theorem 3

*Proof.* Let $\bar{P}$ denote the polynomial in Theorem 6. Theorems 7 and 6, imply for every row of the coupled iterate:

$$\mathbb{E}\left[\|\hat{\bar{a}}_{t_0,N} - a^*\|^2 1\left[\tilde{\mathcal{D}}^{0,N-1}\right]\right] \leq 2\mathbb{E}\left[\|\hat{\bar{a}}_{t_0,N}^v\|^2 1\left[\tilde{\mathcal{D}}^{0,N-1}\right]\right] + 2\mathbb{E}\left[\|\hat{\bar{a}}_{t_0,N}^b - a^*\|^2 1\left[\tilde{\mathcal{D}}^{0,N-1}\right]\right]$$

$$\leq C_2 \frac{d\beta}{\zeta^2\lambda_{\min}(G)B(N-t_0)} + \bar{P}\cdot\left(\gamma^{7/2}T^2 + \gamma^6 T^4\right)$$

$$+ C\|a_0 - a\|^2\left[e^{-c_2\zeta\gamma B\lambda_{\min}t_0}\min\left\{1, \frac{1}{(N-t_0)\zeta\gamma B\lambda_{\min}}\right\}\right]$$
(87)

Thus for the actual process we can use the following decomposition

$$\mathbb{E}\left[\|\hat{a}_{t_0,N} - a^*\|^2 1\left[\mathcal{D}^{0,N-1}\right]\right] \leq \mathbb{E}\left[\|\hat{a}_{t_0,N} - a^*\|^2 1\left[\hat{\mathcal{D}}^{0,N-1}\right]\right] +$$

$$\mathbb{E}\left[\|\hat{a}_{t_0,N} - a^*\|^2 1\left[\mathcal{D}^{0,N-1}\right] 1\left[\tilde{\mathcal{D}}^{0,N-1,C}\right]\right]$$

$$\leq \mathbb{E}\left[\|\hat{a}_{t_0,N} - a^*\|^2 1\left[\hat{\mathcal{D}}^{0,N-1}\right]\right] + C\gamma^2 T^2 R C_\eta \sigma^2 \frac{1}{T^{\alpha/2}}$$

$$+ 2\|a_0 - a^*\|^2 \frac{1}{T^\alpha}$$

where we used the fact on the event $\mathcal{D}^{0,N-1}$

$$\|\hat{a}_{t_0,N} - a^*\|^2 \leq \frac{1}{N-t_0}\sum_{t=t_0+1}^{N}\left(2\|a_0 - a^*\|^2 + 2(Bt)(4\gamma^2 R\sum_{r=1}^{t}\sum_{j=0}^{B-1}|\varepsilon_{-j}^{t-r}|^2)\right) \quad (88)$$

and then used Cauchy-Schwarz inequality for the expectation over $\tilde{\mathcal{D}}^{0,N-1,C}$.

Now using lemma 4 we get

$$\mathbb{E}\left[\|\hat{a}_{t_0,N} - a^*\|^2 1\left[\hat{\mathcal{D}}^{0,N-1}\right]\right] \leq \mathbb{E}\left[\left\|\hat{\bar{a}}_{t_0,N} - a^*\right\|^2 1\left[\hat{\mathcal{D}}^{0,N-1}\right]\right] + C\gamma^2 R^2 T^2 \frac{1}{T^\alpha} \quad (89)$$

since we are choosing $u$ such that $\rho^u \leq \frac{1}{T^\alpha}$. Using $\hat{\mathcal{D}}^{0,N-1} \subset \tilde{\mathcal{D}}0, N-1$ we get

$$\mathbb{E}\left[\|\hat{a}_{t_0,N} - a^*\|^2 1\left[\mathcal{D}^{0,N-1}\right]\right] \leq \mathbb{E}\left[\left\|\hat{\bar{a}}_{t_0,N} - a^*\right\|^2 1\left[\tilde{\mathcal{D}}^{0,N-1}\right]\right] + C\gamma^2 R\sigma^2 \frac{1}{T^{\alpha/2-2}} \quad (90)$$

where we absorbed all terms of order $\frac{1}{T^\alpha}$ (including those depending on $\|a_0 - a^*\|$) to the last term in the above display.

Thus

$$\mathbb{E}\left[\|\hat{\bar{a}}_{t_0,N} - a^*\|^2 1\left[\tilde{\mathcal{D}}^{0,N-1}\right]\right] \leq C_1 \frac{d\beta}{\zeta^2\lambda_{\min}(G)B(N-t_0)} + \bar{P}\cdot\left(\gamma^{7/2}T^2 + \gamma^6 T^4\right)$$

$$+ C_2\|a_0 - a\|^2\left[\frac{e^{-c_2\zeta\gamma B\lambda_{\min}t_0}}{(N-t_0)\zeta\gamma B\lambda_{\min}}\right]$$

$$+ C_3\gamma^2 R\sigma^2 \frac{1}{T^{\alpha/2-2}} \quad (91)$$

Summing over all the rows we get a bound on the Frobenius norm. Lastly if the event $\mathcal{D}^{0,N-1}$ does not occur, the $\hat{A}_{t_0,N}$ is the zero matrix and hence

$$\mathbb{E}\left[\|\hat{A}_{t_0,N} - A^*\|_{\mathsf{F}}^2 \mathbb{1}\left[\mathcal{D}^{0,N-1,C}\right]\right] \le \|A^*\|^2 \frac{1}{T^\alpha}. \tag{92}$$

Therefore:

$$
\begin{aligned}
\mathbb{E}\left[\|\hat{A}_{t_0,N} - A^*\|_{\mathsf{F}}^2\right] \le{}& \bar{C}\frac{d^2\beta}{\zeta^2\lambda_{\min}(G)\,B(N-t_0)} + \bar{P}\cdot d\left(\gamma^{7/2}T^2 + \gamma^6 T^4\right) \\
&+ \bar{C}\,\|A_0 - A^*\|_{\mathsf{F}}^2\left[\frac{e^{-c_2\zeta\gamma B\lambda_{\min}t_0}}{(N-t_0)\zeta\gamma B\lambda_{\min}}\right] \\
&+ \bar{C}\gamma^2 R\sigma^2 d\frac{1}{T^{\alpha/2-2}}
\end{aligned}
\tag{93}
$$

$\square$

# D  Concentration Under Stationary Measure

In this section, we will consider the process $\mathsf{NLDS}(A^*, \mu, \phi)$ and the concentration of measure under its stationary distribution. In what follows, we will use the fact that $\phi$ is 1-Lipschitz as in the definition of NLDS, even when we don't explicitly use Assumption 1.

**Proposition 2.** *Under Assumption 4 with $\rho < 1$, the process is exponentially Ergodic and has a stationary distribution $\pi$. Suppose $X \sim \pi$ then $\mathbb{E}\|X\|^4 < \infty$.*

The result follows from a technique similar to the one used for Proposition [49] by considering the process in the space of measures endowed with the Wasserstein metric.

## D.1  Sub-Gaussian Case: Stable Systems

We will first consider the process with $X_0 = 0$ and prove concentration for $X_t$ for arbitrary $t$, and then use distributional convergence results to prove the concentration results at stationarity. First, we prove some preparatory lemmas.

**Lemma 10.** *Suppose $Y$ is a $\nu^2$ sub-Gaussian random variable with zero mean. Then, for any $\lambda \le \frac{1}{4\nu^2}$, we have:*

$$\mathbb{E}\exp(\lambda Y^2) \le 1 + 8\lambda\nu^2.$$

We refer Section H for the proof. Now, consider the random variable $Z_{t+1} = \|X_{t+1}\|^2 - \sum_{s=0}^{t}\rho^{t-s}\|\eta_s\|^2$. By assumption, we have $X_0 = 0$. Therefore we must have $Z_0 = 0$. We have the following lemma:

**Lemma 11.** *Suppose that Assumptions 3 and 5 hold and $\rho$ be as given in Assumption 5. For any $\lambda$ such that $0 \le \lambda \le \frac{1-\rho}{2\rho C_\eta \sigma^2}$, we have:*

$$\mathbb{E}\exp(\lambda Z_{t+1}) \le 1.$$

*Proof.* First by mean value theorem, we must have: $\phi(A^*X_t) = \phi(A^*X_t) - \phi(0) = DA^*X_t$ for some diagonal matrix $D$ with entries lying in $[0,1]$. Therefore, $\|\phi(A^*X_t)\| \le \|D\|\|A^*\|\|X_t\| \le \rho\|X_t\|$. Using this in Equation (1), we conclude:

$$
\begin{aligned}
\|X_{t+1}\|^2 - \|\eta_t\|^2 &= \|\phi(A^*X_t)\|^2 + 2\langle\eta_t, \phi(A^*X_t)\rangle \\
&\le \rho^2\|X_t\|^2 + 2\langle\eta_t, \phi(A^*X_t)\rangle
\end{aligned}
\tag{94}
$$

Let $\mathcal{F}_s = \sigma(X_0, \eta_0, \ldots, \eta_s)$. It is clear that $X_s \in \mathcal{F}_{s-1}$. Using Equation (94), we conclude:

$$\mathbb{E}\left[\exp(\lambda Z_{t+1})\big|\mathcal{F}_{t-1}\right] = \mathbb{E}\left[\exp\left(\lambda\|X_{t+1}\|^2 - \lambda\|\eta_t\|^2\right)\big|\mathcal{F}_{t-1}\right]\exp\left(-\lambda\sum_{s=0}^{t-1}\rho^{t-s}\|\eta_s\|^2\right)$$

$$\leq \mathbb{E}\left[\exp\left(\lambda\rho^2\|X_t\|^2 + 2\lambda\langle\eta_t, \phi(A^*X_t)\rangle\right)\big|\mathcal{F}_{t-1}\right]\exp\left(-\lambda\sum_{s=0}^{t-1}\rho^{t-s}\|\eta_s\|^2\right)$$

$$\leq \exp\left(\lambda\rho^2\|X_t\|^2 + 2\lambda^2 C_\eta\sigma^2\|\phi(A^*X_t)\|^2\right)\exp\left(-\lambda\sum_{s=0}^{t-1}\rho^{t-s}\|\eta_s\|^2\right)$$

$$\leq \exp\left(\lambda\rho^2\|X_t\|^2 + 2\lambda^2\rho^2 C_\eta\sigma^2\|X_t\|^2\right)\exp\left(-\lambda\sum_{s=0}^{t-1}\rho^{t-s}\|\eta_s\|^2\right)$$

$$\leq \exp\left(\lambda\rho\|X_t\|^2\right)\exp\left(-\lambda\sum_{s=0}^{t-1}\rho^{t-s}\|\eta_s\|^2\right)$$

$$= \exp\left(\lambda\rho Z_t\right) \tag{95}$$

In the fourth step, we have used the fact that $\|\phi(A^*X_t)\| \leq \rho\|X_t\|$. In the fifth step we have used the assumption that $\lambda \leq \frac{1-\rho}{2\rho C_\eta\sigma^2}$ to show $\lambda\rho^2 + 2\lambda^2\rho^2 C_\eta\sigma^2 \leq \rho\lambda$. In the last step, we have used the definition of $Z_t$. We iterate over Equation (95) and use the fact that $Z_0 = 0$ almost surely to conclude that whenever $\lambda \leq \frac{1-\rho}{2\rho C_\eta\sigma^2}$, we must have:

$$\mathbb{E}\exp(\lambda Z_{t+1}) \leq \mathbb{E}\exp(\lambda Z_0) = 1\,.$$

$\square$

Now, let $Y_{t+1} = \sum_{s=0}^t \rho^{t-s}\|\eta_t\|^2$. We will now use Lemma 10 to bound $\mathbb{E}\exp(\lambda Y_{t+1})$ for $\lambda > 0$ small enough.

**Lemma 12.** *Suppose that Assumptions 3 and 5 hold and $\rho$ be as given in Assumption 5. For any $\lambda$ such that $0 \leq \lambda \leq \frac{1}{4dC_\eta\sigma^2}$, we have:*

$$\mathbb{E}\exp(\lambda Y_{t+1}) \leq \exp\left(8\frac{\lambda dC_\eta\sigma^2}{1-\rho}\right)$$

*Proof.* Let $N(\beta) := \mathbb{E}\exp(\beta\|\eta_s\|^2)$ By independence of the noise sequence, we have:

$$\mathbb{E}\exp(\lambda Y_{t+1}) = \prod_{s=0}^t N(\rho^{t-s}\lambda) \tag{96}$$

For $\beta \leq \frac{1}{4dC_\eta\sigma^2}$

$$N(\beta) = \mathbb{E}\exp(\beta\|\eta_s\|^2) = \mathbb{E}\exp(\beta\sum_{i=1}^d\langle e_i, \eta_s\rangle^2)$$

$$\leq \frac{1}{d}\sum_{i=1}^d\mathbb{E}\exp(\beta d\langle e_i, \eta_s\rangle^2) \leq 1 + 8\beta dC_\eta\sigma^2 \tag{97}$$

In the last step, we have used Jensen's inequality for the function $x \to \exp(x)$ and then invoked the Lemma 10. Plugging this into Equation (96), we conclude:

$$\mathbb{E}\exp(\lambda Y_{t+1}) \leq \prod_{s=0}^t\left(1 + 8\lambda d\rho^{t-s}dC_\eta\sigma^2\right) \leq \exp(\sum_{s=0}^t 8\lambda d\rho^{t-s}C_\eta\sigma^2)$$

$$\leq \exp\left(8\frac{\lambda dC_\eta\sigma^2}{1-\rho}\right) \tag{98}$$

$\square$

Based on Lemmas 12 and 11, we will now state the following concentration inequality:

**Theorem 8.** *Suppose Assumptions 3 and 5 hold and $\rho$ be as given in Assumption 5. Let $X$ be distributed according $\pi$, the stationary distribution of $\mathsf{NLDS}(A^*, \mu, \phi)$. Then, for any $0 < \lambda \leq \lambda^* := \min(\frac{1}{8dC_\eta\sigma^2}, \frac{1-\rho}{4\rho C_\eta\sigma^2})$, we have:*

$$\mathbb{E}\exp\left(\lambda\|X\|^2\right) \leq \exp(\tfrac{8\lambda dC_\eta\sigma^2}{1-\rho}).$$

*We conclude:*

1. *Applying Chernoff bound with $\lambda = \lambda^*$, we conclude:*

$$\mathbb{P}\left(\|X\|^2 > \frac{8dC_\eta\sigma^2}{1-\rho} + \beta\right) \leq \exp(-\lambda^*\beta).$$

2.

$$\mathbb{E}\|X\|^2 \leq \frac{8dC_\eta\sigma^2}{1-\rho}$$

*The conclusions still hold when $X$ is replaced by $X_t$ for any $t \in \mathbb{N}$ for the process started at $0$.*

*Proof.* We first note that $\|X_{t+1}\|^2 = Z_{t+1} + Y_{t+1}$. Therefore, by Cauchy-Schwarz inequality, we must have:

$$\begin{aligned}
\mathbb{E}\exp\left(\lambda\|X_{t+1}\|^2\right) &= \mathbb{E}\exp\left(\lambda(Z_{t+1} + Y_{t+1})\right) \\
&\leq \sqrt{\mathbb{E}\exp(2\lambda Z_{t+1})}\sqrt{\mathbb{E}\exp(2\lambda Y_{t+1})} \\
&\leq \exp(\tfrac{8\lambda dC_\eta\sigma^2}{1-\rho}) \qquad\qquad (99)
\end{aligned}$$

Here we have used Lemmas 12 and 11 and the appropriate bounds on $\lambda$. Recall that we started the chain $(X_t)$ with $X_0 = 0$. Denote the law of $X_t$ by $\pi_t$. By proposition 2, we show that $\pi_t$ converges weakly to the stationary distribution $\pi$. We invoke Skhorokhod representation theorem to show that there exist random variables $\bar{X}_t \sim \pi_t$ and $X \sim \pi$ for $t \in \mathbb{N}$, defined on a common probability space such that $\bar{X}_t \to X$ almost surely. Now, we have shown that:

$$\mathbb{E}\exp\left(\lambda\|\bar{X}_{t+1}\|^2\right) \leq \exp(\tfrac{8\lambda dC_\eta\sigma^2}{1-\rho}).$$

Now, applying Fatou's Lemma to the equation above as $t \to \infty$, we conclude:

$$\mathbb{E}\exp\left(\lambda\|X\|^2\right) \leq \exp(\tfrac{8\lambda dC_\eta\sigma^2}{1-\rho}). \qquad\qquad (100)$$

The concentration inequality follows from an application of Chernoff bound and the second moment bound follows from Jensen's inequality to Equation (100) (i.e, $\mathbb{E}\exp(Y) \geq \exp(\mathbb{E}Y)$). $\qquad\square$

### D.2 Sub-Gaussian Case: Possibly Unstable Systems

We consider the case with $(C_\rho, \rho)$ regularity, but we allow $\rho > 1$.

**Lemma 13.** *Under Assumption 4, we have:*

$$\|X_t\| \leq C_\rho \sum_{s=0}^{t-1} \rho^{t-s-1}\|\eta_s\|. \qquad\qquad (101)$$

*No suppose Assumption 3 also holds. Let $\delta \in (0, 1/2)$. Then with probability atleast $1 - \delta$, we must have:*

$$\sup_{0 \leq t \leq T} \|X_t\| \leq CC_\rho\sqrt{C_\eta}S(\rho, T)\sigma\sqrt{d\log(\tfrac{T}{\delta})}.$$

*Where $S(\rho, T) := \sum_{t=0}^{T-1} \rho^{T-t-1}$ and $C$ is some universal constant.*

*Proof.* We consider the notations established in Assumption 4. We will define the process $X_t^{(s)}$ by $X_0^{(s)} = \cdots = X_s^{(s)} = 0$ and $X_{t+1}^{(s)} = \phi(A^* X_t^{(s)}) + \eta_t$ for $t \geq s$, where $\eta_t$ is the same noise sequence driving the process $X_0, X_1, \ldots, X_T$. Note that $X_t^{(s)} = h_{t-s}(0, \eta_s, \eta_{s+1}, \ldots, \eta_{t-1})$.

$$X_t - 0 = X_t - X_t^{(1)} + X_t^{(1)} - X_t^{(2)} + \cdots + X_t^{(t)} - 0$$

$$\implies \|X_t\| \leq \sum_{s=0}^{t-1} \|X_t^{(s)} - X_t^{(s+1)}\|$$

$$= \sum_{s=0}^{t-1} \|h_{t-s-1}(\eta_s, \ldots, \eta_{t-1}) - h_{t-s-1}(0, \eta_{s+1}, \ldots, \eta_{t-1})\|$$

$$\leq \sum_{s=0}^{t-1} C_\rho \rho^{t-s-1} \|\eta_s\| \tag{102}$$

In the last step, we have used Assumption 4. To prove the high probability bound, we note that $\mathbb{P}(\sup_{0 \leq s \leq T-1} \leq \|\eta_s\| > C\sqrt{C_\eta}\sigma \log(\frac{T}{\delta})) \leq \delta$ for some universal constant $C$. $\qquad \square$

### D.3 Heavy Tailed Case: Stable Systems

**Theorem 9.** *Suppose Assumption 4 holds with $\rho < 1$. Suppose that $X$ is distributed as the stationary distribution $\pi$ of the system* $\mathsf{NLDS}(A^*, \mu, \phi)$. *Then, we have:*

1.

$$\mathbb{E}\|X\|^4 \leq \frac{C_\rho^4 M_4}{(1-\rho)^4}.$$

*Where we recall $M_4 = \mathbb{E}\|\eta_t\|^4$.*

2.

$$\mathbb{E}\|X\|^2 \leq \frac{C_\rho^2 d\sigma^2}{(1-\rho)^2}.$$

*Proof.* We use Equation (101) to conclude the desired bound for $X_T$ when the process is started with $X_0 = 0$. We then use Fatou's lemma along with Skhorokhod Representation theorem like in Theorem 8 to conclude the bound at stationarity. $\qquad \square$

## E  Well Conditioned Second Moment Matrices

In this section we will consider a stationary sequence $X_0, \ldots, X_T$ derived from the process $\mathsf{NLDS}(A^*, \mu, \phi)$, with the corresponding noise sequence $\eta_0, \ldots, \eta_T$. We want to show that the matrix $\frac{1}{B}\sum_{t=0}^{B-1} X_t X_t^\top$ behaves similar to $G := \mathbb{E}X_t X_t^\top$. To do this, we will first to control the quantity: $\mathbb{E}\langle X_t, x\rangle^2 \langle X_s, x\rangle^2$ for arbitrary fixed vector $x \in \mathbb{R}^d$. Clearly, $\mathbb{E}\langle X_t, x\rangle^2 = x^\top Gx$.

**Lemma 14.** *Without loss of generality, we suppose that $t > s$. Suppose $X_0, \ldots, X_T$ be a stationary sequence from* $\mathsf{NLDS}(A^*, \mu, \phi)$.

1. *Suppose Assumptions 3 and 5 hold. Then we have:*

$$\mathbb{E}\langle X_t, x\rangle^2 \langle X_s, x\rangle^2 \leq 2(x^\top Gx)^2 + \bar{C}_1 \rho^{2(t-s)} \frac{d\sigma^2}{1-\rho} x^\top Gx \log\left(\frac{d}{1-\rho}\right)$$

   *Where $\bar{C}_1$ depends only on $C_\eta$.*

2. *Suppose Assumption 4 holds with $\rho < 1$. Recall that $M_4 = \mathbb{E}\|\eta_t\|^2$. We have:*

$$\mathbb{E}\langle X_t, x\rangle^2 \langle X_s, x\rangle^2 \leq 2(x^\top Gx)^2 + 8\|x\|^2 C_\rho^6 \rho^{2(t-s)} \frac{M_4}{(1-\rho)^4}$$

We will give the proof of this lemma in Section H. Now, consider the random matrix $\hat{G}_B :=$ $\frac{1}{B}\sum_{t=0}^{B-1} X_t X_t^\top$. Clearly, $\hat{G}_B \succeq 0$ and because of stationarity, $\mathbb{E}\hat{G}_B = G$. We write down the following lemma:

**Lemma 15.** *Suppose $X_0, \dots, X_T$ be a stationary sequence from $\mathsf{NLDS}(A^*, \mu, \phi)$.*

1. *Suppose Assumptions 3 and 5 hold. Let $\bar{C}_1$ be as in Lemma 14. Suppose $B \geq \bar{C}_1 \frac{d}{(1-\rho)(1-\rho^2)} \log\left(\frac{d}{1-\rho}\right)$. Then, for any fixed vector $x \in \mathbb{R}^d$,*

$$\mathbb{P}\left(x^\top \hat{G}_B x \geq \frac{1}{2} x^\top G x\right) \geq p_0 > 0.$$

*Where $p_0$ is a universal constant which can be taken to be $\frac{1}{16}$. Furthermore, for any event $\mathcal{A}$ such that $\mathbb{P}(\mathcal{A}) > 1 - p_0$, we must have:*

$$\mathbb{P}\left(x^\top \hat{G}_B x \geq \frac{1}{2} x^\top G x \Big| \mathcal{A}\right) \geq q_0 := \frac{p_0 - \mathbb{P}(\mathcal{A}^c)}{\mathbb{P}(\mathcal{A})} > 0.$$

2. *Suppose Assumption 4 holds with $\rho < 1$. Whenever $B \geq \frac{8 C_\rho^6 M_4}{(1-\rho)^4(1-\rho^2)\sigma^4}$*

$$\mathbb{P}\left(x^\top \hat{G}_B x \geq \frac{1}{2} x^\top G x\right) \geq p_0 > 0.$$

*Proof.* Without loss of generality, take $\|x\| = 1$. We start with the Paley-Zygmund inequality. Let $Z$ be any random variable such that $Z \geq 0$ almost surely and $\mathbb{E}Z^2 < \infty$. For any $\theta \in [0, 1]$ we must have:

$$\mathbb{P}(Z \geq \theta \mathbb{E}Z) \geq (1 - \theta)^2 \frac{(\mathbb{E}Z)^2}{\mathbb{E}Z^2}.$$

Now consider $Z = x^\top \hat{G}_B x$ and $\theta = \frac{1}{2}$.

1. The simple calculation shows that:

$$\begin{aligned}
\mathbb{P}\left(x^\top \hat{G}_B x \geq \frac{1}{2} x^\top G x\right) &\geq \frac{1}{4} \frac{B^2 (x^\top G x)^2}{\sum_{s,t=0}^{B-1} \mathbb{E}\langle X_t, x\rangle^2 \langle X_s, x\rangle^2} \\
&\geq \frac{1}{4} \frac{B^2 (x^\top G x)^2}{2B^2 (x^\top G x)^2 + \sum_{s,t=0}^{B-1} \bar{C}_1 \rho^{2|t-s|} \frac{d\sigma^2}{1-\rho} x^\top G x \log\left(\frac{d}{1-\rho}\right)} \\
&\geq \frac{1}{4} \frac{B^2 (x^\top G x)^2}{2B^2 (x^\top G x)^2 + 2\sum_{t=0}^{B-1} \bar{C}_1 \frac{d\sigma^2}{(1-\rho)(1-\rho^2)} x^\top G x \log\left(\frac{d}{1-\rho}\right)} \\
&= \frac{1}{4} \frac{B^2 (x^\top G x)^2}{2B^2 (x^\top G x)^2 + 2B\bar{C}_1 \frac{d\sigma^2}{(1-\rho)(1-\rho^2)} x^\top G x \log\left(\frac{d}{1-\rho}\right)} \\
&= \frac{1}{8} \frac{1}{1 + \tau_B} \tag{103}
\end{aligned}$$

Here, $\tau_B := \frac{\bar{C}_1}{x^\top G x} \frac{d\sigma^2}{B(1-\rho)(1-\rho^2)} \log\left(\frac{d}{1-\rho}\right)$. In the second step we have used item 1 of Lemma 14. In the third step, we have summed the infinite series $\sum_{s \geq t} \rho^{2(t-s)}$. Using the hypothesis that $B \geq \bar{C}_1 \frac{d}{(1-\rho)(1-\rho^2)} \log\left(\frac{d}{1-\rho}\right)$ and $G \succeq \sigma^2 I$, we conclude the result.

2. We proceed similarly as above, but use item 2 in Lemma 14 instead.

$\square$

We will now follow the method used to prove [19, Lemma 31]. We now consider the matrix $\tilde{H}^s_{0,B-1}$ under the event $\tilde{\mathcal{D}}^s_{-0}$ in order to prove Theorem 10, where the terms are as defined in Section C.3. For the sake of clarity, we will drop the superscript $s$.

**Remark 5.** *We prove the results below for $\tilde{H}^s_{0,B-1}$ but they hold unchanged when the matrices are all replaced with $\tilde{H}^{s,\top}_{0,B-1}$ given that we reverse the order of taking products whenever they are encountered.*

**Lemma 16.** *Suppose Assumption 1 holds. Suppose that $\gamma RB < \frac{1}{4}$. Then, for any buffer $s$, under the event $\tilde{\mathcal{D}}^s_{-0}$, we have:*

$$I - 4\gamma\left(1 + \frac{2\gamma BR}{1-4\gamma BR}\right)\sum_{i=0}^{B-1}\tilde{X}^s_{-i}\tilde{X}^{s,\top}_{-i} \preceq \tilde{H}^s_{0,B-1}\tilde{H}^{s,\top}_{0,B-1} \preceq I - 4\gamma\left(\zeta - \frac{2\gamma BR}{1-4\gamma BR}\right)\sum_{i=0}^{B-1}\tilde{X}^s_{-i}\tilde{X}^{s,\top}_{-i}$$

*In particular, whenever we have $\gamma BR \leq \frac{\zeta}{4(1+\zeta)}$, we must have:*

$$I - 4\gamma\left(1 + \frac{\zeta}{2}\right)\sum_{i=0}^{B-1}\tilde{X}^s_{-i}\tilde{X}^{s,\top}_{-i} \preceq \tilde{H}^s_{0,B-1}\tilde{H}^{s,\top}_{0,B-1} \preceq I - 2\gamma\zeta\sum_{i=0}^{B-1}\tilde{X}^s_{-i}\tilde{X}^{s,\top}_{-i}$$

*Proof.* The proof follows from the proof of [19, Lemma 28] with minor modifications to account for the fact that $\phi'(\beta) \in [\zeta, 1]$. $\qquad\square$

Combining Lemma 16 with Lemma 14 we will show that $\tilde{H}^s_{0,B-1}$ contracts any given vector with probability at-least $p_0 > 0$.

**Theorem 10.** *Suppose Assumptions 1, 3 and 5 hold. Assume that $B$ and $\gamma$ are such that: $B \geq \bar{C}_1 \frac{d}{(1-\rho)(1-\rho^2)}\log\left(\frac{d}{1-\rho}\right)$ and $\gamma BR \leq \frac{\zeta}{4(1+\zeta)}$ where $\bar{C}_1$ is as given in Lemma 15. We also assume that $\mathbb{P}(\hat{\mathcal{D}}^{b,a}) > \max(\frac{1}{2}, 1 - \frac{p_0}{2})$, where $p_0$ is as given in Lemma 15. Let $a \geq b$. Let $\lambda_{\min}(G)$ denote the smallest eigenvalue of $G$. Conditioned on the event $\tilde{\mathcal{D}}^{b,a}$,*

*(1) $\|\prod_{s=a}^{b}\tilde{H}^{s,\top}_{0,B-1}\| \leq 1$ almost surely*

*(2) Whenever $b - a + 1$ is larger than some universal constant $C_0$,*

$$\mathbb{P}\left(\|\prod_{s=a}^{b}\tilde{H}^{s,\top}_{0,B-1}\| \geq 2(1 - \zeta\gamma B\lambda_{\min}(G))^{c_4(a-b+1)}\Big|\tilde{\mathcal{D}}^{b,a}\right) \leq \exp(-c_3(a-b+1)+c_5d)$$

*Where $c_3, c_4$ and $c_5$ are universal constants.*

*Proof.* The proof of (1) above follows from an application of Lemma 16. So we will just prove (2). We will prove this with an $\epsilon$ net argument over the unit $\ell^2$ sphere in $\mathbb{R}^d$.

Suppose we have arbitrary $x \in \mathbb{R}^d$ such that $\|x\| = 1$. Let $K_v := \prod_{s=v}^{b}\tilde{H}^{s,\top}_{0,B-1}$. When $v \leq b$, we take this product to be identity. Now, define $\hat{G}^v_B := \frac{1}{B}\sum_{j=0}^{B-1}X^v_j X^{v,\top}_j$

Consider the class of events indexed by $v$: $\mathcal{G}_v := \{\|\tilde{H}^{v,\top}_{0,B-1}K_{v-1}x\|^2 \leq \|K_{v-1}x\|^2(1 - \gamma\zeta B\lambda_{\min}(G))\}$. From Lemma 15, we will prove the following claim:

**Claim 6.** *Whenever $v \in [b, a] \cap \mathbb{Z}$:*

$$\mathbb{P}(\mathcal{G}^c_v|\tilde{\mathcal{D}}^{b,a}, \tilde{H}^{s,\top}_{0,B-1} : s < v) \leq 1 - q_0 \qquad (104)$$

*Where $q_0 > 0$ is as given in Lemma 15 and can be taken to be a universal constant under the present hypotheses.*

*Proof.* We will denote $K_{v-1}x$ by $x_v$ for the sake of convenience. We note that when conditioned on $\tilde{H}_{0,B-1}^{s,\top}$ for $s < v$, $x_v$ is fixed. Using Lemma 16, we note that:

$$\mathbb{P}(\mathcal{G}_v^c | \tilde{\mathcal{D}}^{b,a}, \tilde{H}_{0,B-1}^{s,\top} : s < v) \leq \mathbb{P}(x_v^\top \hat{G}_B^v x_v < \tfrac{1}{2} x_v^\top G x_v | \tilde{\mathcal{D}}^{b,a}, \tilde{H}_{0,B-1}^{s,\top} : s < v)$$

We note that $\hat{G}_B^v$ is independent of $\tilde{H}_{0,B-1}^{s,\top}$ for $s \leq v$ (eventhough $\tilde{H}_{0,B-1}^v$ is not necessarily). Now we also note that $\hat{G}_B^v$ is independent of $\tilde{\mathcal{D}}^s$ for $s \neq v$. Therefore, we can apply Lemma 15 to conclude the claim. □

Let $D \subseteq \{b, \ldots, a\}$ such that $|D| = r$. It is also clear from item 1 and the definitions above that whenever the event $\cap_{v \in D} \mathcal{G}_v$ holds, we have:

$$\| \prod_{s=a}^{b} \tilde{H}_{0,B-1}^{s,\top} x \| \leq (1 - \gamma B \lambda_{\min}(G))^{\frac{r}{2}} . \tag{105}$$

Therefore, whenever Equation (105) is violated, we must have a set $D^c \subseteq \{b, \ldots, a\}$ such that $|D^c| \geq b - a - r$ and the event $\cap_{v \in D^c} \mathcal{G}_v^c$ holds. We will union bound all such events indexed by $D^c$ to obtain an upper bound on the probability that Equation (105) is violated. Therefore, using Equation (104) along with the union bound, we have:

$$\mathbb{P}\left( \| \prod_{s=a}^{b} \tilde{H}_{0,B-1}^{s,\top} x \| \geq (1 - \gamma B \lambda_{\min}(G))^{\frac{r}{2}} \bigg| \tilde{\mathcal{D}}^{b,a} \right) \leq \binom{a-b+1}{a-b-r}(1 - q_0)^{a-b-r}$$

Whenever $a - b + 1$ is larger than some universal constant, we can pick $r = c_2(b - a + 1)$ for some constant $c_2 > 0$ small enough such that:

$$\mathbb{P}\left( \| \prod_{s=a}^{b} \tilde{H}_{0,B-1}^{s,\top} x \| \geq (1 - \gamma B \lambda_{\min}(G))^{\frac{r}{2}} \bigg| \tilde{\mathcal{D}}^{b,a} \right) \leq \exp(-c_3(b - a + 1))$$

Now, let $\mathcal{N}$ be a $1/2$-net of the sphere $\mathcal{S}^{d-1}$. Using Corollary 4.2.13 in [50], we can choose $|\mathcal{N}| \leq 6^d$. By Lemma 4.4.1 in [50] we show that:

$$\| \prod_{s=a}^{b} \tilde{H}_{0,B-1}^{s,\top} \| \leq 2 \sup_{x \in \mathcal{N}} \| \prod_{s=a}^{b} \tilde{H}_{0,B-1}^{s,\top} x \| \tag{106}$$

By union bounding Equation (106) for every $x \in \mathcal{N}$, we conclude that:

$$\mathbb{P}\left( \| \prod_{s=a}^{b} \tilde{H}_{0,B-1}^{s,\top} \| \geq 2(1 - \zeta \gamma B \lambda_{\min}(G))^{c_4(b-a+1)} \bigg| \tilde{\mathcal{D}}^{b,a} \right) \leq |\mathcal{N}| \exp(-c_3(a - b + 1))$$
$$= \exp(-c_3(a - b + 1) + c_5 d) \tag{107}$$

□

We will now state the equivalent of [19, Lemma 32]. The proof proceeds similarly, but using Theorem 10 instead. Consider the following operator:

$$Fa, N := \sum_{t=a}^{N-1} \prod_{s=t}^{a+1} \tilde{H}_{0,B-1}^{s,\top} \tag{108}$$

Here we choose the convention that whenever $s > t$, then in any product involving $\tilde{H}_{0,B-1}^{s,\top}$ and $\tilde{H}_{0,B-1}^{t,\top}$, $s$ appears to the right of $t$. Hence, we use the take $\prod_{s=a}^{a+1} \tilde{H}_{0,B-1}^{s,\top} = I$

**Theorem 11.** *Suppose all the conditions in Theorem 10 hold. Then, for any $\delta \in (0,1)$, we have:*

$$\mathbb{P}\left( \|F_{a,N}\| \geq C\left( d + \log \frac{N}{\delta} + \frac{1}{\zeta \gamma B \lambda_{\min}(G)} \right) \bigg| \tilde{\mathcal{D}}^{a,N} \right) \leq \delta$$

*Where $C$ is a universal constant.*

## F    Self Normalized Noise Concentration

We recall the events defined in Section B :

1. $\mathcal{D}_T(R) := \{\sup_{0 \le t \le T} \|X_t\|^2 \le R\}$
2. $\mathcal{E}_T(\kappa) := \{\hat{G} \succeq \frac{\sigma^2 I}{\kappa}\}$
3. $\mathcal{D}_T(R, \kappa) := \mathcal{D}_T(R) \cap \mathcal{E}_T(\kappa)$

From Lemma 13, we conclude that taking $R \ge C_\rho^2 C_\eta (S(\rho, T))^2 \sigma^2 \log(\frac{2T}{\delta})$ ensures that $\mathbb{P}(\mathcal{D}_T(R)) \ge 1 - \frac{\delta}{2}$. Only in this section, we define the following:

1. $\bar{X}_t := \phi(A^* X_t)$
2. $\bar{K}_X := \frac{1}{T} \sum_{t=0}^{T-2} \bar{X}_t \eta_t^\top + \eta_t \bar{X}_t^\top$
3. $\bar{G} := \frac{1}{T} \sum_{t=0}^{T-2} \bar{X}_t \bar{X}_t^\top$
4. $\bar{K}_\eta := \frac{1}{T} \sum_{t=0}^{T-2} \eta_t \eta_t^\top$

**Lemma 17.** *Let $\delta \in (0, \frac{1}{2})$ Take $R = C_\rho^2 C_\eta (S(\rho, T))^2 d\sigma^2 \log(\frac{2T}{\delta})$, $\kappa = 2$ and suppose $T \ge \bar{C}_3 \left( d \log\left(\frac{R}{\sigma^2}\right) + \log \frac{1}{\delta} \right)$ for some constant $\bar{C}_3$ depending only on $C_\eta$. Then, we have:*

$$\mathbb{P}(\mathcal{D}_T(R, \kappa)) \ge 1 - \delta$$

*Proof.* Consider $\hat{G} = \frac{1}{T} \sum_{t=0}^{T-1} X_t X_t^\top = \frac{1}{T} \sum_{t=0}^{T-2} \bar{X}_t \bar{X}_t^\top + \bar{X}_t \eta_t^\top + \eta_t \bar{X}_t^\top + \eta_t \eta_t^\top$. For this proof only, we will define, To show the result, we will prove that $\bar{K}_X$ is not too negative with high probability and that $\bar{K}_\eta$ dominates identity with high probability. Let $x \in \mathcal{S}^{d-1}$ and $\lambda \in \mathbb{R}$ Note that due to the sub-Gaussianity of $\eta_t$ and the definition of the process,

$$M_s := \exp\left( \sum_{s=0}^{t} \lambda \langle x, \eta_s \rangle \langle x, \bar{X}_s \rangle - \frac{C_\eta \sigma^2 \lambda^2}{2} \langle \bar{X}_s, x \rangle^2 \right).$$

is a super martingale with respect to the filtration $\mathcal{F}_t := \sigma(X_0, \eta_0, \ldots, \eta_t)$, we conclude that $\mathbb{E} M_{T-1} \le 1$. An application of Chernoff bound shows that for every $\lambda, \beta > 0$, we must have:

$$\mathbb{P}\left( |\langle x, \bar{K}_X x \rangle| \ge 2C_\eta \sigma^2 \lambda x^\top \bar{G} x + \frac{\beta}{T} \middle| \mathcal{D}_T(R) \right) \le \frac{2}{1-\delta} \exp(-\lambda\beta) \tag{109}$$

We will now invoke Theorem 5.39 in [51] to conclude that for some constant $\bar{C}_2$ which depends only on $C_\eta$:

$$\mathbb{P}\left( \bar{K}_\eta \preceq \left( 1 - \bar{C}_2 \left( \sqrt{\frac{d}{T}} + \sqrt{\frac{\log \frac{1}{\delta}}{T}} \right) \right) \sigma^2 I \middle| \mathcal{D}_T(R) \right) \le \frac{\delta}{4} \tag{110}$$

Consider any $\epsilon$ net $\mathcal{N}_\epsilon$ over $\mathcal{S}^{d-1}$. By Corollary 4.2.13 in [51], we can take $|\mathcal{N}_\epsilon| \le (1 + \frac{2}{\epsilon})^d$. From Equations (109) and (110), we conclude that conditioned on $\mathcal{D}_T(R)$, with probability at-least $1 - \frac{\delta}{4} - |\mathcal{N}_\epsilon| \frac{\exp(-\lambda\beta)}{1-\delta}$ we have:

$$
\begin{aligned}
\inf_{x \in \mathcal{S}^{d-1}} x^\top \hat{G} x &\ge \inf_{y \in \mathcal{N}_\epsilon} y^\top \hat{G} y - 2\|\hat{G}\|\epsilon \\
&\ge \inf_{y \in \mathcal{N}_\epsilon} y^\top \bar{G} y - |y^\top \bar{K}_X y| + y^\top \bar{K}_\eta y - 2\|\hat{G}\|\epsilon \\
&\ge \inf_{y \in \mathcal{N}_\epsilon} y^\top \bar{G} y - |y^\top \bar{K}_X y| + y^\top \bar{K}_\eta y - 2R\epsilon \\
&\ge \inf_{y \in \mathcal{N}_\epsilon} y^\top \bar{G} y (1 - 2\lambda\sigma^2 C_\eta) - \frac{\beta}{T} + \sigma^2 \left( 1 - \bar{C}_2 \left( \sqrt{\frac{d}{T}} + \sqrt{\frac{\log \frac{1}{\delta}}{T}} \right) \right) - 2R\epsilon
\end{aligned}
\tag{111}
$$

In the third step, we have used the fact that under the event $\mathcal{D}_T(R)$, $\|\hat{G}\| \leq R$. Take $\lambda = \frac{1}{2\sigma^2 C_\eta}$ and $\epsilon = \frac{1}{8R\sigma^2}$ and $\beta = 2\sigma^2 dC_\eta \log(16\frac{R}{\sigma^2} + 1) + 2\sigma^2 C_\eta \log\frac{8}{\delta}$. We conclude that whenever $T \geq \bar{C}_3 \left(d\log\left(\frac{R}{\sigma^2}\right) + \log\frac{1}{\delta}\right)$ for some constant $\bar{C}_3$ depending only on $C_\eta$, with probability at-least $1 - \frac{\delta}{2}$ conditioned on $\mathcal{D}_T(R)$, we have: $\hat{G} \succeq \frac{\sigma^2}{2}I$. In the definition of $\mathcal{E}_T(\kappa)$, we take $\kappa = 2$. Therefore, we must have:

$$\mathbb{P}(\mathcal{E}_T(\kappa) \cap \mathcal{D}_T(R)) = \mathbb{P}(\mathcal{E}_T(\kappa)|\mathcal{D}_T(R))\mathbb{P}(\mathcal{D}_T(R)) \geq (1 - \tfrac{\delta}{2})^2 \geq 1 - \delta\,.$$

We conclude the result from the equation above. $\qquad\square$

We now give the proof of Lemma 1.

### F.1 Proof of Lemma 1

*Proof.* We invoke Theorem 1 in [20] with $S_t = T\hat{N}_i$, $V = T\sigma^2 I$, $\bar{V}_t = V + T\hat{G}$. We know that $\langle\eta, e_i\rangle$ is $C_\eta\sigma^2$ sub-Gaussian. So, we take '$R$' in the reference to be $C_\eta\sigma^2$. Therefore, we conclude that with probability at least $1 - \delta$:

$$\hat{N}_i^\top \bar{V}_t^{-1} \hat{N}_i \leq \frac{2C_\eta\sigma^2}{T^2} \log\left(\frac{\det(\bar{V}_t)^{1/2} \det(V)^{-1/2}}{\delta}\right)\,. \tag{112}$$

Under the event $\mathcal{D}_T(R, \kappa)$, we must have: $\bar{V}_t \preceq \sigma^2 TI + TRI$. This implies:

$$\det(\bar{V}_t)^{1/2} \det(V)^{-1/2} \leq (1 + \tfrac{R}{\sigma^2})^{\frac{d}{2}} \tag{113}$$

Now, observe that under the event $\mathcal{D}_T(R, \kappa)$, $\hat{G} \succeq \frac{\sigma^2 I}{2}$. Therefore, $\bar{V}_t \preceq 3T\hat{G}$. This implies:

$$\frac{1}{3T} \hat{N}_i^\top \hat{G}^{-1} \hat{N}_i \leq \hat{N}_i^\top \bar{V}_t^{-1} \hat{N}_i \tag{114}$$

Combining Equations (112), (113) and (114) and using Lemma 17, we conclude that with probability at-least $1 - 2\delta$, we have:

$$\hat{N}_i^\top \hat{G}^{-1} \hat{N}_i \leq \frac{6C_\eta\sigma^2}{T} \left[d\log(1 + \tfrac{R}{\sigma^2}) + \log\tfrac{1}{\delta}\right]$$

Using union bound, we conclude the result. $\qquad\square$

## G  Parameter Recovery Lower Bounds for ReLU-AR Model

We will show that in the case of non-expansive activation functions, parameter recovery can be hard information theoretically. More specifically, even when $\phi = \mathsf{ReLU}$ and $\eta_t \sim \mathcal{N}(0, I)$, and $\|A^*\|_2 = \rho < 1$, we will need exponentially many samples with respect to $d$ in order restimate $A^*$ upto any vanishing accuracy. We do this via the two point method. Henceforth, we will assume that $d \geq 2$. Given $\epsilon > 0$, define with matrix $A(\epsilon) \in \mathbb{R}^{d \times d}$ as:

$$A_{ij}(\epsilon) = \begin{cases} \frac{1}{4} & \text{if } i = j; i \leq d-1 \\ 0 & \text{if } i \neq j; i \leq d-1 \\ -\frac{\epsilon}{\sqrt{d-1}} & \text{if } i \neq j; i = d \\ 0 & \text{if } i = j = d \end{cases} \tag{115}$$

We will consider $\mathsf{NLDS}(A(\epsilon), \mathcal{N}(0, I), \mathsf{ReLU})$ and $\mathsf{NLDS}(A(0), \mathcal{N}(0, I), \mathsf{ReLU})$ as the two points in the two point method. As usual, we will consider $X_0 = 0$ almost surely for the sake of convenience. But it will be shown that this process is rapidly mixing since we intend to pick $\epsilon$ such that $\|A(\epsilon)\| < \frac{1}{2}$, so all the results should easily extend to stationary sequences. We collect some useful results in the following lemma:

**Lemma 18.**     *1. $\|A(\epsilon)\| = \sqrt{\frac{1}{16} + \epsilon^2}$*

2. *Suppose $\epsilon$ is small enough such that $\|A(\epsilon)\| < 1$. Let $X_0, X_1, \ldots, X_T \sim$ NLDS$(A(\epsilon), \mathcal{N}(0, I), \mathsf{ReLU})$ with $X_0 = 0$. For some universal constant $p_1 > 0$, we have for every $i \le d$, $t \ge 0$:*

$$\mathbb{P}\left(\langle X_{t+1}, e_i \rangle \ge \frac{1}{2} \Big| X_t\right) \ge p_1$$

*Proof.*   1. Proof follows from elementary calculations.

2. $X_{t+1} = \mathsf{ReLU}(A(\epsilon)X_t) + \eta_t$. Therefore, for every $i \le d - 1$, must have: $\langle X_{t+1}, e_i \rangle \ge \langle \eta_t, e_i \rangle$. Therefore, we conclude that

$$\mathbb{P}\left(\langle X_{t+1}, e_i \rangle \ge \frac{1}{2} \Big| X_t\right) \ge \mathbb{P}\left(\langle \eta_t, e_i \rangle \ge \tfrac{1}{2} \Big| X_t\right) \ge p_1 > 0.$$

$\square$

We will now show that the last co-ordinate $\langle X_{t+1}, e_d \rangle$ is just noise with a large probability and hence, we cannot estimate the last row of $A(\epsilon)$ even with a large number of samples. Note that the event $\langle X_{t+1}, e_d \rangle \ne \langle \eta_t, e_d \rangle$ is that same as the event $\langle a_d(\epsilon), X_t \rangle > 0$. Let $a_d(\epsilon)$ denote the last row of $A(\epsilon)$, in the form of a column vector.

**Lemma 19.** *Suppose $t \ge 2$. Then, for some universal constants $C_0, C_1 > 0$,*

$$\mathbb{P}\left(\langle a_d(\epsilon), X_t \rangle > 0\right) \le C_0 \exp(-C_1 d).$$

*Proof.* $\langle a_d, X_t \rangle > 0$ if and only if $\langle a_d, \mathsf{ReLU}(A(\epsilon)X_{t-1})\rangle + \langle a_d, \eta_{t-1} \rangle > 0$. We first note that the first $d - 1$ rows of $X_{t-1}$ are i.i.d. by the definition of $A(\epsilon)$. Therefore, we conclude using Lemma 18 that:

$$\mathbb{P}\left(\langle a_d, \mathsf{ReLU}(A(\epsilon)X_{t-1})\rangle \ge -c_0\epsilon\sqrt{d}\right) \le \exp(-c_1 d).$$

for some universal constants $c_0, c_1$. Now, $\langle a_d, \eta_{t-1} \rangle$ is distributed as $\mathcal{N}(0, \epsilon^2)$ and is independent of $X_{t-1}$. Therefore, with a large probability, $\langle a_d, \mathsf{ReLU}(A(\epsilon)X_{t-1})\rangle$ takes a large negative value and by Gaussian concentration, $\langle a_d, \eta_{t-1} \rangle$ concentrates near $0$. Therefore, using elementary calculations we conclude the result. $\square$

In what follows, by $\mathbf{X}_T^\epsilon := (X_0^\epsilon, \ldots, X_T^\epsilon)$ we denote the process such that $X_0^\epsilon = 0$ and $(X_0^\epsilon, \ldots, X_T^\epsilon) \sim \mathsf{NLDS}(A(\epsilon), \mathcal{N}(0, I), \mathsf{ReLU})$.

**Lemma 20.** *When $\epsilon < \frac{1}{4}$, for some universal constants $c_0, c_1 > 0$, we have:*

$$\mathsf{TV}(\mathbf{X}_T^\epsilon, \mathbf{X}_T^0) \le c_0\left(\epsilon + \epsilon\sqrt{T}\exp(-c_1 d)\right).$$

*Proof.* Following the proof of Lemma 36 in [19], we conclude that:

$$\mathsf{KL}(\mathbf{X}_T^\epsilon \| \mathbf{X}_T^0) = \frac{1}{2}\sum_{t=0}^{T-1}\mathbb{E}\|\mathsf{ReLU}(A(\epsilon)X_t^\epsilon) - \mathsf{ReLU}(A(0)X_t^\epsilon)\|^2 \qquad (116)$$

Note here that the expectation is with respect to the randomness in the trajectory $\mathbf{X}^\epsilon$. Applying the definition of $A(\epsilon)$ and $\mathsf{ReLU}$ to Equation (116), we further simplify to show that:

$$\mathsf{KL}(\mathbf{X}_T^\epsilon \| \mathbf{X}_T^0) = \frac{1}{2}\sum_{t=0}^{T-1}\mathbb{E}|\mathsf{ReLU}(\langle a_d(\epsilon), X_t^\epsilon \rangle)|^2 = \frac{1}{2}\sum_{t=1}^{T-1}\mathbb{E}|\mathsf{ReLU}(\langle a_d(\epsilon), X_t^\epsilon \rangle)|^2 \qquad (117)$$

In the equation above, we have used the fact that $X_0^\epsilon = 0$ almost surely. Now, when $t = 1$, we have $X_1^\epsilon = \eta_0 \sim \mathcal{N}(0, I)$ almost surely. Therefore, $\mathbb{E}|\mathsf{ReLU}(\langle a_d(\epsilon), X_1^\epsilon \rangle)|^2 = \frac{\epsilon^2}{2}$. For $t \ge 2$, we will use Lemma 19 and Theorem 8. In this proof only, define $\mathcal{P}_t$ to be the event $\{\langle a_d, X_t \rangle > 0\}$. Therefore, we conclude that:

$$\mathbb{E}|\mathsf{ReLU}(\langle a_d(\epsilon), X_t^\epsilon\rangle)|^2 = \mathbb{E}\mathbb{1}(\mathcal{P}_t)|\langle a_d(\epsilon), X_t^\epsilon\rangle|^2$$
$$\leq \mathbb{E}\mathbb{1}(\mathcal{P}_t)\|a_d(\epsilon)\|^2\|X_t^\epsilon\|^2$$
$$= \mathbb{E}\mathbb{1}(\mathcal{P}_t)\epsilon^2\|X_t^\epsilon\|^2$$
$$\leq \epsilon^2\sqrt{\mathbb{P}(\mathcal{P}_t)}\sqrt{\mathbb{E}\|X_t\|^4} \tag{118}$$

From Lemma 19, we conclude that $\mathbb{P}(\mathcal{P}_t) \leq C_0 \exp(-C_1 d)$ and from Theorem 8, it is clear that the concentration inequalities hold we show that $\sqrt{\mathbb{E}\|X_t\|^4} \leq C_2 d$ for some universal constant $C_2$. Combining the results above with Equation (117), we obtain that for universal constants $C_0, C_1$:

$$\mathsf{KL}(\mathbf{X}_T^\epsilon\|\mathbf{X}_T^0) \leq C_0\left(\epsilon^2 dT \exp(-C_1 d) + \epsilon^2\right) \tag{119}$$

We then use Pinsker's inequality to conclude the result. $\qquad\square$

## G.1 Proof of Theorem 4

*Proof.* Let the notations below be as defined in the statement of the theorem. Since the minimax loss upper bounds any Bayesian loss, we will consider the uniform prior over $\{\mathsf{NLDS}(A(\epsilon), \mathcal{N}(0, I), \mathsf{ReLU}), \mathsf{NLDS}(A(0), \mathcal{N}(0, I), \mathsf{ReLU})\}$. For any estimator $\mathcal{A}$ with input $X_0, \ldots, X_T$, we denote $\mathcal{A}(\mathbf{X})$ as its output. We use the notation $\mathbf{X}^\epsilon$ and $\mathbf{X}^0$ as defined in Lemma 20. Therefore,

$$\mathcal{L}(\Theta(\tfrac{1}{2}), T) \geq \inf_{\mathcal{A}} \frac{1}{2}\mathcal{L}(\mathcal{A}, T, A(0)) + \frac{1}{2}\mathcal{L}(\mathcal{A}, T, A(\epsilon))$$
$$= \frac{1}{2}\inf_{\mathcal{A}}\mathbb{E}\|\mathcal{A}(\mathbf{X}^0) - A(0)\|_F^2 + \mathbb{E}\|\mathcal{A}(\mathbf{X}^\epsilon) - A(\epsilon)\|_F^2 \tag{120}$$

Where the expectation is over the respective trajectories $\mathbf{X}^\epsilon$ and $\mathbf{X}^0$. Now, from Lemma 20 and the coupling calculation of $\mathsf{TV}$ distance, we can define the trajectories $\mathbf{X}^\epsilon$ and $\mathbf{X}^0$ on a common probability space such that $\mathbb{P}(\mathbf{X}^0 \neq \mathbf{X}^\epsilon) \leq \mathsf{TV}(\mathbf{X}^0, \mathbf{X}^\epsilon)$. Picking $\epsilon$ to be small enough such that $\mathsf{TV}(\mathbf{X}^0, \mathbf{X}^\epsilon) \leq \frac{1}{2}$ (which is true for a choice of $\epsilon^2 = c_0 \min(\frac{\exp(c_1 d)}{T}, 1)$ for universal constants $c_0, c_1$). Define the event $\mathcal{S}_T := \{\mathbf{X}^0 = \mathbf{X}^\epsilon\}$, we conclude from Equation (120) that:

$$\mathcal{L}(\Theta(\tfrac{1}{2}), T) \geq \frac{1}{2}\inf_{\mathcal{A}}\mathbb{E}\mathbb{1}(\mathcal{S}_T)\|\mathcal{A}(\mathbf{X}^0) - A(0)\|_F^2 + \mathbb{E}\mathbb{1}(\mathcal{S}_T)\|\mathcal{A}(\mathbf{X}^\epsilon) - A(\epsilon)\|_F^2 \tag{121}$$

Note that over the event $\mathcal{S}_T$, we must have $\mathcal{A}(\mathbf{X}^\epsilon) = \mathcal{A}(\mathbf{X}^0)$ in the case of a deterministic estimator. In case of a random estimator, the proof is same once we observe that their distribution is same and hence can be coupled almost surely. By convexity, we must have: $\|a - b\|^2 + \|b - c\|^2 \geq \frac{1}{2}\|a - c\|^2$. Combining these considerations into the equation above, we have:

$$\mathcal{L}(\Theta(\tfrac{1}{2}), T) \geq \frac{\mathbb{P}(\mathcal{S}_t)}{4}\|A(\epsilon) - A(0)\|_F^2$$
$$\geq \frac{\epsilon^2}{8}. \tag{122}$$

From the choice of $\epsilon^2$ above, we conclude the result.

$\qquad\square$

# H  Proofs of Technical Lemmas

## H.1  Proof of Lemma 10

*Proof.* The proof follows from integrating the tails. Let $Z := \exp(\lambda Y^2)$. For any $\gamma \in \mathbb{R}^+$, we have from the definition of sub-Gaussianity.

$$\mathbb{P}(Z \geq \gamma) = \begin{cases} 1 \text{ if } \gamma \leq 1 \\ \mathbb{P}\left(|Y| \geq \sqrt{\frac{\log(\gamma)}{\lambda}}\right) \text{ if } \gamma > 1 \end{cases} \tag{123}$$

Now,

$$\begin{aligned}
\mathbb{E}Z &= \int_0^\infty \mathbb{P}(Z \geq \gamma)d\gamma \\
&= \int_0^1 d\gamma + \int_1^\infty \mathbb{P}\left(|Y| \geq \sqrt{\frac{\log(\gamma)}{\lambda}}\right)d\gamma \\
&\leq 1 + \int_1^\infty 2\exp\left(-\frac{\log(\gamma)}{2\nu^2\lambda}\right)d\gamma \\
&= 1 + 2\int_1^\infty \gamma^{-\frac{1}{2\nu^2\lambda}}d\gamma \\
&= 1 + \frac{4\nu^2\lambda}{1 - 2\nu^2\lambda} \\
&\leq 1 + 8\lambda\nu^2
\end{aligned} \tag{124}$$

$\square$

## H.2  Proof of Lemma 14

*Proof.* We draw $\tilde{X}_s \sim \pi$, independent of $X_s$. We obtain $\tilde{X}_{s+k}$ by running the markov chain with the same noise sequence. i.e, $\tilde{X}_{s+k+1} = \phi(A^*\tilde{X}_{s+k}) + \eta_{s+k}$. We then obtain $\tilde{X}_t$. Then, it is clear that:

$$\begin{aligned}
\langle X_t, x\rangle^2 \langle X_s, x\rangle^2 &= \langle X_t - \tilde{X}_t + \tilde{X}_t, x\rangle^2 \langle X_s, x\rangle^2 \\
&\leq 2\langle \tilde{X}_t, x\rangle^2 \langle X_s, x\rangle^2 + 2\langle X_t - \tilde{X}_t, x\rangle^2 \langle X_s, x\rangle^2
\end{aligned}$$

Taking expectation on both sides and noting that $\tilde{X}_t$ is independent of $X_s$, we conclude:

$$\mathbb{E}\langle X_t, x\rangle^2 \langle X_s, x\rangle^2 \leq 2(x^\top G x)^2 + 2\mathbb{E}\langle X_t - \tilde{X}_t, x\rangle^2 \langle X_s, x\rangle^2 \tag{125}$$

By Assumption 4, we have: $\|X_t - \tilde{X}_t\|^2 \leq C_\rho^2 \rho^{2(t-s)}\|X_s - \tilde{X}_s\|^2$. Plugging this into Equation (125), we conclude:

$$\begin{aligned}
\mathbb{E}\langle X_t, x\rangle^2 \langle X_s, x\rangle^2 &\leq 2(x^\top G x)^2 + 2\mathbb{E}\|x\|^2 C_\rho^2 \rho^{2(t-s)}\|X_s - \tilde{X}_s\|^2 \langle X_s, x\rangle^2 \\
&\leq 2(x^\top G x)^2 + 4\|x\|^2 C_\rho^2 \rho^{2(t-s)}\mathbb{E}\left(\|X_s\|^2 + \|\tilde{X}_s\|^2\right)\langle X_s, x\rangle^2 \\
&= 2(x^\top G x)^2 + 4\|x\|^2 C_\rho^2 \rho^{2(t-s)}\left[\mathbb{E}\|X_s\|^2 \langle X_s, x\rangle^2 + x^\top G x \mathbb{E}\|\tilde{X}_s\|^2\right] \tag{126}
\end{aligned}$$

We can evaluate $\mathbb{E}\|\tilde{X}_s\|^2$ from Theorems 8 and 9.

1. First we consider the Sub-Gaussian setting with $\|A^*\| = \rho < 1$ and $C_\rho = 1$. Fix $R > 0$. We will use the notation from Theorem 8 below. We can then write,

$$\mathbb{E}\|X_s\|^2\langle X_s, x\rangle^2 = \mathbb{E}\|X_s\|^2\langle X_s, x\rangle^2 \mathbb{1}(\|X_s\|^2 \le R) + \mathbb{E}\|X_s\|^2\langle X_s, x\rangle^2 \mathbb{1}(\|X_s\|^2 > R)$$
$$\le \mathbb{E}R\langle X_s, x\rangle^2 \mathbb{1}(\|X_s\|^2 \le R) + \mathbb{E}\|X_s\|^4 \mathbb{1}(\|X_s\|^2 > R)$$
$$\le Rx^\top Gx + \mathbb{E}\|X_s\|^4 \mathbb{1}(\|X_s\|^2 > R)$$
$$\le Rx^\top Gx + \sqrt{\mathbb{E}\|X_s\|^8}\sqrt{\mathbb{P}(\|X_s\|^2 > R)} \tag{127}$$

From Theorem 8 and Proposition 2.7.1 in [50], we show that $\mathbb{E}\|X_s\|^8 \le C\left(\frac{dC_\eta\sigma^2}{1-\rho}\right)^4$ for some universal constant $C$. Again, taking $R = \frac{8dC_\eta\sigma^2}{1-\rho} + \frac{2\log\frac{1}{\delta}}{\lambda^*} \le \frac{24dC_\eta\sigma^2\log(\frac{1}{\delta})}{1-\rho}$, we have: $\sqrt{\mathbb{P}(\|X_s\|^2 > R)} \le \delta$. We plug this into Equation (127), take $\delta = \frac{(1-\rho)x^\top Gx}{d\sigma^2}$ after noting that $x^\top Gx \ge \sigma^2\|x\|^2$ to show that:

$$\mathbb{E}\|X_s\|^2\langle X_s, x\rangle^2 \le \bar{C}\frac{d\sigma^2}{1-\rho}x^\top Gx \log\left(\frac{d}{1-\rho}\right) \tag{128}$$

Where $\bar{C}$ is a constant which depends only on $C_\eta$. Using Equation (128) in Equation (126), we conclude that:

$$\mathbb{E}\langle X_t, x\rangle^2\langle X_s, x\rangle^2 \le 2(x^\top Gx)^2 + \bar{C}_1 C_\rho^2\rho^{2(t-s)}\frac{d\sigma^2}{1-\rho}x^\top Gx \log\left(\frac{d}{1-\rho}\right) \tag{129}$$

2. From Equation (126), we directly conclude via Cauchy Schwarz inequality that:
$$\mathbb{E}\langle X_t, x\rangle^2\langle X_s, x\rangle^2 \le 2(x^\top Gx)^2 + 8\|x\|^2 C_\rho^2\rho^{2(t-s)}\mathbb{E}\|X_s\|^4$$

We then use the bound on $\mathbb{E}\|X_s\|^4$ given in Theorem 9 to conclude the result.

$\square$

### H.3   Proof of Lemma 2

*Proof.* Consider $\hat{G} = \frac{1}{T}\sum_{\tau=0}^{T-1} X_\tau X_\tau^\top$ and consider the coupled process as in Defintion 2. We will divide the times into buffers of size $B$, with gaps of size $u$ and also consider related notation as given in Section C. Now, $\hat{G} \succeq \frac{B}{NS}\sum_{t=1}^N G^{(t)}$, where $G^{(t)}$ is the empirical second moment matrix of the buffer $t$ given by $\hat{G}^{(t)} := \frac{1}{B}\sum_{i=0}^{B-1} X_i^{t,\top} X_i^t$. Now, consider the coupled second moment matrix defined on buffer $t$ given by $\tilde{G}^{(t)} := \frac{1}{B}\sum_{t=0}^{B-1} \tilde{X}_i^{t,\top}\tilde{X}_i^t$. By Lemma 3, we know that $\|\hat{G}^{(t)} - \tilde{G}^{(t)}\| \le 4N^* C_\rho\rho^u$. Where $M^* = \sup_{\tau \le T}\max\left(\|X_\tau\|^2, \|\tilde{X}_\tau\|^2\right)$. Now, observe that by definition of the coupling, we have that $\tilde{G}_t$ are i.i.d. Combining the considerations above, and letting $B \ge u$ to conclude $\frac{B}{S} \ge \frac{1}{2}$

$$\hat{G} \succeq \frac{B}{SN}\sum_{t=1}^N \hat{G}^{(t)} \succeq \frac{1}{2N}\sum_{t=1}^N \tilde{G}^{(t)} - 4M^* C_\rho\rho^u I \tag{130}$$

Before proceeding further, we will give a high probability bound on $M^*$. By Markov's inequality, and Theorem 9

$$\mathbb{P}(M^* > R) \le \frac{2T\mathbb{E}\|X_\tau\|^2}{R} \le \frac{2TC_\rho^2 d\sigma^2}{(1-\rho)^2\alpha}\,.$$

Consider the event $\hat{\mathcal{D}}_T(R) = \{M^* \le R\}$. Letting $R = \frac{4TdC_\rho^2\sigma^2}{(1-\rho)^2\delta}$ only in this proof, we conclude that:

$$\mathbb{P}(\mathcal{D}_T(R)) \ge 1 - \frac{\delta}{2}$$

Recall that $G = \mathbb{E}X_\tau X_\tau^\top$. Now, as shown by item 2 in Lemma 15, we have whenever $B \ge \frac{4C_\rho^6 M_4}{(1-\rho)^4(1-\rho^2)\sigma^4}$

$$\mathbb{P}\left(x^\top\tilde{G}^{(t)}x \ge \frac{1}{2}x^\top Gx\right) \ge p_0 > 0\,. \tag{131}$$

Now, consider any arbitrary, fixed vector $x \in \mathbb{R}^d$. Using independence of $\hat{G}^{(t)}$ and Equation (131), we conclude that for some universal constants $c_0, c_1$, we must have:

$$\mathbb{P}(\frac{1}{N}\sum_{t=1}^{N} x^\top \tilde{G}^{(t)} x \leq c_0 x^\top G x) \leq \exp(-c_1 N). \tag{132}$$

Rewriting the equation above by taking $x = G^{-1/2}y$, we have:

$$\mathbb{P}(\frac{1}{N}\sum_{t=1}^{N} y^\top G^{-1/2}\tilde{G}^{(t)}G^{-1/2}y \leq c_0\|y\|^2) \leq \exp(-c_1 N). \tag{133}$$

For simplicity of exposition, we will take $J := \frac{1}{N}\sum_{t=1}^{N} G^{-1/2}\tilde{G}^{(t)}G^{-1/2}$ in the calculations below. Before proceeding with a bound on the operator norm, we will give a bound on $\|J\|$. Since $G \succeq \sigma^2 I$, we must have: $\|J\| \leq \frac{M^*}{\sigma^2}$.

Now, we will apply an epsilon net argument. Let $\mathcal{N}_\epsilon$ be an $\epsilon$-net over $\mathcal{S}^{d-1}$. We can take $|\mathcal{N}_\epsilon| \leq (1 + \frac{2}{\epsilon})^d$.

$$\inf_{y \in \mathcal{S}^{d-1}} y^\top J y \geq \inf_{y \in \mathcal{N}_\epsilon} y^\top J y - 2\|J\|\epsilon.$$

We let $\epsilon = \frac{c\sigma^2}{R}$ for some constant $c > 0$ small enough and $R$ as defined earlier in this proof. By union bound over $\mathcal{D}_T^c(R)$ and the event given in Equation (133), we conclude that for some universal constant $c_2 > 0$ small enough:

$$\mathbb{P}(\inf_{y \in \mathcal{S}^{d-1}} y^\top J y > c_2, M^* \leq R) \geq 1 - \exp(Cd\log(\frac{R}{\sigma^2}) - c_1 N) - \frac{\delta}{2} \tag{134}$$

Now, using Equation (130), we conclude that:

$$G^{-1/2}\hat{G}G^{-1/2} \succeq \frac{J}{2} - 4M^* C_\rho \rho^u G^{-1}$$

Using the fact that $G^{-1} \preceq \frac{I}{\sigma^2}$ and using Equation (134), we conclude that whenever $T \geq Cd\log(\frac{1}{\delta})B\log(R/\sigma^2)$, $B \geq u$, $B \geq \frac{4C_\rho^6 M_4}{(1-\rho)^4(1-\rho^2)\sigma^4}$ and $u \geq \frac{\log\left(\frac{RC_1 C_\rho}{\sigma^2}\right)}{\log\left(\frac{1}{\rho}\right)}$, we conclude that for some constant $c_0 > 0$ small enough, with probability atleast $1 - \delta$, we have:

$$\hat{G} \succeq c_0 G$$

$\qquad\qquad\qquad\qquad\qquad\qquad\qquad\qquad\qquad\qquad\qquad\qquad\qquad\qquad\qquad\qquad$ $\square$

## H.4    Proof of Lemma 7

First, we will obtain a crude upper bound on $\|\tilde{a}_j^{t-1} - a^*\|$ using Theorem 10. That is, we want to show that $\|\tilde{a}_j^{t-1} - a^*\|$ does not grow too large with high probability.

**Proposition 3.** *Let $\lambda_{\min} \equiv \lambda_{\min}(G)$. Conditional on $\tilde{\mathcal{D}}^{0,t-1} \cap \cap_{r=0}^{t-1}\mathcal{E}_{0,B-1}^r$, with probability at least $1 - N\delta$, for all $1 \leq t \leq N$, all $1 \leq j \leq B$ we have*

$$\|\tilde{a}_j^{t-1} - a^*\| \leq \|a_0 - a\| + 2\gamma B\sqrt{R\bar{\beta}}C\left(d + \log\frac{N}{\delta} + \frac{1}{\zeta\gamma B\lambda_{\min}}\right) \tag{135}$$

*where $C$ is constant depending only on $C_\eta$.*

*Proof.* Let us start with the expression for $\tilde{a}_j^{t-1} - a^*$

$$(\tilde{a}_j^{t-1} - a^*)^\top = (a_0 - a^*)^\top \left( \prod_{s=0}^{t-2} \tilde{H}_{0,B-1}^s \right) \tilde{H}_{0,j-1}^{t-1} + 2\gamma \sum_{i=0}^{j-1} \phi'(\tilde{\xi}_{-i}^{t-1}) \varepsilon_{-i}^{t-1} \tilde{X}_{-i}^{t-1,\top} \tilde{H}_{i+1,j-1}^{t-1}$$

$$+ 2\gamma \sum_{r=2}^{t} \sum_{i=0}^{B-1} \phi'(\tilde{\xi}_{-i}^{t-r}) \varepsilon_{-i}^{t-r} \tilde{X}_{-i}^{t-r,\top} \tilde{H}_{i+1,B-1}^{t-r} \left( \prod_{s=r-1}^{1} \tilde{H}_{0,B-1}^{t-s} \right) \quad (136)$$

We will work on the event $\tilde{\mathcal{D}}^{0,t-1} \cap \cap_{r=0}^{t-1} \mathcal{E}_{0,B-1}^r$. It is clear from Equation (136) that:

$$\left\| \tilde{a}_j^{t-1} - a^* \right\| \leq \| a_0 - a^* \| + 2\gamma B \sqrt{R\beta} + 2\gamma \sqrt{R\beta} B \sum_{r=2}^{t} \left\| \left( \prod_{s=r-1}^{1} \tilde{H}_{0,B-1}^{t-s} \right) \right\|$$

We use Theorem 11 (with appropriate constant $C > 1$ to account for minor differences in indexing) to show that conditional on $\tilde{\mathcal{D}}^{0,t-1} \cap \cap_{r=0}^{t-1} \mathcal{E}_{0,B-1}^r$, for fixed $t$, with probability at least $1 - \delta$, for all $1 \leq j \leq B$

$$\left\| \tilde{a}_j^{t-1} - a^* \right\| \leq \| a_0 - a^* \| + 2\gamma B \sqrt{R\beta} C \left( d + \log \frac{N}{\delta} + \frac{1}{\zeta \gamma B \lambda_{\min}} \right)$$

Thus taking union bound we get that conditional on $\tilde{\mathcal{D}}^{0,t-1} \cap \cap_{r=0}^{t-1} \mathcal{E}_{0,B-1}^r$ with probability at least $1 - N\delta$, for all $1 \leq t \leq N - 1$ and all $1 \leq j \leq B$

$$\left\| \tilde{a}_j^{t-1} - a^* \right\| \leq \| a_0 - a^* \| + 2\gamma B \sqrt{R\beta} C \left( d + \log \frac{N}{\delta} + \frac{1}{\zeta \gamma B \lambda_{\min}} \right)$$

$\square$

*Proof of Lemma 7.* On the event $\mathcal{E}_{0,j}^r \cap \tilde{\mathcal{D}}^{r,N-1}$, we note the following inequalities

$$\bar{\tilde{a}}_i^s = \tilde{a}_i^s \ 0 \leq s < r, \ 0 \leq i \leq B - 1 \quad (137)$$

$$\bar{\tilde{a}}_0^r = \tilde{a}_0^r \quad (138)$$

$$\left\| \bar{\tilde{a}}_i^s - \tilde{a}_i^s \right\| \leq \begin{cases} 4i\gamma\sqrt{R\beta} + \sum_{k=0}^{i-1} 4\gamma R \left\| \tilde{a}_k^r - a^* \right\|, & s = r, \ 1 \leq i \leq j \\ 4(j+1)\gamma\sqrt{R\beta} + \sum_{k=0}^{j-1} 4\gamma R \left\| \tilde{a}_k^r - a^* \right\|, s = r, & j+1 \leq i \leq B-1 \ (139) \\ 4(j+1)\gamma\sqrt{R\beta} + \sum_{k=0}^{j-1} 4\gamma R \left\| \tilde{a}_k^r - a^* \right\|, r < s, & 0 \leq i \leq B-1 \end{cases}$$

The result then follows from an application of Proposition 3 with $\delta$ chosen as in C.4 $\square$

### H.5 Proof of Lemma 4

We first state and prove the following result:

**Lemma 21.** *Let $R_{\max} := \sup_{\tau \leq T}(\| X_\tau \|^2, \left\| \tilde{X} \right\|^2)$ and suppose $\gamma \leq \frac{1}{2R_{\max}}$. For every $t \in [N]$ and $i \in [B]$ we have:*

$$\| a_i^t \| \leq 2\gamma R_{\max} T .$$

*Proof.* Let the row under consideration be the $k$-th row and $e_k$ be the standard basis vector. Consider the $\mathsf{SGD} - \mathsf{RER}$ iteration:

$$a_{i+1}^t = a_i^t - 2\gamma \left( \phi(\langle a_i^t, X_{-i}^t \rangle) - X_{-(i-1)}^t \right) X_{-i}^t$$

$$= (I - 2\gamma \zeta_{t,i} X_{-i}^t X_{-i}^{t,\top}) a_i^t + 2\gamma \langle X_{-(i-1)}^t, e_k \rangle X_{-i}^t \quad (140)$$

Where $\zeta_{t,i} := \frac{\phi(\langle a_i^t, X_{-i}^t\rangle)}{\langle a_i^t, X_{-i}^t\rangle} \in [\zeta, 1]$ exists in a weak sense due to our assumptions on $\phi$. Observe that for our choice of $\gamma$, we have $\|(I - 2\gamma\zeta_{t,i}X_{-i}^t X_{-i}^{t,\top})\| \leq 1$ and $\|\langle X_{-(i-1)}^t, e_k\rangle X_{-i}^{t,\top}\| \leq R_{\max}$. Therefore, triangle inequality implies:

$$\|a_{i+1}^t\| \leq \|a_i^t\| + 2\gamma R_{\max}$$

We conclude the bound in the Lemma.

$\square$

*Proof of Lemma 4.* Let the row under consideration be the $k$-th row and $e_k$ be the standard basis vector.

$$\begin{aligned}
a_{i+1}^t &= a_i^t - 2\gamma(\phi(\langle a_i^t, X_{-i}^t\rangle) - \langle e_k, X_{-(i-1)}^t\rangle)X_{-i}^t \\
&= a_i^t - 2\gamma(\phi(\langle a_i^t, \tilde{X}_{-i}^t\rangle) - \langle e_k, \tilde{X}_{-(i-1)}^t\rangle)\tilde{X}_{-i}^t + \Delta_{t,i} \quad (141)
\end{aligned}$$

Where

$$\Delta_{t,i} := 2\gamma\left(\phi(\langle a_i^t, \tilde{X}_{-i}^t\rangle)\tilde{X}_{-i}^t - \phi(\langle a_i^t, X_{-i}^t\rangle)X_{-i}^t\right) + 2\gamma\left(\langle X_{-(i-1)}^t, e_k\rangle X_{-i}^t - \langle \tilde{X}_{-(i-1)}^t, e_k\rangle \tilde{X}_{-i}^t\right).$$

Using Lemmas 21 and 3, we conclude that:

$$\|\Delta_{t,i}\| \leq (16\gamma^2 R_{\max}^2 T + 8\gamma R_{\max})\rho^u$$

Using the recursion for $\tilde{a}_i^t$, we conclude:

$$\begin{aligned}
a_{i+1}^t - \tilde{a}_{i+1}^t &= (I - 2\gamma\tilde{\zeta}_{t,i}\tilde{X}_i^t \tilde{X}_i^{t,\top})(a_i^t - \tilde{a}_i^t) + \Delta_{t,i} \\
\implies \|a_{i+1}^t - \tilde{a}_{i+1}^t\| &\leq \|a_i^t - \tilde{a}_i^t\|\left\|(I - 2\gamma\tilde{\zeta}_{t,i}\tilde{X}_i^t \tilde{X}_i^{t,\top})\right\| + (16\gamma^2 R_{\max}^2 T + 8\gamma R_{\max})\rho^u \\
\implies \|a_{i+1}^t - \tilde{a}_{i+1}^t\| &\leq \|a_i^t - \tilde{a}_i^t\| + (16\gamma^2 R_{\max}^2 T + 8\gamma R_{\max})\rho^u \quad (142)
\end{aligned}$$

In the first step, $\tilde{\zeta}_{t,i} := \frac{\phi(\langle a_i^t, \tilde{X}_{-i}^t\rangle) - \phi(\langle \tilde{a}_i^t, \tilde{X}_{-i}^t\rangle)}{\langle a_i^t, \tilde{X}_{-i}^t\rangle - \langle \tilde{a}_i^t, \tilde{X}_{-i}^t\rangle} \in [\zeta, 1]$. In the last step we have used the fact that under the conditions on $\gamma$, we must have $\left\|(I - 2\gamma\tilde{\zeta}_{t,i}\tilde{X}_i^t \tilde{X}_i^{t,\top})\right\| \leq 1$. We conclude the statement of the lemma from Equation (142). $\square$

## H.6 Proof of Claim 3

*Proof.* Let $r_2 > r_1$. As in proof of Claim 2, let $\text{Cr}'$ denote the resampled version of Cr obtained by re-sampling $\eta_{-j_1}^{t-r_1}$ i.e.,

$$\begin{aligned}
\text{Cr}'(t, r_1, r_2, j_1, j_2) &:= 4\gamma^2 \varepsilon_{-j_1}^{t-r_1} \varepsilon_{-j_2}^{t-r_2} \mathcal{R}_{-j_1}^{t-r_1}\left[\tilde{X}_{-j_2}^{t-r_2,\top} \tilde{H}_{j_2+1,B-1}^{t-r_2}\left(\prod_{s=r_2-1}^{1}\tilde{H}_{0,B-1}^{t-s}\right)\cdot\right. \\
&\left.\left(\prod_{s=1}^{r_1-1}\tilde{H}_{0,B-1}^{t-s,\top}\right)\tilde{H}_{j_1+1,B-1}^{t-r_1,\top}\tilde{X}_{-j_1}^{t-r_1}\right] \\
&= 4\gamma^2 \varepsilon_{-j_1}^{t-r_1} \varepsilon_{-j_2}^{t-r_2} \tilde{X}_{-j_2}^{t-r_2,\top}\left(\tilde{H}_{j_2+1,B-1}^{t-r_2}\right)\left(\prod_{s=r_2-1}^{r_1+1}\tilde{H}_{0,B-1}^{t-s}\right)\cdot \\
&\mathcal{R}_{-j_1}^{t-r_1}\left(\prod_{s=r_1}^{1}\tilde{H}_{0,B-1}^{t-s}\right)\mathcal{R}_{-j_1}^{t-r_1}\left(\prod_{s=1}^{r_1-1}\tilde{H}_{0,B-1}^{t-s,\top}\right)\mathcal{R}_{-j_1}^{t-r_1}\left(\tilde{H}_{j_1+1,B-1}^{t-r_1,\top}\right)\tilde{X}_{-j_1}^{t-r_1}
\end{aligned}$$

$$(143)$$

Here we have used the fact that $\mathcal{R}_{-j_1}^{t-r_1}$ does not affect the buffers up to $t - r_1 - 1$ and only $\tilde{X}$s that are affected are in the term $\tilde{H}_{0,j_1-1}^{t-r_1}$. Like in Claim 2, notice that

$$\mathbb{E}\left[\sum_{r_2 > r_1}\sum_{j_1, j_2}\text{Cr}'(t, r_1, r_2, j_1, j_2)\right] = 0$$

Applying Lemma 9, we conclude that:

$$
\sum_{r_2 > r_1} \sum_{j_1, j_2} \mathrm{Cr}'(t, r_1, r_2, j_1, j_2)
$$

$$
= 2\gamma \sum_{r_1=1}^{t-1} \sum_{j_1=0}^{B-1} (\tilde{a}_B^{t-r_1-1,v})^\top \mathcal{R}_{-j_1}^{t-r_1} \left( \tilde{H}_{0,B-1}^{t-r_1} \right) \mathcal{R}_{-j_1}^{t-r_1} \left( \prod_{s=r_1-1}^{1} \tilde{H}_{0,B-1}^{t-s} \right) \cdot
$$

$$
\mathcal{R}_{-j_1}^{t-r_1} \left( \prod_{s=1}^{r_1-1} \tilde{H}_{0,B-1}^{t-s,\top} \right) \mathcal{R}_{-j_1}^{t-r_1} \left( \tilde{H}_{j_1+1,B-1}^{t-r_1,\top} \right) \tilde{X}_{-j_1}^{t-r_1} \varepsilon_{-j_1}^{t-r_1} \tag{144}
$$

We cannot continue our analysis like in Claim 2 because due to resampling of $\varepsilon_{-j_1}^{t-r_1}$, $\tilde{H}_{0,B-1}^{t-r_1}$ changes not just because of the iterates $\tilde{a}_i^{t-r_1}$ but also due to $\tilde{X} \to \bar{\tilde{X}}$.

Further

$$
\mathbb{E}\left[ \sum_{r_2 > r_1} \sum_{j_1, j_2} \mathrm{Cr}'(t, r_1, r_2, j_1, j_2) \mathbf{1}\left[ \tilde{\mathcal{D}}^{0,t-r_1-1} \right] \mathbf{1}\left[ \tilde{\mathcal{D}}^{t-r_1+1,t-1} \right] \mathbf{1}\left[ \tilde{\mathcal{D}}_{-j_1}^{t-r_1} \right] \right] = 0 \tag{145}
$$

Next we have simple lemma

**Lemma 22.** *Consider for each* $(r_1, j_1)$, *the re-sampling operator* $\mathcal{R}_{-j_1}^{t-r_1}$

$$
\left\| \mathbb{E}\left[ \sum_{r_2 > r_1} \sum_{j_1, j_2} \mathrm{Cr}(t, r_1, r_2, j_1, j_2) \mathbf{1}\left[ \tilde{\mathcal{D}}^{0,t-1} \right] \right] \right\| \le 4\gamma^2 R \frac{(Bt)^2}{2} C_\eta \sigma^2 \frac{1}{T^{\alpha/2}} +
$$

$$
\left\| \mathbb{E}\left[ \sum_{r_2 > r_1} \sum_{j_1, j_2} \mathrm{Cr}(t, r_1, r_2, j_1, j_2) \mathbf{1}\left[ \tilde{\mathcal{D}}^{0,t-1} \right] \mathcal{R}_{-j_1}^{t-r_1} \mathbf{1}\left[ \tilde{\mathcal{D}}_{-0}^{t-r_1} \right] \right] \right\| \tag{146}
$$

*Proof.* We have

$$
\mathbf{1}\left[ \tilde{\mathcal{D}}^{0,t-1} \right] = \mathbf{1}\left[ \tilde{\mathcal{D}}^{0,t-1} \right] \mathcal{R}_{-j_1}^{t-r_1} \mathbf{1}\left[ \tilde{\mathcal{D}}_{-0}^{t-r_1} \right] + \mathbf{1}\left[ \tilde{\mathcal{D}}^{0,t-1} \right] \mathcal{R}_{-j_1}^{t-r_1} \mathbf{1}\left[ \tilde{\mathcal{D}}_{-0}^{t-r_1,C} \right] \tag{147}
$$

Hence

$$
\left\| \mathbb{E}\left[ \sum_{r_2 > r_1} \sum_{j_1, j_2} \mathrm{Cr}(t, r_1, r_2, j_1, j_2) \mathbf{1}\left[ \tilde{\mathcal{D}}^{0,t-1} \right] \right] \right\|
$$

$$
\le \left\| \mathbb{E}\left[ \sum_{r_2 > r_1} \sum_{j_1, j_2} \mathrm{Cr}(t, r_1, r_2, j_1, j_2) \mathbf{1}\left[ \tilde{\mathcal{D}}^{0,t-1} \right] \mathcal{R}_{-j_1}^{t-r_1} \mathbf{1}\left[ \tilde{\mathcal{D}}_{-0}^{t-r_1} \right] \right] \right\| +
$$

$$
4\gamma^2 R \frac{(Bt)^2}{2} C_\eta \sigma^2 \frac{1}{T^{\alpha/2}} \tag{148}
$$

where we used $\mathcal{R}_{-j_1}^{t-r_1} \mathbf{1}\left[ \tilde{\mathcal{D}}_{-0}^{t-r_1,C} \right]$ is identically distributed as $\mathbf{1}\left[ \tilde{\mathcal{D}}_{-0}^{t-r_1,C} \right]$ and hence $\mathbb{E}\left[ \mathcal{R}_{-j_1}^{t-r_1} \mathbf{1}\left[ \tilde{\mathcal{D}}_{-0}^{t-r_1,C} \right] \right] \le \frac{1}{T^\alpha}$

$\square$

So, based on the above lemma, we focus on bounding

$$
\left\| \mathbb{E}\left[ \sum_{r_2 > r_1} \sum_{j_1, j_2} \mathrm{Cr}(t, r_1, r_2, j_1, j_2) \mathbf{1}\left[ \tilde{\mathcal{D}}^{0,t-1} \right] \mathcal{R}_{-j_1}^{t-r_1} \mathbf{1}\left[ \tilde{\mathcal{D}}_{-0}^{t-r_1} \right] \right] \right\|
$$

Now notice that

$$
\mathbb{E}\left[\sum_{r_2>r_1}\sum_{j_1,j_2}\mathrm{Cr}'(t,r_1,r_2,j_1,j_2)\cdot\right.
$$
$$
\left.1\left[\tilde{\mathcal{D}}^{0,t-r_1-1}\right]1\left[\tilde{\mathcal{D}}^{t-r_1+1,t-1}\right]1\left[\tilde{\mathcal{D}}_{-j_1}^{t-r_1}\right]\mathcal{R}_{-j_1}^{t-r_1}1\left[\tilde{\mathcal{D}}_{-0}^{t-r_1}\right]\right]
$$
$$
=0 \tag{149}
$$

Hence

$$
\mathbb{E}\left[\sum_{r_2>r_1}\sum_{j_1,j_2}\mathrm{Cr}'(t,r_1,r_2,j_1,j_2)1\left[\tilde{\mathcal{D}}^{0,t-1}\right]\mathcal{R}_{-j_1}^{t-r_1}1\left[\tilde{\mathcal{D}}_{-0}^{t-r_1}\right]\right]=0-
$$
$$
\mathbb{E}\left[\sum_{r_2>r_1}\sum_{j_1,j_2}\mathrm{Cr}'(t,r_1,r_2,j_1,j_2)1\left[\tilde{\mathcal{D}}^{0,t-r_1-1}\right]1\left[\tilde{\mathcal{D}}^{t-r_1+1,t-1}\right]\cdot\right.
$$
$$
\left.1\left[\tilde{\mathcal{D}}_{-j_1}^{t-r_1}\right]1\left[\cup_{i=0}^{j_1-1}\tilde{\mathcal{C}}_{-i}^{t-r,C}\right]\mathcal{R}_{-j_1}^{t-r_1}1\left[\tilde{\mathcal{D}}_{-0}^{t-r_1}\right]\right] \tag{150}
$$

Thus

$$
\left|\mathbb{E}\left[\sum_{r_2>r_1}\sum_{j_1,j_2}\mathrm{Cr}'(t,r_1,r_2,j_1,j_2)1\left[\tilde{\mathcal{D}}^{0,t-1}\right]\mathcal{R}_{-j_1}^{t-r_1}1\left[\tilde{\mathcal{D}}_{-0}^{t-r_1}\right]\right]\right|\leq 2\gamma^2 R\frac{(Bt)^2}{2}C_\eta\sigma^2\frac{1}{T^{\alpha/2}} \tag{151}
$$

Now, similar to lemma 8, on the event $\mathcal{E}_{0,j_1}^{r_1}\cap\tilde{\mathcal{D}}^{0,t-1}\cap\mathcal{A}^{t-1}$ we have:

$$
\left\|\left(\prod_{s=r_1-1}^{1}\tilde{H}_{0,B-1}^{t-s}\right)-\mathcal{R}_{-j_1}^{t-r_1}\left(\prod_{s=r_1-1}^{1}\tilde{H}_{0,B-1}^{t-s}\right)\right\|\leq CBt\|\phi''\|\gamma^2 R^3 B\frac{\sqrt{\beta}}{\zeta\lambda_{\min}} \tag{152}
$$

Next, similar to lemma 8 for $\gamma R\leq\frac{1}{2}$, on the event $\tilde{\mathcal{D}}_{-0}^{t-r_1}\cap\cap_{i=0}^{B-1}\left\{\left\|\mathcal{R}_{j_1}^{t-r_1}\tilde{X}_{-i}^{t-r_1}\right\|^2\leq R\right\}$ we have

$$
\left\|\tilde{H}_{0,B-1}^{t-r_1}-\mathcal{R}_{-j_1}^{t-r_1}\left(\tilde{H}_{0,B-1}^{t-r_1}\right)\right\|\leq 4\gamma RB \tag{153}
$$

Finally we can bound the norm of the expected difference of sums of Cr and Cr$'$ using lemma 9 and (144) as

$$
\left|\mathbb{E}\left[\sum_{r_2>r_1}\sum_{j_1,j_2}(\mathrm{Cr}-\mathrm{Cr}')1\left[\tilde{\mathcal{D}}^{0,t-1}\right]\mathcal{R}_{-j_1}^{t-r_1}1\left[\tilde{\mathcal{D}}_{-0}^{t-r_1}\right]1\left[\cap_{s=0}^{t-1}\cap_{i=0}^{B-1}\mathcal{E}_i^s\right]1\left[\mathcal{A}^{t-1}\right]\right]\right|
$$
$$
\leq 2\gamma\mathbb{E}\left[\sum_{r_1=1}^{t-1}\sum_{j_1}\sqrt{R}|\varepsilon_{-j_1}^{t-r_1}|\left[\left\|\tilde{a}_B^{t-r_1-1,v}\right\|1\left[\tilde{\mathcal{D}}^{0,t-r_1-1}\right]\right]\cdot\right.
$$
$$
\left.\left(C\|\phi''\|\gamma^2 TR^3 B\frac{\sqrt{\beta}}{\zeta\lambda_{\min}}+C\gamma RB\right)\right]
$$
$$
\leq\left(C\|\phi''\|\gamma^3 T^2 R^3 B\frac{\sqrt{\beta}}{\zeta\lambda_{\min}}+C\gamma^2 TRB\right)\sqrt{RC_\eta\sigma^2}\sqrt{\sup_{s\leq N-1}\mathbb{E}\left[\|\tilde{a}_B^{s,v}\|^2 1\left[\tilde{\mathcal{D}}^{0,s}\right]\right]} \tag{154}
$$

Thus

$$\left| \mathbb{E}\left[ \sum_{r_2 > r_1} \sum_{j_1, j_2} (\mathrm{Cr} - \mathrm{Cr}') \mathbb{1}\left[ \tilde{\mathcal{D}}^{0,t-1} \right] \mathcal{R}_{-j_1}^{t-r_1} \mathbb{1}\left[ \tilde{\mathcal{D}}_{-0}^{t-r_1} \right] \right] \right|$$

$$\leq C \left( \|\phi''\| \gamma^3 T^2 R^3 B \frac{\sqrt{\beta}}{\zeta \lambda_{\min}} + \gamma^2 T R B \right) \sqrt{R C_\eta \sigma^2} \sqrt{\sup_{s \leq N-1} \mathbb{E}\left[ \|\tilde{a}_B^{s,v}\|^2 \mathbb{1}\left[ \tilde{\mathcal{D}}^{0,s} \right] \right]}$$

$$+ C(Bt)^2 \left[ \gamma^2 R C_\eta \sigma^2 \left( \sqrt{\mathbb{P}\left[ \cup_{s=0}^{N-1} \cup_{i=0}^{B-1} \mathcal{E}_i^{s,C} \right]} + \sqrt{\mathbb{P}\left[ \mathcal{A}^{t-1,C} \right]} \right) \right]$$

$$\leq C \left( \|\phi''\| \gamma^3 T^2 R^3 B \frac{\sqrt{\beta}}{\zeta \lambda_{\min}} + \gamma^2 T R B \right) \sqrt{R C_\eta \sigma^2} \sqrt{\sup_{s \leq N-1} \mathbb{E}\left[ \|\tilde{a}_B^{s,v}\|^2 \mathbb{1}\left[ \tilde{\mathcal{D}}^{0,s} \right] \right]}$$

$$+ C(Bt)^2 \gamma^2 R C_\eta \sigma^2 \frac{1}{T^{\alpha/2}}$$

$$(155)$$

Combining everything we conclude the claim.

$\square$