# OpenReview forum: "Near-optimal Offline and Streaming Algorithms for Learning Non-Linear Dynamical Systems"
_NeurIPS.cc/2021/Conference — NeurIPS 2021 Spotlight_

### Official Review · Reviewer_mBux · 2021-07-07

**Rating:** 6
**Confidence:** 2

**Summary:**

This work considers learning NLDS. Previous works can only learn mixing systems with sub-gaussian noise in the offline setting, with sample complexity depending on the mixing time. This work improves existing results in several ways, assuming the link function is expansive:
1, it's shown that the offline Quasi Newton Method achieves near-optimal convergence rate for non-mixing systems.
2, the offline Quasi Newton Method still achieves near-optimal convergence rate with heavy-tail noise, assuming mixing systems.
3, a new algorithm SGD-RER is introduced, which achieves near-optimal convergence rate in the online setting.
4, an exponential lower bound is proven for the ReLU link function, showing the expansivity assumption is necessary.

**Limitations And Societal Impact:**

Yes

**Main Review:**

Originality: This work focuses on a classic problem: learning NLDS. Nevertheless, the settings considered in this paper are less touched in previous works, and the use of SGD-RER is novel.

Quality: Although the authors claim that the upper bounds in this work is independent of the mixing time, a new term $\lambda_{min} (\hat{G})$ is introduced which would make the upper bounds vacuous when being very small. It's unclear whether we can give a good lower bound for $\lambda_{min} (\hat{G})$, thus the result in this work isn't a $\text{strict}$ improvement over previous works. In line 164 there is no support why $\lambda_{min} (\hat{G})\ge \sigma^2$ should hold. In line 165, $\rho<1+C/T$ seems problematic.

The assertion of Theorem 2 is non-standard (and actually weaker). High-probability upper bounds of errors are preferred in common. The secondary choice is proving an upper bound for the expectation of error. However, Theorem 2 proves a high-probability bound for the expectation of errors, which might fail in the two settings above.

Clarity: The notations are heavy. Hiding ignorable variables and the parameter setting can improve the readability of theorems. The proof sketch of Section 3 is too brief compared with that of Theorem 4. More details should be included, for example, why can we use the convex proxy loss in line 127?

Significance: This work improves existing results for learning NLDS in several ways. The new algorithms/improved analyses can be useful for solving practical problems in RL and control.

Other comments: in line 163, number 'of' samples.
In line 179, $\tau_{mix} \approx 1/(1-\rho)$.
In line 180, the probability measure should be specified for the sigma algebra.

**Time Spent Reviewing:**

12

---

> ### Author Response · Authors · 2021-08-09
> **Response to the concerns raised**
>
> We thank the reviewer for the kind comments.
>
> ### Dependence on $ \lambda_{\min}(\hat{G})$:
> There is no additional assumption regarding $ \lambda_{\min}(\hat{G})$. Indeed, we prove that  $ \lambda_{\min}(\hat{G}) \geq \frac{\sigma^2}{2}$ (with probability $1-\delta$) in Lemma 17 (under the conditions of Theorem 1) and it is also stated in Theorem 1. We believe that the reviewer misinterpreted our claim regarding $ \lambda_{\min}(\hat{G})$; we will state it more clearly in the next version of the manuscript.
>
>
> ### Concerns about $\rho<1+\frac{C}{T}$:
> We are not sure what concern the reviewer has with  $\rho<1+\frac{C}{T}$ setting. Presuming that the reviewer is asking about $\rho>1+\frac{C}{T}$ setting, we agree that  further research is needed to understand the 'explosive’ case where $\rho-1 \gg \frac{C}{T}$. But, we believe that studying $\rho < 1+ \frac{C}{T}$ is important for the following reasons: a) even for offline estimation of linear dynamical systems, the existing results apply to $\rho<1+\frac{C}{T}$ only, unless magnitude of all the eigenvalues are larger than $1+\frac{1}{T}$(see [10], [11]).  b) the current setting already covers a lot of systems of interest, including all stable systems.
>
>
> ### Concerns about Theorem 2:
> We agree with the reviewer’s claim that the assertion of Theorem 2 is non-standard. However, it is easy to obtain high probability bounds as well by using standard median-of-means estimators along with Theorem 2. For instance we can split the samples into $k=\log T$ blocks of $\frac{T}{\log T}$ samples each with a gap of $\tau_{\mathsf{mix}} \log T$ samples in between. Now running algorithm on each of these blocks gives us independent estimates $\{ \hat A_1,\cdots \hat A_k \}$  for $A^*$ such that $\mathbb{P}\left( \| \hat A_i-A^* \|^{2}_{\mathsf{F}} \leq O(\tfrac{d^2 \log T}{T}) \right) \geq \frac{2}{3}$ (as can be guaranteed by Theorem 2). Then applying [Hsu and Sabato, Algo 3] with $\rho(,)$ begin the Frobenius norm gives an estimate $\hat A$ such that $\mathbb{P}\left( \| \hat{A}-A^* \|^{2} \leq O(\tfrac{d^2 \log T}{T}) \right) \geq 1- \frac{1}{T^{50}} $. We will add a note clarifying this aspect in the next version of the manuscript.
>
> ### Clarity:
> We will improve the readability of the paper as recommended by making the theorem statements easily parseable and adding more detailed proof sketches.
>
> ### Refs:
> Hsu, Daniel, and Sivan Sabato. "Loss minimization and parameter estimation with heavy tails." The Journal of Machine Learning Research 17(1) (2016): 543-582.

---

> > ### Comment · Reviewer_mBux · 2021-09-02
> > **Reply**
> >
> > Thank you for your clarification. I think the results are solid but the writing can be improved. Some brief assertions can use further explanation for readability. Putting less important intermediate steps all to appendix might be a good idea as well?

---

> > > ### Author Response · Authors · 2021-09-02
> > > **Response**
> > >
> > > Thank you for the useful feedback regarding readability. We will indeed move some of the less important details to the appendix in order to make some space for relevant discussions regarding the results, especially regarding the interpretation of the results of Theorems 1 and 2 as detailed in the review and response.

---

### Official Review · Reviewer_WJnt · 2021-07-11

**Rating:** 7
**Confidence:** 1

**Summary:**

The authors consider non-linear dynamical systems of the form $X_{t+1} = \phi(A^* X_t) + \eta_t$, where $\eta_t$ is a noise term, $A$ is a $d \times d$ matrix and $\phi$ acts component-wise, and provide some results relating to the estimation of $A^* $ under certain assumptions. A key assumption is that $\phi$ is "expansive", which I understand to be a sort of converse Lipschitz condition. (Since 1-Lipschitzness is also assumed, this means that $\zeta |x - y| \le |\phi(x) - \phi(y)| \le |x - y|$ for some $\zeta \in (0, 1)$. I wasn't sure why this was called "expansive", because it seems to me more to be a limit on contraction than expansion per se, but this seemed understandable.) Under these conditions, they provide:

1. An algorithm that, given sub-Gaussian noise, can recover $A^*$ with squared-norm error scaling in $O(d^2/T)$ (though I have questions about this, see main review).
2. An algorithm that, given system stability and finite-fourth-moment noise, can recover $A^*$ with squared-norm error scaling in $O(d^2/T)$ (though I have questions about this too).
3. A _streaming_ algorithm that, given sub-Gaussian noise, system stability and twice-differentiable $\phi$, can recover $A^*$ with squared-norm error scaling in $O(d^2/T)$.
4. A negative result with the non-expansive ReLU function, showing a lower bound on squared-norm error in this case of $\Theta(e^{cd}/T)$, implying a need for exponentially many samples to recover $A^*$ satisfactorily.


**Limitations And Societal Impact:**

The paper is almost entirely theoretical in nature and I don't foresee any direct societal impact as a result of this work.

**Main Review:**

Firstly—there are a lot of results in this paper. Some of the proofs are very involved and the authors had space only for just over one page for all four proof sketches, leaving 36 pages for the supplementary material. I congratulate the authors on their work. I suspect this could have comfortably been two papers if the authors had so wished. On the other hand, this volume made it an apparent challenge for the authors to structure the paper and highlight the key ideas in their work. For this reason, I wonder if this may even have been a stronger submission with just half its content but more thoroughly explained.

**Clarity:** The summary Table 1 was very helpful in understanding how the results fit together. The main assumptions of the paper are listed in one place, which makes for some back-and-forth reference, but I can appreciate why this structure was chosen. The notation was generally clear and I didn't find it too confusing. If the authors can find a way to make pagination/space work, a minor suggestion is to show most the auxiliary definitions in Theorems 1 and 2 in display mode (rather than inline).

I had a couple of clarifying questions about the results:

1. In Theorem 1, $R^*$ appears in the $d^2$ term, and it is proportional to $d (\sum_{t=1}^T \rho^t)^2$ (ignoring log terms). If $\rho < 1$ this would not be an issue but I believe this theorem is specifically about the case where $\rho > 1$, which if I'm not mistaken would make $R^* = O(d \rho^{2T})$ (ignoring log terms), $\rho^T$ being the dominating term in the summation (and then squared). But that makes $R^*$ exponential in $T$. Then wouldn't we have $$\log\left(1 + \tfrac{R^*}{\sigma^2}\right) = O(\log d + 2T \log \rho),$$  (ignoring log-log terms) thereby cancelling out the $T$ in the denominator of the squared error expression? Then we would have an error in $O(d^2 \log d)$ rather than $O(d^2/T)$ as claimed? I have probably missed something here and politely invite the authors to clarify.

2. The left-hand side of the main line of this theorem has $\mathbb{1}(\mathcal{W})$ in it, which if I'm not mistaken is a random variable (being the indicator of an event). This makes the left-hand side of that expression a random variable (since it is an expectation multiplied by a random variable). Then is the inequality in this theorem intended to be an almost-sure inequality? Or are the authors trying to say that with probability at least $1 - \delta$,
$$\mathbb{E}\lVert A_m - A^* \rVert_F^2 \le \frac{C_0 d^2 \sigma^2}{\zeta^2 T \lambda_\mathrm{min}(G)}?$$
I again hope the authors can clarify.

**Significance:** Since there were so many results, the implications and significance were sometimes hard to follow. Since all of the results appear to be interrelated and highly specific, I would have appreciated a section explicitly discussing how results fit into existing knowledge and what gaps remain—preferably for all of the results of the paper in one place. I appreciate that this may necessitate even further trimming some sections to allow it to fit, but I think it would be helpful.

My impression in general is that the results are significant and likely to be built on, but I think this paper would be much stronger with a clearer discussion of what the authors envisage with respect to this.

**Quality:** I could not check any of the proofs in the supplementary material in detail, but there was nothing unsound that I could detect from my review of the main paper (except perhaps question 1 above), and a quick scroll through the section A of the supplementary material. I tend to think the nature of the assumptions makes the weaknesses of these results somewhat self-explanatory, and the authors make references to these in passing, though this is part of the discussion that I mentioned above might be useful.

**Originality:** As far as I can tell, the proof techniques aren't brand new but their applications in this setting may be somewhat new, and the cases addressed in the paper are new contributions. However, I am not sufficiently familiar with the surrounding literature to provide a confident assessment of how original this is given the prior art.

**Typos:** (not part of assessment, just for benefit of authors)
- Line 17, stray period after "via".
- Line 160–161, "for _a_ possibly unstable".
- Line 162, "in _the_ linear system".
- Line 163, "mixing _systems_" (plural).
- Theorem 1, "Assumptions _1, 3_ and 4" (missing comma).
- Theorem 2, the expression for $m$ has $R^*$ in it, I assume this was meant to be $R$?
- Line 181, Section B.1, I assume this was meant to say Section A.2.
- Also, a reference to the proof of Theorem 1 was omitted.
- Remark 2, "_Theorem_ 3" (missing capital letter).
- Line 273, missing period.
- Line 277, "SGD-ER".

_Edit, 02 Sep 2021: Raised rating from 6 to 7, as discussed in below responses/comments._

**Time Spent Reviewing:**

7

---

> ### Author Response · Authors · 2021-08-09
> **Response to the comments**
>
> We thank the reviewer for the kind comments.
>
> ### Clarity, exposition and volume:
> We believe that the work presents a complete story about learning from trajectories of data and explores the limits of such algorithms in various directions (online vs. offline, expansive vs. non-expansive, sub-gaussian vs heavy tailed etc.). Thus we believe that all these results form part of a single coherent manuscript. We will work on structuring the paper in a better way and include a discussion which ties together all the results to bridging out the overarching theme.
>
>
> ### Regarding clarifying questions:
> 1. We agree that when $\rho > 1 + c $, the recovery guarantees are vacuous (see the discussion below Theorem 1 for further exposition). We intend the theorem to be applied in the setting of $\rho < 1+\frac{C}{T}$ (similar to [10] which studies a similar regime in offline linear systems). We would like to stress that the result is novel for $\rho \leq 1$ setting as well.
>
> 2.  The indicator is within the expectation (we will add brackets to clarify this), so the inequality as stated is deterministic. We interpret this as taking expectation over a high probability event. However, it is easy to obtain high probability bounds as well by using standard median-of-means estimators. For instance we can split the samples into $k=\log T$ blocks of $\frac{T}{\log T}$ samples each with a gap of $\tau_{\mathsf{mix}} \log T$ samples in between. Now running algorithm on each of these blocks gives us independent estimates $\{ \hat A_1,\cdots \hat A_k \}$  for $A^*$ such that $\mathbb{P}\left( \| \hat A_i-A^* \|^{2}_{\mathsf{F}} \leq O(\tfrac{d^2 \log T}{T}) \right) \geq \frac{2}{3}$ . Then applying [Hsu and Sabato, Algo 3] with $\rho(,)$ begin the Frobenius norm gives an estimate $\hat A$ such that $\mathbb{P}\left( \| \hat{A}-A^* \|^{2} \leq O(\tfrac{d^2 \log T}{T}) \right) \geq 1- \frac{1}{T^{50}} $. We will add a note clarifying this aspect in the next version of the manuscript.
>
>
> ### Significance:
> We do attempt to clarify the positioning of our results vis-a-vis existing works in Table 1 and in contributions listed in Introduction. Space permitting, we will add more discussion in this regard in Section 8 (Conclusion).
>
> ### Regarding originality:
> While many of the techniques we have used in the paper are well known (modified Newton method and self normalized martingales), we would like to point out that there are many technical novelties introduced in the work. Reverse experience replay is a fairly new technique which was previously only applied to the linear system identification problem. We believe that the non-linear bias variance decomposition and its analysis using algorithmic stability is novel and an important technical contribution.
>
> ### Regarding minor typos:
> We will correct them as soon as possible.
>
> ### Refs:
> Hsu, Daniel, and Sivan Sabato. "Loss minimization and parameter estimation with heavy tails." The Journal of Machine Learning Research 17(1) (2016): 543-582.

---

> > ### Comment · Reviewer_WJnt · 2021-09-02
> > **Thanks for the response!**
> >
> > I apologise for the tardiness of my response and thank the authors very much for their clarifications. They're very helpful and I'm happy to raise my rating to a 7. Adding brackets in the main equation of Theorem 2 as indicated will be really helpful. I appreciate the commitment to add a clearer discussion about the implications of the results in the context of the general research area in Section 8.

---

> > > ### Author Response · Authors · 2021-09-02
> > > **Response**
> > >
> > > We thank the reviewer for the prompt response.

---

### Official Review · Reviewer_99Uh · 2021-07-15

**Rating:** 7
**Confidence:** 3

**Summary:**

The authors study a very special case of "non-linear dynamical systems" where there is no hidden state and there is some expansive, component-wise "link function" \phi in the evolution X(t+1) = \phi(AX(t)) + noise. For this very special case, they show both off-line and on-line algorithms. They also show lower bounds on the sample complexity of learning systems with with non-expansive link functions (such as ReLU).


**Ethical Concerns:**

None.

**Limitations And Societal Impact:**

None.

**Main Review:**

The authors study a very special case of "non-linear dynamical systems" where there is no hidden state and there is some expansive, component-wise "link function" \phi in the evolution X(t+1) = \phi(AX(t)) + noise. For this very special case, they show both off-line and on-line algorithms. They also show lower bounds on the sample complexity of learning systems with with non-expansive link functions (such as ReLU).

Originality:

The work seems very original.

Quality and significance:

Removing the T_mix term in the sample complexity is a major contribution, albeit to a problem where real-world applications are "yet to be found".

Clarity:

While I have not checked the proofs in the supplementary material in detail, the main body of the text is very well written.

**Time Spent Reviewing:**

3

---

> ### Author Response · Authors · 2021-08-09
> **Response to Review**
>
> We thank the reviewer for the kind comments.
>
> ### Regarding applications of the results/model :
>
> We would like to emphasize that while this work solves a specific problem, the larger aim is to develop techniques which can be used to learn from a single trajectory of Markovian data without reducing it to learning from i.i.d. data by considering exactly one every $\tau_{\mathsf{mix}}$ samples. Therefore, we study an expressive model of dynamical systems where a closed solution to the empirical risk is not known. Previous results of this kind analyzed closed form solutions (example: OLS estimator for linear dynamical systems) to arrive at similar results. We expect the techniques introduced here to be widely applicable in fields where learning from dependent data is essential like Reinforcement Learning. Indeed SGD-RER rigorizes the heuristics which have been successfully used in RL and presents a roadmap to analyze such algorithms.

---

### Official Review · Reviewer_iKUq · 2021-07-16

**Rating:** 6
**Confidence:** 3

**Summary:**

This work has four major contributions: a) authors provide the first offline algorithm that can learn non-linear dynamical systems without the mixing assumption, b) authors significantly improve upon the sample complexity of existing results for mixing systems, c) in the much harder one-pass, streaming setting authors study a SGD with Reverse Experience Replay (SGD − RER) method, and demonstrate that for mixing systems, it achieves the same sample complexity as our offline algorithm, d) authors justify the expansivity assumption by showing that for the popular ReLU link function — a non-expansive but easy to learn link function with i.i.d. samples — any method would require exponentially many samples (with respect to dimension of Xt) from the dynamical system.

This paper is comprehensive with both solid theoretical analysis and simulations.


**Ethical Concerns:**

Not observed

**Limitations And Societal Impact:**

Not observed any potential negative societal impact

**Main Review:**

Originality: authors studied the problem of learning non-linear dynamical systems from a single trajectory and analyzed offline and online algorithms to obtain near-optimal error guarantees. The results are original.

Quality: Overall the quality is good. However, compared to the theoretical analysis, the empirical experiments are bit too short.
It would be helpful to consider multiple different data generating system, instead of just one simple case study. In addition, it would be great if authors can provide some ablation study on the impact of the hyper-parameters of the algorithm, and how they are selected in the simulation (e.g. B = 240 and u = 10).
Also authors claimed "step sizes for GLMtron have to be chosen to be small in-order to ensure that the algorithm does not diverge", it would be helpful to add some results to help understand how the step size can influence the performance for the GLMtron in appendix, as a smaller step size usually lead to slower convergence.
Also in the theoretical analysis authors proved the convergence rate with heavy tail noise, while only studied Gaussian noise in the empirical study. It would be great to have multiple experiments with different noise.

Clarity: Overall the paper is clear. My only concern is this paper contains too much results,so a 8-page conference paper might not be the best venue compared to journal article.

Significance: this works has both theoretical contribution, and methodological contribution.

**Time Spent Reviewing:**

2

---

> ### Author Response · Authors · 2021-08-09
> **Response to reviews.**
>
> ### Regarding hyper-parameters:
> The hyper-parameters were chosen such that $B = O(\tau_{\mathsf{mix}})$ and u was kept small enough so as to not waste too many samples. Indeed, our experiments show that the choice of buffer sizes is quite robust as long as they are moderately large (i.e, B = O(t_mix)), even when they are much smaller than the buffer sizes used in the theoretical guarantees. We see this robustness as a feature of our algorithm. Experiments with various buffer sizes can be accessed in the following plot:
> https://drive.google.com/file/d/1KWreS5D-7l_fqrT6CsU4cl0w5mOcIDDo/view?usp=sharing
>
> **Explanation**:
> This shows the performance of SGD-RER with various values of the buffer size $B$. Notice that the performance remains the same for a large range of buffer sizes from $B = 90$ to $B = 2000$. However the performance degrades when the buffer size is too large (~10000). We believe this is the case since the number of buffers decreases as the buffer size increases and the output is averaged over too few number of iterates (In the case of B = 10000, the final output is just an average of 10 iterates).
>
> ### Regarding GLMtron step sizes:
> In smooth convex optimization, it is typically the case that the iterates diverge to infinity if the step size is chosen to be too large. Theoretically, this largest step-size is $\sim \frac{1}{L}$ where $L$ is the largest eigenvalue of the Hessian. In the case of GLMtron, it was experimentally observed that if the step size was chosen to be ~1.5 times the step size reported in the manuscript, the iterates diverged. Quasi Newton method essentially normalizes the gradient with the inverse of the Hessian (or rather an approximation of the Hessian) in order to let it converge faster with large step sizes. We will add a detailed discussion about the choice of step sizes in the next version of the manuscript. Experiment with various step sizes can be found here: https://drive.google.com/file/d/1ar9qaTeJE0SWGoifrvLhrhjjTSbvWX5o/view?usp=sharing
>
> **Explanation**:
> We have shown the performance of GLMtron for various step sizes and we compare it to Quasi Newton. It is seen that Quasi Newton has superior convergence and varying step sizes of GLMtron does not close this gap. In fact, in the experiment the highest possible step size for GLMtron is an order of magnitude smaller than that of Quasi Newton. The number of iterations for GLMtron and Quasi Newton are 300 and 40 respectively.
>
> ### Simulation with Heavy Tailed Noise:
>
> A typical experiment with heavy tailed noise is given below. We will add more extensive plots with various scenarios.
>
> **compute time comparison**:
> https://drive.google.com/file/d/1unXaSuJBScfHeXMUvhwUGpZbvw27O7zV/view?usp=sharing
>
> **streaming algorithm comparion**:
> https://drive.google.com/file/d/1P1tvxvyNw7BtmoVdTQr-LToU9kfmDPBt/view?usp=sharing
>
> **Explanation**:
> The typical behavior of Forward SGD, SGD-ER, SGD-RER and Quasi Newton methods seems to be the same as Sub-Gaussian noise case. However, GLMtron requires much smaller step sizes to ensure convergence and hence it takes much longer. We believe that the reason for this is related to the explanation given above for GLMtron step sizes. In the heavy tailed case, the largest eigenvalue of the Hessian $L$ depends on the quantity $X_t X_t^{\intercal}$. This is much larger than the sub-gaussian case and hence we need to pick much smaller step sizes - but further research is needed to confirm this.  We also note that we did not provide theoretical guarantees for SGD-RER in the heavy tailed noise case. But it is still seen to typically perform very well.
>
>
> ### Regarding publication in NeurIPS and too many results:
>
> Non-linear Dynamical Systems are widespread in the ML literature, especially in the context of Reinforcement Learning. So we believe that a wide exposure to the ML community along with the discussions that it entails would be helpful in building upon this effort and solving important questions in this domain. Therefore we believe that NeurIPS is a great vehicle for this work.

---

### Author Response · Authors · 2021-09-01
**Regarding Discussions**

 We would like to thank the reviewers for the detailed feedback again. It would be great if you can get back to us regarding our response soon. Looking forward to it.

---

### Decision · Program_Chairs · 2021-09-27

**Decision:**

Accept (Spotlight)

**Comment:**

The contributions were considered to be significant with high potential for impact. The main concern was that the paper contains, in a sense, "too much": there are many results, which made it harder for some reviewers to appreciate the level of contribution/improvement, and which raises the question of how to package the results into a coherent final version of the paper. Perhaps the authors can also consider submitting to a journal.